# ContiFormer: Continuous-Time Transformer for Irregular Time Series Modeling

**Yuqi Chen**[1,2,][*] **Kan Ren**[2,][†] **Yansen Wang**[2], **Yuchen Fang**[2,3], **Weiwei Sun**[1], **Dongsheng Li**[2]
[1] School of Computer Science & Shanghai Key Laboratory of Data Science, Fudan University
[2] Microsoft Research Asia, [3] Shanghai Jiao Tong University

## Abstract

Modeling continuous-time dynamics on irregular time series is critical to account for data evolution and correlations that occur continuously. Traditional methods including recurrent neural networks or Transformer models leverage inductive bias via powerful neural architectures to capture complex patterns. However, due to their discrete characteristic, they have limitations in generalizing to continuous-time data paradigms. Though neural ordinary differential equations (Neural ODEs) and their variants have shown promising results in dealing with irregular time series, they often fail to capture the intricate correlations within these sequences. It is challenging yet demanding to concurrently model the relationship between input data points and capture the dynamic changes of the continuous-time system. To tackle this problem, we propose ContiFormer that extends the relation modeling of vanilla Transformer to the continuous-time domain, which explicitly incorporates the modeling abilities of continuous dynamics of Neural ODEs with the attention mechanism of Transformers. We mathematically characterize the expressive power of ContiFormer and illustrate that, by curated designs of function hypothesis, many Transformer variants specialized in irregular time series modeling can be covered as a special case of ContiFormer. A wide range of experiments on both synthetic and real-world datasets have illustrated the superior modeling capacities and prediction performance of ContiFormer on irregular time series data. The project link is https://seqml.github.io/contiformer/.

## 1 Introduction

Irregular time series are prevalent in real-world applications like disease prevention, financial decision-making, and earthquake prediction [4, 27, 64]. Their distinctive properties set them apart from regular time series data. First, irregular time series data are characterized by irregularly generated or non-uniformly sampled observations with variable time intervals, as well as missing data due to technical issues, or data quality problems, which pose challenges for traditional time series analysis techniques [6, 55]. Second, even though the observations are irregularly sampled, the underlying data-generating process is assumed to be continuous [30, 42, 43]. Third, the relationships among the observations can be intricate and continuously evolving. All these characteristics require elaborate modeling approaches for better understanding these data and making accurate predictions.

The continuity and intricate dependency of these data samples pose significant challenges for model design. Simply dividing the timeline into equally sized intervals can severely damage the continuity of the data [36]. Recent works have suggested that underlying continuous-time process is appreciated for irregular time series modeling [6, 30, 39, 43], which requires the modeling procedure to capture

---

[*]The work was conducted during the internship of Yuqi Chen and Yuchen Fang at Microsoft Research.
[†]Correspondence to Kan Ren, contact: rk.ren@outlook.com.

the continuous dynamic of the system. Furthermore, due to the continuous nature of data flow, we argue that the correlation within the observed data is also constantly changing over time. For instance, stock prices of tech giants (e.g., MSFT and GOOG) show consistent evolving trends, yet they are affected by short-term events (e.g., large language model releases) and long-term factors.

To tackle these challenges, researchers have pursued two main branches of solutions. Neural ordinary differential equations (Neural ODEs) and state space models (SSMs) have illustrated promising abilities for capturing the dynamic change of the system over time [9, 19, 28, 49]. However, these methods overlook the intricate relationship between observations and their recursive nature can lead to cumulative errors if the number of iterations is numerous [9]. Another line of research capitalizes on the powerful inductive bias of neural networks, such as various recurrent neural networks [6, 39, 45] and Transformer models [38, 48, 54, 60]. However, their use of fixed-time encoding or learning upon certain kernel functions fails to capture the complicated input-dependent dynamic systems.

Based on the above analysis, modeling the relationship between observations while capturing the temporal dynamics is a challenging task. To address this problem, we propose a Continuous-Time Transformer, namely *ContiFormer*, which incorporates the modeling abilities of continuous dynamics of Neural ODEs within the attention mechanism of Transformers and breaks the discrete nature of Transformer models. Specifically, to capture the dynamics of the observations, ContiFormer begins by defining latent trajectories for each observation in the given irregularly sampled data points. Next, to capture the intricate yet continuously evolving relationship, it extends the discrete dot-product in Transformers to a continuous-time domain, where attention is calculated between continuous dynamics. With the proposed attention mechanism and Transformer architecture, ContiFormer effectively models complex continuous-time dynamic systems.

The contributions of this paper can be summarized below.

- *Continuous-Time Transformer.* To the best of our knowledge, we are the first to incorporate a continuous-time mechanism into attention calculation in Transformer, which is novel and captures the continuity of the underlying system of the irregularly sampled time-series data.

- *Parallelism Modeling.* To tackle the conflicts between continuous-time calculation and the parallel calculation property of the Transformer model, we propose a novel reparameterization method, allowing us to parallelly execute the continuous-time attention in the different time ranges.

- *Theoretical Analysis.* We mathematically characterize that various Transformer variants [34, 38, 48, 54, 60] can be viewed as special instances of ContiFormer. Thus, our approach offers a broader scope that encompasses Transformer variants.

- *Experiment Results.* We examine our method on various irregular time series settings, including time-series interpolation, classification, and prediction. The extensive experimental results have illustrated the superior performance of our method against strong baseline models.

## 2   Related Work

**Time-Discretized Models.**    There exists a branch of models based on the time-discrete assumption which transforms the time-space into a discrete one. Recurrent Neural Networks (RNN) [14, 24] and Transformers [51] are powerful time-discrete sequence models, which have achieved great success in natural language processing [15, 37, 58], computer vision [20, 63] and time series forecasting [31, 35, 52, 53, 56, 61, 62]. Utilizing them for irregular time series data necessitates discretizing the timeline into time bins or padding missing values, potentially resulting in information loss and disregarding inter-observation dynamics. [13, 36, 41]. An alternative is to construct models that can utilize these various time intervals [34, 60]. Additionally, several recent approaches have also encoded time information into features to model irregularly sampled time series, including time representation approaches [54, 60] and kernel-based methods [38], which allow learning for time dynamics. For instance, Mercer [54] concatenates the event embedding with the time representation. Despite these advances, these models are still insufficient in capturing input-dependent dynamic systems.

**Continuous-Time Models.**    Other works shift to the continuous-time modeling paradigm. To overcome the discrete nature of RNNs, a different strategy involves the exponential decay of the hidden state between observations [6, 39]. Nevertheless, the applicability of these techniques is restricted to decaying dynamics specified between successive observations [9]. Neural ODEs are a class of

models that leverage the theory of ordinary differential equations to define the dynamics of neural networks [9, 27]. Neural CDE [30] and Neural RDE [40] further utilize the well-defined mathematics of controlled differential equations [2, 29] and rough path theory [5] to construct continuous-time RNN. ODE-RNN [43] combines Neural ODE with Gated Recurrent Unit (GRU) [12] to model continuous-time dynamics and input changes. Whereas these approaches usually fail to capture the evolving relationship between observations [45]. CADN [11] and ANCDE [26] extend ODE-RNN and Neural CDE by incorporating an attention mechanism to capture the temporal relationships, respectively. Another line of research explores the use of state space models (SSMs) to represent continuous-time dynamic systems [1, 18, 19, 49]. However, these models adopt a recursive schema, which hampers its efficiency and can lead to cumulative errors [9]. In contrast, our proposed model, ContiFormer leverages the parallelism advantages of the Transformer architecture and employs a non-autoregressive paradigm. Furthermore, extending the Transformer to the continuous-time domain remains largely unexplored by researchers. Such extension provides a more flexible and powerful modeling ability for irregular time series data.

## 3  Method

The ContiFormer is a Transformer [51] model for processing time series data with irregular time intervals. Formally, an irregular time series is defined as $\Gamma = [(X_1, t_1), \ldots, (X_N, t_N)]$, where observations may occur at any time and the observation time points $\boldsymbol{\omega} = (t_1, \ldots, t_N)$ are with irregular intervals. $X_i$ is the input feature for the $i$-th observation. We denote $X = [X_1; X_2; \ldots, X_N] \in \mathbb{R}^{N \times d}$.

As aforementioned, it is challenging yet demanding to concurrently model the relationship between input data points and capture the dynamic changes of the continuous-time system. To tackle the challenge, we propose a Continuous-Time Transformer architecture as shown in Figure 1. Each layer of ContiFormer takes as input an irregular time series $X$ and the sampled time $\boldsymbol{\omega}$ and outputs a latent continuous trajectory that captures the dynamic change of the underlying system. With such a design, ContiFormer transforms the discrete observation sequence into the continuous-time domain. At each layer, the core attention module takes a continuous perspective and expands the dot-product operation in the vanilla Transformer to the continuous-time domain, which not only models the underlying continuous dynamics but also captures the evolving input-dependent process. For better understanding, the description of each layer will omit the layer index, without causing confusion.

Throughout the remaining section, we adopt the notation $\boldsymbol{\omega}$ to denote a sequence of (reference) time points, while $t$ is the random variable representing a query time point. Matrices are donated using uppercase letters, and lowercase letters except $t$ represent continuous functions.

### 3.1  Continuous-Time Attention Mechanism

The core of the ContiFormer layer is the proposed continuous-time multi-head attention (CT-MHA) module, as shown in Figure 1. At each layer, we first transform the input irregular time series $X$ into $Q = [Q_1; Q_2; \ldots; Q_N]$ for queries, $K = [K_1; K_2; \ldots; K_N]$ for keys, and $V = [V_1; V_2; \ldots; V_N]$ for values. At a high level, the CT-MHA module transforms the irregular time series inputs to latent trajectories and outputs a continuous dynamic system that captures the time-evolving relationship between observations. To accomplish this, it utilizes ordinary differential equations to define the latent trajectories for each observation. Within the latent state, it assumes that the underlying dynamics evolve following linear ODEs. Subsequently, it constructs a continuous query function by approximating the underlying sample process of the input. Ultimately, it produces a continuous-time function that captures the evolving relationship and represents the complex dynamic system.

**Continuous Dynamics from Observations**  Transformer calculates scaled dot-product attention for queries, keys, and values [51]. In continuous form, we first employ ordinary differential equations to define the latent trajectories for each observation. Specifically, assuming that the first observation and last observation come at time point $t_1$ and $t_N$ respectively, we define continuous keys and values as

$$
\begin{aligned}
\mathbf{k}_i(t_i) = K_i \,, \quad \mathbf{k}_i(t) = \mathbf{k}_i(t_i) + \int_{t_i}^{t} f\left(\tau, \mathbf{k}_i(\tau); \theta_k\right) \mathrm{d}\tau \,, \\
\mathbf{v}_i(t_i) = V_i \,, \quad \mathbf{v}_i(t) = \mathbf{v}_i(t_i) + \int_{t_i}^{t} f\left(\tau, \mathbf{v}_i(\tau); \theta_v\right) \mathrm{d}\tau \,,
\end{aligned}
\tag{1}
$$

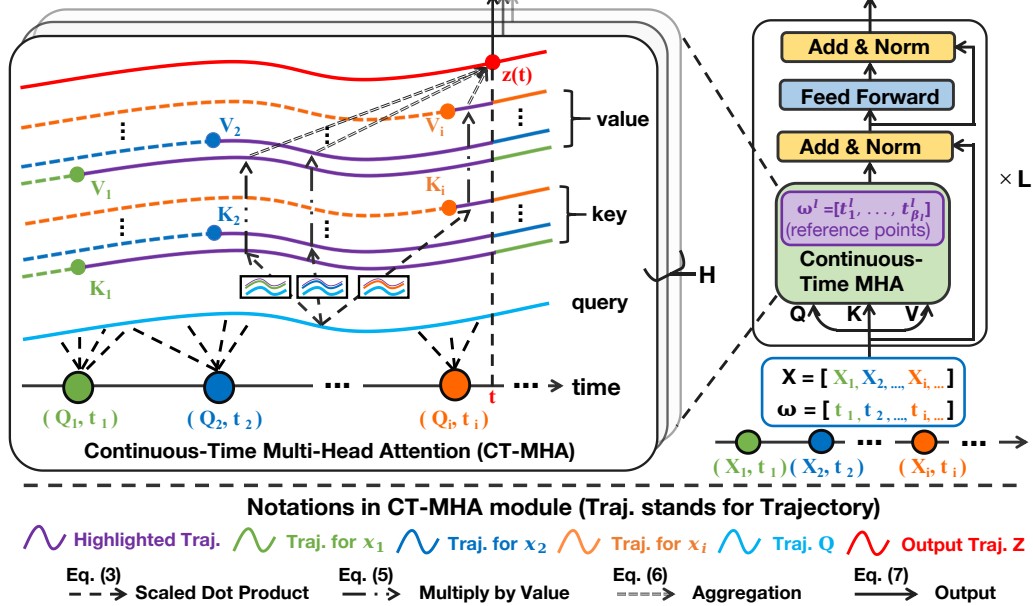

Figure 1: Architecture of the ContiFormer layer. ContiFormer takes an irregular time series and its corresponding sampled time points as input. Queries, keys, and values are obtained in continuous-time form. The attention mechanism (CT-MHA) performs a scaled inner product in a continuous-time manner to capture the evolving relationship between observations, resulting in a complex continuous dynamic system. Feedforward and layer normalization are adopted, similar to the Transformer. Finally, a sampling trick is employed to make ContiFormer stackable. Note that the highlighted trajectories in purple indicate the part of functions that are involved in the calculation of the output.

where $t \in [t_1, t_N]$, $\mathbf{k}_i(\cdot), \mathbf{v}_i(\cdot) \in \mathbb{R}^d$ represent the ordinary differential equation for the $i$-th observation with parameters $\theta_k$ and $\theta_v$, and the initial state of $\mathbf{k}_i(t_i)$ and $\mathbf{v}_i(t_i)$ respectively and the function $f(\cdot) \in \mathbb{R}^{d+1} \to \mathbb{R}^d$ controls the change of the dynamics.

**Query Function**   While keys and values are associated with the input and output of the attention mechanism, a query specifies what information to extract from the input. To model a dynamic system, queries can be modeled as a function of time that represents the overall changes in the input to the dynamic system at a specific time point $t$, i.e., $\mathbf{q}(t)$. Specifically, we adopt a common assumption that irregular time series is a discretization of an underlying continuous-time process. Thus, similar to [30], we define a closed-form continuous-time interpolation function (e.g., natural cubic spline) with knots at $t_1, \dots, t_N$ such that $\mathbf{q}(t_i) = Q_i$ as an approximation of the underlying process.

**Scaled Dot Product**   The self-attention mechanism is the key component in Transformer architecture. At its core, self-attention involves calculating the correlation between queries and keys. This is achieved through the inner product of two matrices in discrete form, i.e., $Q \cdot K^\top$. Extending the discrete inner-product to its continuous-time domain, given two real functions $f(x)$ and $g(x)$, we define the inner product of two functions in a closed interval $[a, b]$ as

$$\langle f, g \rangle = \int_a^b f(x) \cdot g(x) \mathrm{d}x . \tag{2}$$

Intuitively, it can be thought of as a way of quantifying how much the two functions "align" with each other over the interval. Inspired by the formulation of the inner product in the continuous-time domain, we model the evolving relationship between the $i$-th sample and the dynamic system at time point $t$ as the inner product of $\mathbf{q}$ and $\mathbf{k}_i$ in a closed interval $[t_i, t]$, i.e.,

$$\boldsymbol{\alpha}_i(t) = \frac{\int_{t_i}^t \mathbf{q}(\tau) \cdot \mathbf{k}_i(\tau)^\top \mathrm{d}\tau}{t - t_i} . \tag{3}$$

Due to the nature of sequence data, it is common to encounter abnormal points, such as events with significantly large time differences. To avoid numeric instability during training, we divide the

integrated solution by the time difference. As a consequence, Eq. (3) exhibits discontinuity at $\boldsymbol{\alpha}_i(t_i)$. To ensure the continuity of the function $\boldsymbol{\alpha}_i(\cdot)$, we define $\boldsymbol{\alpha}_i(t_i)$ as

$$\boldsymbol{\alpha}_i(t_i) = \lim_{\epsilon \to 0} \frac{\int_{t_i}^{t_i+\epsilon} \mathbf{q}(\tau) \cdot \mathbf{k}_i(\tau)^\top \mathrm{d}\tau}{\epsilon} = \mathbf{q}(t_i) \cdot \mathbf{k}_i(t_i)^\top \ . \tag{4}$$

**Expected Values**   Given a query time $t \in [t_1, t_N]$, the value of an observation at time point $t$ is defined as the expected value from $t_i$ to $t$. Without loss of generality, it holds that the expected values of an observation sampled at time $t_i$ is $\mathbf{v}_i(t_i)$ or $V_i$. Formally, the expected value is defined as

$$\widehat{\mathbf{v}}_i(t) = \mathbb{E}_{t \sim [t_i, t]}\left[\mathbf{v}_i(t)\right] = \frac{\int_{t_i}^{t} \mathbf{v}_i(\tau) \mathrm{d}\tau}{t - t_i} \ . \tag{5}$$

**Multi-Head Attention**   The continuous-time attention mechanism is a powerful tool used in machine learning that allows for the modeling of complex, time-varying relationships between keys, queries, and values. Unlike traditional attention mechanisms that operate on discrete time steps, continuous-time attention allows for a more fine-grained analysis of data by modeling the input as a continuous function of time. Specifically, given the forehead-defined queries, keys, and values in continuous-time space, the continuous-time attention given a query time $t$ can be formally defined as

$$\mathrm{CT\text{-}ATTN}(Q, K, V, \boldsymbol{\omega})(t) = \sum_{i=1}^{N} \widehat{\boldsymbol{\alpha}}_i(t) \cdot \widehat{\mathbf{v}}_i(t) \ ,$$

$$\text{where } \widehat{\boldsymbol{\alpha}}_i(t) = \frac{\exp\left(\boldsymbol{\alpha}_i(t)/\sqrt{d_k}\right)}{\sum_{j=1}^{N} \exp\left(\boldsymbol{\alpha}_j(t)/\sqrt{d_k}\right)} \ . \tag{6}$$

Multi-head attention, an extension of the attention mechanism [51], allows simultaneous focus on different input aspects. It stabilizes training by reducing attention weight variance. We extend Eq. (6) by incorporating multiple sets of attention weights, i.e.,

$$\mathrm{CT\text{-}MHA}(Q, K, V, \boldsymbol{\omega})(t) = \mathrm{Concat}\left(\mathrm{head}_{(1)}(t), \ldots, \mathrm{head}_{(\mathrm{H})}(t)\right) W^O \ ,$$

$$\text{where } \mathrm{head}_{(\mathrm{h})}(t) = \mathrm{CT\text{-}ATTN}\left(Q W_{(\mathrm{h})}^Q, K W_{(\mathrm{h})}^K, V W_{(\mathrm{h})}^V, \boldsymbol{\omega}\right)(t) \ , \tag{7}$$

where $W^O$, $W_{(\mathrm{h})}^Q$, $W_{(\mathrm{h})}^K$ and $W_{(\mathrm{h})}^V$ are parameter matrices, $\mathrm{h} \in [1, \mathrm{H}]$ and H is the head number.

### 3.2   Continuous-Time Transformer

Despite the widespread adoption of Transformers [51] in various research fields, their extension for modeling continuous-time dynamic systems is underexplored. We propose ContiFormer that directly builds upon the original implementation of vanilla Transformer while extending it to the continuous-time domain. Specifically, we apply layer normalization (LN) after both the multi-head self-attention (MHA) and the feed-forward blocks (FFN). We formally characterize the ContiFormer layer below

$$\tilde{\mathbf{z}}^l(t) = \mathrm{LN}\left(\mathrm{CT\text{-}MHA}(X^l, X^l, X^l, \boldsymbol{\omega}^l)(t) + \mathbf{x}^l(t)\right) \ ,$$

$$\mathbf{z}^l(t) = \mathrm{LN}\left(\mathrm{FFN}(\tilde{\mathbf{z}}^l(t)) + \tilde{\mathbf{z}}^l(t)\right) \ , \tag{8}$$

where $\mathbf{z}^l(t)$ is the output from the $l$-th ContiFormer layer at time point $t$. Additionally, we adopt residual connection to avoid potential gradient vanishing [23]. To incorporate the residual connection with the continuous-time output, we approximate the underlying process of the discrete input $X^l$ using a continuous function $\mathbf{x}^l(t)$ based on the closed-form continuous-time interpolation function.

**Sampling Process**   As described before, each ContiFormer layer derives a continuous function $\mathbf{z}^l(t)$ w.r.t. time as the output, while receiving discrete sequence $X^l$ as input. However, $\mathbf{z}^l(t)$ can not be directly incorporated into neural network architectures that expect inputs in the form of fixed-dimensional vectors and sequences [48], which places obstacles when stacking layers of ContiFormer. To address this issue, we establish reference time points for the output of each layer. These points are used to discretize the layer output, and can correspond to either the input time points (i.e., $\boldsymbol{\omega}$) or task-specific time points. Specifically, assume that the reference points for the $l$-th layer is $\boldsymbol{\omega}^l = [t_1^l, t_2^l, ..., t_{\beta_l}^l]$, the input to the next layer $X^{l+1}$ can be sampled as $\{\mathbf{z}^l(t_j^l) | j \in [1, \beta_l]\}$.

## 3.3 Complexity Analysis

We consider an autoregressive task where the output of each observation is required for a particular classification or regression task. Therefore, the reference points for each layer are defined as $\boldsymbol{\omega}^l = \boldsymbol{\omega}$.

To preserve the parallelization of Transformer architecture and meanwhile implement the continuous-time attention mechanism in Eq. (6), we first adopt time variable ODE [8] to reparameterize ODEs into a single interval $[-1, 1]$, followed by numerical approximation method to approximate the integrals, i.e.,

Table 1: Per-layer complexity (Comp. per Layer), minimum number of sequential operations (Seq. Op.), and maximum path lengths (Max. Path Len.). $N$ is the sequence length, $d$ is the representation dimension, $T$ is the number of function evaluations (NFE) for the ODE solver in a single forward pass from $t_1$ to $t_N$, and $S$ represents the NFE from $-1$ to $1$.

| Model | Comp. per Layer | Seq. Op. | Max. Path Len. |
|---|---|---|---|
| Transformer [51] | $O(N^2 \cdot d)$ | $O(1)$ | $O(1)$ |
| RNN [25] | $O(N \cdot d^2)$ | $O(N)$ | $O(N)$ |
| Neural ODE [9] | $O(T \cdot d^2)$ | $O(T)$ | $O(T)$ |
| ContiFormer | $O(N^2 \cdot S \cdot d^2)$ | $O(S)$ | $O(1)$ |

$$\boldsymbol{\alpha}_i(t_j) = \frac{\int_{t_i}^{t_j} \mathbf{q}(\tau) \cdot \mathbf{k}_i(\tau)^\top d\tau}{t_j - t_i} = \frac{1}{2} \int_{-1}^{1} \tilde{\mathbf{q}}_{i,j}(\tau) \cdot \tilde{\mathbf{k}}_{i,j}(\tau)^\top d\tau \approx \frac{1}{2} \sum_{p=1}^{P} \gamma_p \tilde{\mathbf{q}}_{i,j}(\xi_p) \cdot \tilde{\mathbf{k}}_{i,j}(\xi_p)^\top \,, \quad (9)$$

where $P$ denotes the number of intermediate steps to approximate an integral, $\xi_p \in [-1, 1]$ and $\gamma_p \geq 0$, $\tilde{\mathbf{k}}_{i,j}(s) = \mathbf{k}_i\left((s(t_j - t_i) + t_i + t_j)/2\right)$, and $\tilde{\mathbf{q}}_{i,j}(s) = \mathbf{q}_i\left((s(t_j - t_i) + t_i + t_j)/2\right)$. Finally, $\boldsymbol{\alpha}_i(t_j)$ can be solved with one invokes of ODESolver that contains $N^2$ systems, i.e.,

$$\underbrace{\begin{bmatrix} \tilde{\mathbf{k}}_{1,1}(\xi_{p-1}) \\ \vdots \\ \tilde{\mathbf{k}}_{N,N}(\xi_{p-1}) \end{bmatrix}}_{\mathbf{K}(\xi_{p-1})} + \int_{\xi_{p-1}}^{\xi_p} f(s, \mathbf{K}(s); \theta_k) ds = \underbrace{\begin{bmatrix} \tilde{\mathbf{k}}_{1,1}(\xi_p) \\ \vdots \\ \tilde{\mathbf{k}}_{N,N}(\xi_p) \end{bmatrix}}_{\mathbf{K}(\xi_p)} , \quad (10)$$

where $p \in [1, ..., P]$ and we further define $\xi_0 = -1$. Additionally, $\tilde{\mathbf{q}}_{i,j}(\xi_p)$ can be obtained by interpolating on the close-form continuous-time query function. Similar approach is adopted for solving $\hat{\mathbf{v}}_i(t_j)$ in Eq. (5). The detailed implementation is deferred to Appendix A.

We consider three criteria to analyze the complexity of different models, i.e., per-layer complexity, minimum number of sequential operations, and maximum path lengths [51]. Complexity per layer refers to the computational resources required to process within a single layer of a neural network. Sequential operation refers to the execution of operations that are processed iteratively or in a sequential manner. Maximum path length measures the longest distance information must traverse to establish relationships, reflecting the model's ability to learn long-range dependencies.

As noted in Table 1, vanilla Transformer is an efficient model for sequence learning with $O(1)$ sequential operations and $O(1)$ path length because of the parallel attention mechanism, whereas the recurrent nature of RNN and Neural ODE rely on autoregressive property and suffer from a large number of sequential operations. Utilizing Eq. (9) and leveraging the parallelism inherent in the Transformer architecture, our model, ContiFormer, achieves $O(S)$ sequential operations and $O(1)$ path length, while also enjoying the advantage of capturing complex continuous-time dynamic systems. We have $S \ll T$ and we generally set $S < N$ in our experiments.

## 3.4 Representation Power of ContiFormer

In the previous subsections, we introduce a novel framework for modeling dynamic systems. Then a natural question is: *How powerful is ContiFormer?* We observe that by choosing proper weights, ContiFormer can be an extension of vanilla Transformer [51]. Further, many Transformer variants tailored for irregular time series, including time embedding methods [48, 54] and kernelized attention methods [34, 38], can be special cases of our model. We provide an overview of the main theorem below and defer the proof to Appendix B.

**Theorem 1** (Universal Attention Approximation Theorem). *Given query $(Q)$ and key $(K)$ matrices, such that $\|Q_i\|_2 < \infty, \|Q_i\|_0 = d$ for $i \in [1, ..., N]$. For certain attention matrix, i.e.,*

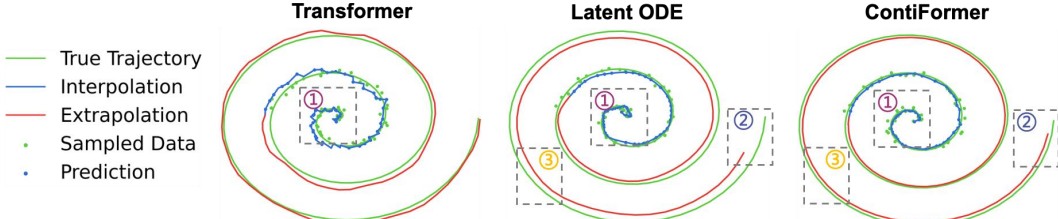

Figure 2: Interpolation and extrapolation of spirals with irregularly-samples time points by Transformer, Neural ODE, and our model.

$\mathrm{Attn}(Q, K) \in \mathbb{R}^{N \times N}$ *(see Appendix B.1 for more information), there always exists a family of continuously differentiable vector functions* $\mathbf{k}_1(\cdot), \mathbf{k}_2(\cdot), \ldots, \mathbf{k}_N(\cdot)$, *such that the discrete definition of the continuous-time attention formulation, i.e.,* $[\boldsymbol{\alpha}_1(\cdot), \boldsymbol{\alpha}_2(\cdot), ..., \boldsymbol{\alpha}_N(\cdot)]$ *in Eq. (4), given by*

$$\widetilde{\mathrm{Attn}}(Q, K) = \left[ \begin{array}{cccc} \boldsymbol{\alpha}_1(t_1) & \boldsymbol{\alpha}_2(t_1) & \cdots & \boldsymbol{\alpha}_N(t_1) \\ \boldsymbol{\alpha}_1(t_2) & \boldsymbol{\alpha}_2(t_2) & \cdots & \boldsymbol{\alpha}_N(t_2) \\ \vdots & \vdots & \ddots & \vdots \\ \boldsymbol{\alpha}_1(t_N) & \boldsymbol{\alpha}_2(t_N) & \cdots & \boldsymbol{\alpha}_N(t_N) \end{array} \right], \tag{11}$$

*satisfies that* $\widetilde{\mathrm{Attn}}(Q, K) = \mathrm{Attn}(Q, K)$.

## 4 Experiments

In this section, we evaluate ContiFormer on three types of tasks on irregular time series data, i.e., interpolation and extrapolation, classification, event prediction, and forecasting. Additionally, we conduct experiments on pendulum regression task [19, 45], the results are listed in Appendix C.5.

**Implementation Details**   By default, we use the natural cubic spline to construct the continuous-time query function. The vector field in ODE is defined as $f(t, \mathbf{x}) = \mathrm{Actfn}(\mathrm{LN}(\mathrm{Linear}^{d,d}(\mathrm{Linear}^{d,d}(\mathbf{x}) + \mathrm{Linear}^{1,d}(t))))$, where $\mathrm{Actfn}(\cdot)$ is either *tanh* or *sigmoid* activation function, $\mathrm{Linear}^{a,b}(\cdot) : \mathbb{R}^a \to \mathbb{R}^b$ is a linear transformation from dimension $a$ to dimension $b$, LN denotes the layer normalization. We adopt the Gauss-Legendre Quadrature approximation to implement Eq. (9). In the experiment, we choose the Runge-Kutta-4 [44] (RK4) algorithm to solve the ODE with a fixed step of fourth order and a step size of 0.1. Thus, the number of forward passes to integrate from $-1$ to $1$, i.e., $S$ in Table 1, is 80. All the experiments were carried out on a single 16GB NVIDIA Tesla V100 GPU.

### 4.1 Modeling Continuous-Time Function

The first experiment studies the effectiveness of different models from different categories on continuous-time function approximation. We generated a dataset of 300 2-dimensional spirals, sampled at 150 equally-spaced time points. To generate irregular time points, we randomly sample 50 points from each subsampled trajectory without replacement. We visualize the interpolation and extrapolation results of the irregular time series for different models, namely Latent ODE (w/ RNN encoder) [9], Transformer [51] and our proposed

Table 2: Interpolation and extrapolation results of different models on 2-dimensional spirals. ↓ indicates the lower the better.

| Metric | Transformer [51] | Latent ODE [9] | **ContiFormer** |
|---|---|---|---|
| Task = *Interpolation* ($\times 10^{-2}$) | | | |
| RMSE ($\downarrow$) | $1.37 \pm 0.14$ | $2.09 \pm 0.22$ | $\mathbf{0.49 \pm 0.06}$ |
| MAE ($\downarrow$) | $1.42 \pm 0.13$ | $1.95 \pm 0.25$ | $\mathbf{0.52 \pm 0.06}$ |
| Task = *Extrapolation* ($\times 10^{-2}$) | | | |
| RMSE ($\downarrow$) | $1.36 \pm 0.10$ | $1.59 \pm 0.05$ | $\mathbf{0.64 \pm 0.09}$ |
| MAE ($\downarrow$) | $1.49 \pm 0.12$ | $1.52 \pm 0.05$ | $\mathbf{0.65 \pm 0.08}$ |

ContiFormer. Besides, we list the interpolation and extrapolation results in Table 2 using rooted mean squared error (RMSE) and mean absolute error (MAE) metrics. More visualization results and experimental settings can be found in Appendix C.1. As shown in Figure 2, both Latent ODE and ContiFormer can output a smooth and continuous function approximation, while Transformer fails to interpolate it given the noisy observations (①). From Table 2, we can observe that our model outperforms both Transformer and Latent ODE by a large margin. The improvement lies in two

Table 3: Experimental results on irregular time series classification. Avg. ACC. stands for average accuracy over 20 datasets and Avg. Rank stands for average ranking over 20 datasets. ↑ (↓) indicates the higher (lower) the better.

| Mask Ratio | 30% Dropped | | 50% Dropped | | 70% Dropped | |
|---|---|---|---|---|---|---|
| Metric | Avg. ACC. (↑) | Avg. Rank (↓) | Avg. ACC. (↑) | Avg. Rank (↓) | Avg. ACC. (↑) | Avg. Rank (↓) |
| GRU-D [30] | 0.7284 | 6 | 0.7117 | 5.8 | 0.6725 | 6.15 |
| GRU-$\Delta$t [6] | 0.7298 | 5.75 | 0.7157 | 5.65 | 0.6795 | 5.7 |
| ODE-RNN [43] | 0.7304 | 5.45 | 0.7000 | 5.55 | 0.6594 | 6.4 |
| Neural CDE [30] | 0.7142 | 6.85 | 0.6929 | 6.5 | 0.6753 | 6.3 |
| mTAN [48] | 0.7381 | 5.7 | 0.7118 | 5.85 | 0.6955 | 5.4 |
| CADN [10] | 0.7402 | 5.4 | 0.7211 | 5.55 | 0.7183 | 3.9 |
| S5 [49] | 0.7854 | 4.4 | 0.7638 | 4.25 | 0.7401 | 4.45 |
| TST [57] | 0.8089 | 2.75 | 0.7793 | 3.15 | 0.7297 | 4.2 |
| **ContiFormer** | **0.8126** | **2.4** | **0.7997** | **1.9** | **0.7749** | **2.1** |

aspects. First, compared to Transformer, our ContiFormer can produce an almost[3] continuous-time output, making it more suitable for modeling continuous-time functions. Second, compared to Latent ODE, our ContiFormer excels at retaining long-term information (②), leading to lower prediction error in extrapolating unseen time series. Conversely, Latent ODE is prone to cumulative errors (③), resulting in poorer performance in extrapolation tasks. Therefore, we conclude that ContiFormer is a more suitable approach for modeling continuous-time functions on irregular time series.

## 4.2 Irregularly-Sampled Time Series Classification

The second experiment evaluates our model on real-world irregular time series data. To this end, we first examine the effectiveness of different models for irregular time series classification. We select 20 datasets from UEA Time Series Classification Archive [3] with diverse characteristics in terms of the number, dimensionality, and length of time series samples. More detailed information and experimental results can be found in Appendix C.2. To generate irregular time series data, we follow the setting of [30] to randomly drop either 30%, 50% or 70% observations.

We compare the performance of ContiFormer with RNN-based methods (GRU-$\Delta$t [30], GRU-D [6]), Neural ODE-based methods (ODE-RNN [43], CADN [10], Neural CDE [30]), SSM-based models (S5 [49]), and attention-based methods (TST [57], mTAN [48]).

Table 3 presents the average accuracy and rank number under different drop ratios. ContiFormer outperforms all the baselines on all three settings. The complete results are shown in Appendix C.2.3. Moreover, attention-based methods perform better than both RNN-based methods and ODE-based methods, underscoring the significance of effectively modeling the intercorrelation of the observations. Additionally, we find that our method is more robust than other models. ContiFormer exhibits the smallest performance gap between 30% and 50%.

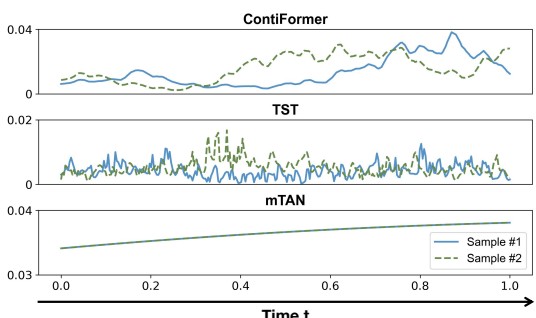

Besides, we study the learned attention patterns with different models. Attention captures the impact of previous observations on the future state of a system by measuring the correlation between different observations at different time points. Consequently, it is expected that a model

Figure 3: Visualization of attention scores on UWaveGestureLibrary dataset. Colors indicate the attention scores for different instances at time $t = 0$. Observations at time $t = 0$ are observed and normalize the time interval to $[0, 1]$.

effectively capturing the underlying dynamics would exhibit continuous, smooth, and input-dependent intermediate outcomes. As illustrated in Figure 3, ContiFormer and TST are both capable of

---

[3]The output of ContiFormer can be considered "continuous" only if we overlook the approximation and numerical errors in the ODESolver. Additionally, traditional Transformers are commonly trained with dropout to prevent overfitting, which can result in discontinuous outputs. To mitigate this issue, a straightforward approach is to set the dropout rate to 0, thereby producing an "almost" continuous output for ContiFormer.

Table 4: Prediction result of compared models for event prediction. LL for log-likelihood and ACC for accuracy. ↑ (↓) indicates the higher (lower) the better. (bold values indicate the best performance and $^+$ indicates outperforming the best baseline by at least 3 standard deviations.)

| Model | Metric | Synthetic | Neonate | Traffic | MIMIC | StackOverflow | BookOrder |
|---|---|---|---|---|---|---|---|
| HP [22] | LL (↑) | -3.084 ± .005 | -4.618 ± .005 | -1.482 ± .005 | -4.618 ± .005 | -5.794 ± .005 | -1.036 ± .000 |
| | Accuracy (↑) | 0.756 ± .000 | – | 0.570 ± .000 | 0.795 ± .000 | 0.441 ± .000 | 0.604 ± .000 |
| | RMSE (↓) | 0.953 ± .000 | 10.957 ± .012 | 0.407 ± .000 | 1.021 ± .000 | 1.341 ± .000 | 3.781 ± .000 |
| RMTPP [16] | LL (↑) | -1.025 ± .030 | -2.817 ± .023 | -0.546 ± .012 | -1.184 ± .023 | -2.374 ± .001 | -0.952 ± .007 |
| | Accuracy (↑) | 0.841 ± .000 | – | 0.805 ± .002 | 0.823 ± .004 | 0.461 ± .000 | 0.624 ± .000 |
| | RMSE (↓) | 0.369 ± .014 | 9.517 ± .023 | 0.337 ± .001 | 0.864 ± .017 | 0.955 ± .000 | 3.647 ± .003 |
| NeuralHP [39] | LL (↑) | -1.371 ± .004 | -2.795 ± .012 | -0.643 ± .004 | -1.239 ± .027 | -2.608 ± .000 | -1.104 ± .005 |
| | Accuracy (↑) | 0.841 ± .000 | – | 0.759 ± .001 | 0.814 ± .001 | 0.450 ± .000 | 0.621 ± .000 |
| | RMSE (↓) | 0.631 ± .002 | 9.614 ± .013 | 0.358 ± .001 | 0.846 ± .007 | 1.022 ± .000 | 3.734 ± .003 |
| SAHP [54] | LL (↑) | -0.619 ± .063 | -2.646 ± .057 | 0.372 ± .022 | **-1.110 ± .030** | -2.404 ± .002 | -0.304 ± .002 |
| | Accuracy (↑) | 0.841 ± .000 | – | 0.780 ± .001 | 0.830 ± .004 | 0.455 ± .000 | 0.622 ± .000 |
| | RMSE (↓) | 0.521 ± .055 | 9.403 ± .060 | 0.337 ± .000 | 0.851 ± .006 | 0.963 ± .000 | 3.676 ± .011 |
| THP [64] | LL (↑) | -0.589 ± .017 | -2.702 ± .045 | -0.569 ± .015 | -1.137 ± .038 | -2.354 ± .001 | -1.102 ± .052 |
| | Accuracy (↑) | 0.841 ± .000 | – | 0.818 ± .001 | 0.834 ± .005 | -0.468 ± .000 | 0.622 ± .004 |
| | RMSE (↓) | 0.205 ± .006 | 9.471 ± .045 | 0.332 ± .000 | 0.843 ± .017 | 0.951 ± .000 | 3.688 ± .004 |
| NSTKA [38] | LL (↑) | -1.001 ± .025 | -2.747 ± .061 | -0.667 ± .027 | -1.188 ± .012 | -2.406 ± .003 | -1.098 ± .019 |
| | Accuracy (↑) | 0.842 ± .000 | – | 0.767 ± .001 | 0.833 ± .003 | 0.465 ± .000 | 0.621 ± .000 |
| | RMSE (↓) | 0.379 ± .015 | 9.502 ± .049 | 0.346 ± .001 | 0.842 ± .008 | 0.956 ± .000 | 3.731 ± .007 |
| GRU-Δt [9] | LL (↑) | -0.871 ± .050 | -2.736 ± .031 | -0.613 ± .062 | -1.164 ± .026 | -2.389 ± .002 | -0.915 ± .006 |
| | Accuracy (↑) | 0.841 ± .000 | – | 0.800 ± .004 | 0.832 ± .007 | 0.466 ± .000 | 0.627 ± .000 |
| | RMSE (↓) | 0.249 ± .013 | 9.421 ± .050 | 0.335 ± .001 | 0.850 ± .010 | 0.950 ± .000 | 3.666 ± .016 |
| ODE-RNN [43] | LL (↑) | -1.032 ± .102 | -2.732 ± .080 | -0.491 ± .011 | -1.183 ± .028 | -2.395 ± .001 | -0.988 ± .006 |
| | Accuracy (↑) | 0.841 ± .000 | – | 0.812 ± .000 | 0.827 ± .006 | 0.467 ± .000 | 0.624 ± .000 |
| | RMSE (↓) | 0.342 ± .030 | 9.289 ± .048 | 0.334 ± .000 | 0.865 ± .021 | 0.952 ± .000 | **3.605 ± .004** |
| mTAN [48] | LL (↑) | -0.920 ± .036 | -2.722 ± .026 | -0.548 ± .023 | -1.149 ± .029 | -2.391 ± .002 | -0.980 ± .004 |
| | Accuracy (↑) | 0.842 ± .000 | – | 0.811 ± .002 | 0.832 ± .009 | 0.466 ± .000 | 0.620 ± .000 |
| | RMSE (↓) | 0.286 ± .008 | 9.363 ± .042 | 0.334 ± .001 | 0.848 ± .006 | 0.950 ± .000 | 3.680 ± .015 |
| ContiFormer | LL (↑) | **-0.535 ± .028$^+$** | **-2.550 ± .026** | **0.635 ± .019$^+$** | -1.135 ± .023 | **-2.332 ± .001$^+$** | **-0.270 ± .010$^+$** |
| | Accuracy (↑) | **0.842 ± .000** | – | **0.822 ± .001$^+$** | **0.836 ± .006** | **0.473 ± .000$^+$** | **0.628 ± .001$^+$** |
| | RMSE (↓) | **0.192 ± .005** | **9.233 ± .033** | **0.328 ± .001$^+$** | **0.837 ± .007** | **0.948 ± .000$^+$** | 3.614 ± .020 |

capturing input-dependent patterns effectively, while mTAN struggles due to the fact that its attention mechanism depends solely on the time information. In addition, ContiFormer excels in capturing complex and smooth patterns compared to TST, showcasing its ability to model time-evolving correlations effectively and potentially more robust to noisy data. Consequently, we believe that ContiFormer better captures the nature of irregular time series.

## 4.3 Predicting Irregular Event Sequences

Next, we evaluate different models for predicting the type and occurrence time of the next event with irregular event sequences [7, 21, 46, 47], a.k.a, marked temporal point process (MTPP) task. To this end, we use one synthetic dataset and five real-world datasets for evaluation, namely Synthetic, Neonate [50], Traffic [32], MIMIC [16], BookOrder [16] and StackOverflow [33]. We use the 4-fold cross-validation scheme for Synthetic, Neonate, and Traffic datasets following [17], and the 5-fold cross-validation scheme for the other three datasets following [39, 64]. We repeat the experiment 3 times and report the mean and standard deviations in Table 4. More information about the task description, dataset information, and training algorithm can be found in Appendix C.3.

We first compare ContiFormer with 3 models on irregular time series, i.e., GRU-Δt, ODE-RNN, and mTAN, which respectively belong to RNN-based, ODE-based, and attention-based approaches. Additionally, we compare ContiFormer with baselines specifically designed for the MTPP task. These models include parametric method (HP [22]), RNN-based methods (RMTPP [16], NeuralHP [39]), and attention-based methods (SAHP [59], THP [64], NSTKA [38]).

The experimental results, as presented in Table 4, demonstrate the overall statistical superiority of ContiFormer to the compared baselines. Notably, the utilization of specific kernel functions, e.g., NSTKA and mTAN, results in subpar performance on certain datasets, highlighting their limitations in modeling complex data patterns in real-world scenarios. Furthermore, parametric methods like HP, perform poorly on most datasets, indicating their drawbacks in capturing real-life complexities.

Table 5: Results of regular time series forecasting. Input length for ILI is 36 and 96 for the others. Prediction lengths for ETTm2 and ETTh2 are $\{24, 48, 168, 336\}$, $\{96, 192, 336, 540\}$ for Exchange and Weather, and $\{24, 36, 48, 60\}$ for ILI. Results are averaged over the four prediction lengths. Bold/underlined values indicate the best/second-best. MSE stands for mean squared error.

| Model | ContiFormer | | TimesNet[52] | | DLinear[56] | | FEDformer[62] | | Autoformer[53] | | Transformer[51] | |
|---|---|---|---|---|---|---|---|---|---|---|---|---|
| Metric | MSE | MAE | MSE | MAE | MSE | MAE | MSE | MAE | MSE | MAE | MSE | MAE |
| Exchange | 0.339 | 0.399 | 0.320 | 0.396 | **0.277** | **0.377** | 0.392 | 0.446 | 0.440 | 0.477 | 1.325 | 0.949 |
| ETTm2 | 0.226 | 0.300 | **0.214** | **0.276** | 0.220 | 0.305 | 0.226 | 0.300 | 0.231 | 0.307 | 0.689 | 0.633 |
| ETTh2 | 0.356 | 0.395 | **0.321** | **0.364** | 0.358 | 0.396 | 0.351 | 0.393 | 0.352 | 0.394 | 2.081 | 1.179 |
| ILI | **2.632** | **1.042** | 2.874 | 1.066 | 4.188 | 1.493 | 4.467 | 1.547 | 4.046 | 1.446 | 5.130 | 1.507 |
| Weather | **0.257** | 0.290 | 0.259 | **0.285** | 0.260 | 0.311 | 0.315 | 0.360 | 0.318 | 0.359 | 0.672 | 0.579 |

## 4.4 Regular Time Series Forecasting

Lastly, we evaluate ContiFormer's performance in the context of regular time series modeling. To this end, we conducted an extensive experiment following the experimental settings from [53]. We employed the ETT, Exchange, Weather, and ILI datasets for time series forecasting tasks. We compare ContiFormer with the recent state-of-the-art model (TimesNet [52]), Transformer-based models (FEDformer[62], Autoformer[53]) and MLP-based model (DLinear [56]).

Table 5 summarizes the experimental results. Contiformer achieves the best performance on ILI dataset. Specifically, it gives **10%** MSE reduction ($2.874 \rightarrow 2.632$). Also, Contiformer outperforms Autoformer and FEDformer on Exchange, ETTm2, and Weather datasets. Therefore, it demonstrates that Contiformer performs competitively with the state-of-the-art model on regular time series forecasting. For a more comprehensive exploration of the results, please refer to Appendix C.4.

## 5 Efficiency Analysis

Employing continuous-time modeling in ContiFormer often results in substantial time and GPU memory overhead. Hence, we conduct a comparative analysis of ContiFormer's actual time costs against those of Transformer-based models (TST), ODE-based models (ODE-RNN, Neural CDE), and SSM-based models (S5).

Table 6 shows that both TST and S5 are effective models. Additionally, ContiFormer leverages a parallel mechanism for faster inference, resulting in faster inference speed as compared to ODE-RNN and Neural CDE. Furthermore, ContiFormer demonstrates competitive performance even with a larger step size, as detailed in Appendix D.3. Therefore, ContiFormer's robustness to step size allows it to achieve a superior balance between time cost and performance. Please refer to Appendix E for more details.

Table 6: Actual time cost during training for different models on RacketSports dataset (0% dropped) from UEA dataset. We adopt RK4 solver for ODE-RNN, Neural CDE, and ContiFormer. Step size refers to the amount of time increment for the RK4 solver. $\uparrow$ ($\downarrow$) indicates the higher (lower) the better.

| Model | Step Size | Time Cost ($\downarrow$) | Accuracy ($\uparrow$) |
|---|---|---|---|
| TST [57] | – | 0.14× | 0.826 |
| S5 [49] | – | 1× | 0.772 |
| ODE-RNN [43] | 0.1 | 2.47× | 0.827 |
| | 0.01 | 14.64× | 0.796 |
| Neural CDE [42] | 0.1 | 3.48× | 0.743 |
| | 0.01 | 25.03× | 0.748 |
| ContiFormer | 0.1 | 1.88× | **0.836** |
| | 0.01 | 5.87× | **0.836** |

## 6 Conclusion

We propose ContiFormer, a novel continuous-time Transformer for modeling dynamic systems of irregular time series data. Compared with the existing methods, ContiFormer is more flexible since it abandons explicit closed-form assumptions of the underlying dynamics, leading to better generalization in various circumstances. Besides, similar to Neural ODE, ContiFormer can produce a continuous and smooth outcome. Taking advantage of the self-attention mechanism, ContiFormer can also capture long-term dependency in irregular observation sequences, achieving superior performance on a variety of irregular time series modeling tasks.

# 7 Acknowledgement

This research is supported in part by the National Natural Science Foundation of China under grant 62172107.

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

# A  Implement Details and Complexity Analysis

## A.1  Background: Time Variable ODE

A numerical ODE solver integrates a single ODE system $\frac{d\mathbf{x}}{dt} = f(t, \mathbf{x})$, where $\mathbf{x} \in \mathbb{R}^d$ and $f : \mathbb{R}^{d+1} \to \mathbb{R}^d$. Given an initial state $\mathbf{x}_{t_0}$, the ODE solver produces $\mathbf{x}_{t_1}, \mathbf{x}_{t_2}, ..., \mathbf{x}_{t_N}$, which represents the latent state for each time point [9], i.e.,

$$\mathbf{x}_{t_1}, \mathbf{x}_{t_2}, \ldots, \mathbf{z}_{t_N} = \text{ODESolve}\,(\mathbf{x}_{t_0}, f, t_0, \ldots, t_N) \; . \tag{12}$$

To obtain the solution of Eq. (12), it requires the ODE solver to integrate over the interval $[t_0, t_N]$.

**Varied time intervals**  Now, we suppose that we need to solve $M$ dynamic system that has different time intervals. Let denote for the $i$-th system, the start point is $t_0^{(i)}$ and the endpoint is $t_1^{(i)}$ and the ODE system is defined by $\frac{d\mathbf{x}}{dt} = f(t, \mathbf{x})$. We can construct a dummy variable so that the integral interval always starts from $-1$ and ends at 1 and perform a change of variables to transform every system to use this dummy variable.

For simplifying purposes, we assume that a single system that integrates from $t_0$ to $t_1$ with $f(t, \mathbf{x})$ can control the differential equation and the solution of $\mathbf{x}(t)$. We introduce a dummy variable $s = \frac{2t - t_0 - t_1}{t_1 - t_0}$ and a solution $\tilde{\mathbf{x}}(s)$ in a fix interval $[-1, 1]$ satisfies

$$\tilde{\mathbf{x}}(s) = \mathbf{x}\big(\frac{s(t_1 - t_0) + t_0 + t_1}{2}\big) \; . \tag{13}$$

The differential equation that solve $\mathbf{x}$ follows that

$$
\begin{aligned}
\tilde{f}(s, \tilde{\mathbf{x}}(s)) = \frac{d\tilde{\mathbf{x}}(s)}{ds} &= \frac{d\mathbf{x}(t)}{dt}\bigg|_{t = \frac{s(t_1 - t_0) + t_0 + t_1}{2}} \frac{dt}{ds} \\
&= f(t, \mathbf{x}(t))|_{t = \frac{s(t_1 - t_0) + t_0 + t_1}{2}} \left(\frac{t_1 - t_0}{2}\right) \\
&= f\left(\frac{s(t_1 - t_0) + t_0 + t_1}{2}, \tilde{\mathbf{x}}(s)\right)\left(\frac{t_1 - t_0}{2}\right) \; .
\end{aligned}
\tag{14}
$$

Since $\tilde{\mathbf{x}}(-1) = \mathbf{x}(t_0)$ and $\tilde{\mathbf{x}}(1) = x_{t_1}$, thus we have

$$\mathbf{x}(t_1) = \text{ODESolve}(\mathbf{x}(t_0), f, t_0, t_1) = \text{ODESolve}(\tilde{\mathbf{x}}(-1), \tilde{f}, -1, 1) \; . \tag{15}$$

## A.2  Background: Approximating Definite Integral of a Function

Given a function $f$ that integrates over an interval $[-1, 1]$, a simple method to approximate it is to assume that the change is linear over time, i.e.,

$$\int_{-1}^{1} f(x)dx \approx \frac{f(-1) + f(1)}{2} \; . \tag{16}$$

Besides, in numerical analysis, Gauss–Legendre Quadrature provides a more accurate way to approximate an integral over an interval $[-1, 1]$, i.e.,

$$\int_{-1}^{1} f(x)\mathrm{d}x \approx \sum_{j} \omega_j f\left(\xi_j\right) \; , \tag{17}$$

where $\{\omega_j\}$ and $\{\xi_j\}$ are quadrature weights and nodes such that $-1 \leq \xi_1 < ... < \xi_k \leq 1$.

## A.3  Putting it all together

Given an irregular-sampled sequence, $\Gamma = [(x_1, t_1), (x_2, t_2), ..., (x_N, t_N)]$, where $x_i \in \mathbb{R}^d$ is the input feature of the $i$-th tokens happen at time $t_i$ and $N$ is the length of the sequence. We analyze the time and space complexity for an autoregressive task where the output of each observation is required for a particular classification or regression task. Specifically, for Eq. (3), $\boldsymbol{\alpha}_i(t_1), \boldsymbol{\alpha}_i(t_2), \ldots, \boldsymbol{\alpha}_i(t_N)$

is required to form the output of Transformer. Therefore, the key difficulty is to solve Eq. (3). By using the numerical approximation method, we have

$$\boldsymbol{\alpha}_i(t_j) = \frac{\int_{t_i}^{t_j} \mathbf{q}(\tau) \cdot \mathbf{k}_i(\tau)^\top d\tau}{t_j - t_i} = \frac{1}{2} \int_{-1}^{1} \tilde{\mathbf{q}}_{i,j}(\tau) \cdot \tilde{\mathbf{k}}_{i,j}(\tau)^\top d\tau \approx \frac{1}{2} \sum_{p=1}^{P} \gamma_p \tilde{\mathbf{q}}_{i,j}(\xi_p) \cdot \tilde{\mathbf{k}}_{i,j}(\xi_p)^\top , \quad (18)$$

where $\tilde{\mathbf{q}}_{i,j}$ and $\tilde{\mathbf{k}}_{i,j}$ follows that:

$$\begin{aligned}
\tilde{\mathbf{q}}_{i,j}(s) &= \mathbf{q}\left(\frac{s(t_i - t_j) + t_i + t_j}{2}\right) , \\
\tilde{\mathbf{k}}_{i,j}(s) &= \mathbf{q}_j\left(\frac{s(t_i - t_j) + t_i + t_j}{2}\right) .
\end{aligned} \quad (19)$$

We use $\hat{\mathbf{k}}_{i,j}$ to denote $\frac{d\tilde{\mathbf{k}}_{i,j}(s)}{ds}$. Thus, $\boldsymbol{\alpha}_i(t_j)$ can be solved by paralleling $N^2$ ODE solvers. Specifically, $\tilde{\mathbf{k}}_{i,j}(\xi_p)$ is given by

$$\tilde{\mathbf{k}}_{i,j}(\xi_1), \tilde{\mathbf{k}}_{i,j}(\xi_2), ..., \tilde{\mathbf{k}}_{i,j}(\xi_p) = \text{ODESolver}(\mathbf{k}_j(t_j), \hat{\mathbf{k}}_{i,j}, -1, \xi_1, \xi_2, ..., \xi_p) . \quad (20)$$

As for $\tilde{\mathbf{q}}_{i,j}(\xi_k)$, since the input sequence is irregularly sampled and discrete in time, we propose to use interpolation methods [30] (e.g., linear interpolation or cubic interpolation) method on $Q$ to construct a continuous-time query. Therefore $\tilde{\mathbf{q}}_{i,j}(\xi_k)$ can be easily calculated in $O(d)$ time complexity.

## B   Representation Power of ContiFormer

In the section, we mathematically prove that by curated designs of function hypothesis, many Transformer variants can be covered as a special case of ContiFormer. For simplification, we only discuss the self-attention mechanism, where $Q, K, V \in \mathbb{R}^{N \times d}$.

### B.1   Transformer Variants

We first show several Transformer-based methods for irregular time series. Most of these methods shown below calculate $N \times N$ attention matrix and multiply it with $V$[4].

**Vanilla Transformer**   Given query $Q \in \mathbb{R}^{N \times d}$ and $V \in \mathbb{R}^{N \times d}$, the attention matrix of a vanilla transformer [51] is calculated as

$$\text{Attn}(Q, K) = \begin{bmatrix} Q_1 \cdot K_1^\top & Q_1 \cdot K_2^\top & \cdots & Q_1 \cdot K_N^\top \\ Q_2 \cdot K_1^\top & Q_2 \cdot K_2^\top & \cdots & Q_2 \cdot K_N^\top \\ \vdots & \vdots & \ddots & \vdots \\ Q_N \cdot K_1^\top & Q_N \cdot K_2^\top & \cdots & Q_N \cdot K_N^\top \end{bmatrix} , \quad (21)$$

and the output of the self-attention mechanism is obtained by[5]

$$Z_i = \text{Softmax}(\text{Attn}(Q, K)_i)V . \quad (22)$$

**Kernelized Attention**   Kernelized attention mechanism [6] incorporate time information through defining specific kernel function $k(x, x')$ and replace the attention matrix with

$$\text{Attn}(Q, K) = \begin{bmatrix} Q_1 \cdot K_1^\top * k(t_1, t_1) & Q_1 \cdot K_2^\top * k(t_1, t_2) & \cdots & Q_1 \cdot K_N^\top * k(t_1, t_N) \\ Q_2 \cdot K_1^\top * k(t_2, t_1) & Q_2 \cdot K_2^\top * k(t_2, t_2) & \cdots & Q_2 \cdot K_N^\top * k(t_2, t_N) \\ \vdots & \vdots & \ddots & \vdots \\ Q_N \cdot K_1^\top * k(t_N, t_1) & Q_N \cdot K_2^\top * k(t_N, t_2) & \cdots & Q_N \cdot K_N^\top * k(t_N, t_N) \end{bmatrix} . \quad (23)$$

---

[4]Throughout the proof section, we define $\mathbf{v}_i(t_i) = \mathbf{v}_i(t_i) = V_i$, and focus on the proof of attention matrix.

[5]For simplification, we assume single-head attention is adopted throughout the proof.

[6]Not to be confused with methods that regard kernels as a form of approximation of the attention matrix.

Kernel functions are typically symmetric and hold that $\phi(x, x) = 1$ for any $x$. For instance, NSTKA [38] propose a learnable Generalized Spectral Mixture (GSM) kernel, i.e.,

$$k_{\text{GSM}}(x, x') = \sum_i \phi_i(x)\phi_i(x')\, k_{\text{Gibbs},i}(x, x') \cos\left(2\pi\left(\mu_i(x)x - \mu_i(x') x'\right)\right),$$

$$\text{where } k_{\text{Gibbs},i}(x, x') = \sqrt{\frac{2l_i(x)l_i(x')}{l_i(x)^2 + l_i(x')^2}} \exp\left(-\frac{(x - x')^2}{l_i(x)^2 + l_i(x')^2}\right). \tag{24}$$

where $\mu_i(x)$ denotes the mean, lengthscale as $l_i(x)$ and input-dependent weight as $\phi_i(x)$.

**Time Embedding Methods**  Time embedding methods learn a time representation, adding or concatenating it with the input embedding, i.e.,

$$\widehat{\text{Attn}}(Q, K) = \begin{bmatrix} (Q_1, \Phi(t_1)) \cdot (K_1, \Phi(t_1))^\top & \cdots & (Q_1, \Phi(t_1)) \cdot (K_N, \Phi(t_N))^\top \\ (Q_2, \Phi(t_2)) \cdot (K_1, \Phi(t_1))^\top & \cdots & (Q_2, \Phi(t_2)) \cdot (K_N, \Phi(t_N))^\top \\ \vdots & \ddots & \vdots \\ (Q_N, \Phi(t_N)) \cdot (K_1, \Phi(t_1))^\top & \cdots & (Q_N, \Phi(t_N)) \cdot (K_N, \Phi(t_N))^\top \end{bmatrix}, \tag{25}$$

where $(\cdot, \cdot)$ is the concatenation of two vectors, $\Phi(\cdot) : \mathbb{R} \to \mathbb{R}^d$ is a learnable time representation. For simplification, we assume that time embedding is concatenated with the input embedding and ignore linear transformation on the time embedding.[7]

For instance, Mercer [54] transforms the periodic kernel function into the high-dimensional space spanned by truncated Fourier basis under certain frequencies ($\{\omega_i\}$) and coefficients $\{c_i\}$, i.e.,

$$\Phi(t) \mapsto \Phi_d^{\mathcal{M}} = \left[\Phi_{\omega_1,d}^{\mathcal{M}}(t), \ldots, \Phi_{\omega_k,d}^{\mathcal{M}}(t)\right]^\top,$$

$$\text{where } \Phi_\omega^{\mathcal{M}}(t) = \left[\sqrt{c_1}, \ldots, \sqrt{c_{2j}} \cos\left(\frac{j\pi t}{\omega}\right), \sqrt{c_{2j+1}} \sin\left(\frac{j\pi t}{\omega}\right), \ldots\right]. \tag{26}$$

mTAN [48] generalizes the notion of a positional encoding used in Transformer-based models to continuous-time domain, i.e.,

$$\Phi(t) = \phi(t) \cdot \mathbf{w}, \tag{27}$$

where $\mathbf{w} \in \mathbb{R}^{d \times d}$ is a linear transformation [8] and

$$\phi(t)_i = \begin{cases} \omega_0 \cdot t + \phi_0, & \text{if } i = 0 \\ \sin(\omega_i \cdot t + \phi_i), & \text{if } 0 < i \leq d \end{cases}, \tag{28}$$

where $\omega$ and $\phi$ are learnable weights. Furthermore, mTAN assumes that $Q = K = \mathbf{0}$.

Besides, since $\text{Softmax}(\mathbf{x} + c) = \text{Softmax}(\mathbf{x})$, where $\mathbf{x}$ is a vector and $c$ is a constant. The following equation holds,

$$\text{Softmax}(\text{Attn}(Q, K)) = \text{Softmax}(\widehat{\text{Attn}}(Q, K)), \tag{29}$$

where the matrix form of $\text{Softmax}$ normalize each row of the matrix and

$\text{Attn}(Q, K) =$

$$\begin{bmatrix} Q_1 \cdot K_1^\top & \cdots & Q_1 \cdot K_N + \Phi(t_1) \cdot (\Phi(t_N) - \Phi(t_1))^\top \\ Q_2 \cdot K_1 + \Phi(t_2) \cdot (\Phi(t_2) - \Phi(t_1))^\top & \cdots & Q_2 \cdot K_N + \Phi(t_2) \cdot (\Phi(t_N) - \Phi(t_2))^\top \\ \vdots & \ddots & \vdots \\ Q_N \cdot K_1 + \Phi(t_N) \cdot (\Phi(t_1) - \Phi(t_N))^\top & \cdots & Q_N \cdot K_N^\top \end{bmatrix}. \tag{30}$$

---

[7]Using the concatenation operation, we have $(Q_i, \Phi(t_i)) \cdot (K_j, \Phi(t_j))^\top = Q_i \cdot K_j^\top + \Phi(t_i) \cdot \Phi(t_j)^\top$.

[8]In the original paper, the equation is defined as $\Phi(t_i, t_j) = \phi(t_i)\mathbf{w}\mathbf{v}^\top\phi(t_j)$. For simplicity purpose, we assume that $\mathbf{w} = \mathbf{v}$.

## B.2 Proof of Extension and Universal Approximation

**Lemma 1** (Existence of Continuously Differentiable Vector Function). *Given a continuously differentiable vector function $u : \mathbb{R} \to \mathbb{R}^d \in \mathcal{C}^1$, such that $\forall x \in \mathbb{R}, \|u(x)\|_2 < \infty, \|u(x)\|_0 = d$ and a continuously differentiable function $w : \mathbb{R} \to \mathbb{R} \in \mathcal{C}^1$, there always exists another continuously differentiable vector function $v : \mathbb{R} \to \mathbb{R}^d \in \mathcal{C}^1$ with a fixed point $p$ satisfying $u(p) \cdot v(p)^\top = w(p)$, such that $\forall x \in \mathbb{R}, u(x) \cdot v(x)^\top = w(x)$ holds.* [9]

*Proof.* Throughout the proof, we define $\xi := v(p)$.

Since $\forall x \in \mathbb{R}, 0 < \|u(x)\|_2 < \infty$, we define

$$v(x) = \frac{u(x)}{\|u(x)\|_2^2} \cdot w(x) + k(x) , \tag{31}$$

where $k(x) \in \mathcal{C}^1$ such that $\forall x \in \mathbb{R}, u(x) \cdot k(x)^\top = 0$ holds. Therefore, for $\forall x \in \mathbb{R}$, we have

$$u(x) \cdot v(x)^\top = u(x) \cdot \left( \frac{u(x)}{\|u(x)\|_2^2} \cdot w(x) + k(x) \right) = w(x) + u(x) \cdot k(x) = w(x) . \tag{32}$$

Thus, we only need to prove the existence of $k(x)$. Since $v(x)$ has a fixed point at $p$, therefore,

$$k(p) = \xi - \frac{u(p)}{\|u(p)\|_2^2} \cdot w(p) . \tag{33}$$

Let $z[i]$ denote the $i$-th element of a vector $z$, thus, Eq. (33) is equivalent to

$$k(p)[i] = \xi[i] - \frac{u(p)[i]}{\|u(p)\|_2^2} * w(p)[i] , \tag{34}$$

for $i \in [1, d]$. Since we have $u(p) \cdot v(p)^\top = w(p)$, thus,

$$k(p) \cdot u(p)^\top = \xi \cdot u(p)^\top - w(p) = 0 . \tag{35}$$

We denote $c_i := k(p)[i] * u(p)[i]$. Since Eq. (35) holds, therefore $\sum_{i=1}^d c_i = 0$. Finally, we can construct $k(x)$ as

$$k(x) = \begin{bmatrix} k(x)[1] \\ k(x)[2] \\ \vdots \\ k(x)[d] \end{bmatrix}^\top = \begin{bmatrix} c_1/u(x)[1] \\ c_2/u(x)[2] \\ \vdots \\ c_d/u(x)[d] \end{bmatrix}^\top . \tag{36}$$

Since $u, w \in \mathcal{C}^1$, therefore $k \in \mathcal{C}^1$, thus $v \in \mathcal{C}^1$. $\qquad\square$

**Lemma 2** (Existence of $\boldsymbol{k}_i(t)$). *Given a continuously differentiable vector function $\mathbf{q} : \mathbb{R} \to \mathbb{R}^d \in \mathcal{C}^1$ such that $\forall x \in \mathbb{R}, \|\mathbf{q}(x)\|_2 < \infty, \|\mathbf{q}(x)\|_0 = d$ and a continuously differentiable function $f : \mathbb{R} \to \mathbb{R} \in \mathcal{C}^2$ and a vector $K_i \in \mathbb{R}^d$ and satisfies $f(t_i) = \mathbf{q}(t_i) \cdot K_i^\top$, there always exist a continuously differentiable vector function $\mathbf{k}_i : \mathbb{R} \to \mathbb{R}^d$, such that the following statements hold:*

- $\mathbf{k}_i(t_i) = K_i$,

- $\forall t \in \mathbb{R}, t \neq t_i, \boldsymbol{\alpha}_i(t) := \frac{\int_{t_i}^t \mathbf{q}(\tau) \cdot \mathbf{k}_i(\tau)^\top d\tau}{t - t_i} = f(t)$.

*Proof.* Since $t \neq t_i$, we have

$$\begin{aligned}
&\boldsymbol{\alpha}_i(t) = f(t) \\
\iff &\boldsymbol{\alpha}_i(t)(t - t_i) = f(t)(t - t_i) \\
\overset{\text{①}}{\iff} &\frac{d\boldsymbol{\alpha}_i(t)(t - t_i)}{dt} = \frac{df(t)(t - t_i)}{dt} \\
\iff &\mathbf{q}(t) \cdot \mathbf{k}_i(t)^\top = \frac{df}{dt}(t - t_i) + f(t) ,
\end{aligned} \tag{37}$$

---

[9]Consider an open set $\mathcal{U}$ on the real line and a function $f$ defined on $\mathcal{U}$ with real values. Let $k$ be a nonnegative integer. The function $f$ is said to be of differentiability class $\mathcal{C}^k$ if the derivatives $f', f'', \dots, f^{(k)}$ exist and are continuous on $\mathcal{U}$.

where ① holds since we can define $\boldsymbol{\alpha}_i(t_i) = f(t_i)$. Besides, since

$$\lim_{|\epsilon| \to 0} \boldsymbol{\alpha}_i(t_i + \epsilon) = \lim_{|\epsilon| \to 0} \frac{\int_{t_i}^{t_i + \epsilon} \mathbf{q}(\tau) \cdot \mathbf{k}_i(\tau) d\tau}{\epsilon} = \mathbf{q}(t_i) \cdot \mathbf{k}_i(t_i)^\top = \mathbf{q}(t_i) \cdot K_i^\top \,, \tag{38}$$

such a definition does not break the continuity of $\boldsymbol{\alpha}$.

Since $f \in \mathcal{C}^2$, therefore we denote $g(t) := \frac{\mathrm{d}f}{\mathrm{d}t}(t - t_i) + f(t) \in \mathcal{C}^1$. Besides,

$$g(t_i) = f(t_i) = \mathbf{q}(t_i) \cdot K_i^\top = \mathbf{q}(t_i) \cdot \mathbf{k}_i(t_i)^\top \tag{39}$$

By using **Lemma 1** with $g$ corresponding to $w$, $\mathbf{q}$ to $u$, $\mathbf{k}_i$ to $v$ and $p = t_i$, we can prove the existence of $\mathbf{k}_i$ that satisfies Eq. (37). $\qquad\square$

**Theorem 2** (Universal Attention Approximation Theorem). *Given query (Q) and key (K), such that $\|Q_i\|_2 < \infty, \|Q_i\|_0 = d$ for $i \in [1, ..., N]$. For any attention matrix with formulation defined in Appendix B.1, i.e., $\mathrm{Attn}(Q, K)$, there always exists a family of continuously differentiable vector functions $k_1(\cdot), k_2(\cdot), \ldots, k_N(\cdot)$, such that the discrete formulation of $[\boldsymbol{\alpha}_1(\cdot), \boldsymbol{\alpha}_2(\cdot), ..., \boldsymbol{\alpha}_N(\cdot)]$ defined by*

$$\widetilde{\mathrm{Attn}}(Q, K) = \begin{bmatrix} \boldsymbol{\alpha}_1(t_1) & \boldsymbol{\alpha}_2(t_1) & \cdots & \boldsymbol{\alpha}_N(t_1) \\ \boldsymbol{\alpha}_1(t_2) & \boldsymbol{\alpha}_2(t_2) & \cdots & \boldsymbol{\alpha}_N(t_2) \\ \vdots & \vdots & \ddots & \vdots \\ \boldsymbol{\alpha}_1(t_N) & \boldsymbol{\alpha}_2(t_N) & \cdots & \boldsymbol{\alpha}_N(t_N) \end{bmatrix} \,, \tag{40}$$

*satisfies that $\widetilde{\mathrm{Attn}}(Q, K) = \mathrm{Attn}(Q, K)$, where*

$$\boldsymbol{\alpha}_i(t) = \frac{\int_{t_i}^t \mathbf{q}(\tau) \cdot \mathbf{k}_i(\tau)^\top d\tau}{t - t_i}, \text{if } t \neq t_i \,, \tag{41}$$

*and $\boldsymbol{\alpha}_i(t_i) = \mathbf{q}(t_i) \cdot \mathbf{k}_i(t_i)^\top$, $\mathbf{q}(t)$ satisfies $\mathbf{q}(t_i) = Q_i$.*

*Proof.* First, we show that the diagonal elements in Eq. (40), i.e.,

$$\boldsymbol{\alpha}_i(t_i) = \mathbf{q}(t_i) \cdot \mathbf{k}_i(t_i)^\top = Q_i \cdot K_i^\top \,, \tag{42}$$

holds for different forms of attention matrix defined in Appendix B.1.

Next, for a given $i \in [1, N]$, there exists a continuous cubic spline $h_i : \mathbb{R} \to \mathbb{R}$ satisfies $h_i(t_j) = \mathrm{Attn}(Q, K)_{j,i}$ for $\forall j \in [1, \ldots, N]$ and $h_i \in \mathcal{C}^2$. By using **Lemma 2** with $h_i$ corresponding to $f$, there exists a function $\mathbf{k}_i$ such that $\boldsymbol{\alpha}_i(t) = h_i(t)$. Therefore, $\boldsymbol{\alpha}_i(t_j) = h_i(t_j)$ for $j \in [1, ..., N]$, and we finish the proof. $\qquad\square$

# C  Experiment Details

## C.1  Modeling Continuous-Time Function

### C.1.1  Background: 2D Spiral

A spiral is a curve that starts from a central point and gradually moves away from it while turning around it. We consider two types of spirals that start from time $s$ and end at time $e$. The first type of spiral is defined as

$$\begin{aligned} x(t) &= (a + bt) \cdot cos(t) \,, \\ y(t) &= (a + bt) \cdot sin(t) \,, \end{aligned} \tag{43}$$

while the second type of spiral is defined as

$$\begin{aligned} x(t) &= (a + 50b/(e - t)) \cdot cos(e - t) \,, \\ y(t) &= (a + 50b/(e - t)) \cdot sin(e - t) \,, \end{aligned} \tag{44}$$

where $a$ and $b$ are parameters of the Archimedean spiral.

Table 7: More experimental results on spiral 2d.

| $\alpha$ | Metric | Transformer[51] | Neural ODE[9] | Latent ODE[9] (w/ RNN Enc) | Latent ODE[43] (w/ ODE Enc) | Neural CDE[42] | **ContiFormer** |
|---|---|---|---|---|---|---|---|
| | | | | Task = *Interpolation* ($\times 10^{-2}$) | | | |
| 0.02 | RMSE ($\downarrow$) | $1.37 \pm 0.15$ | $1.90 \pm 0.17$ | $2.09 \pm 0.22$ | $4.53 \pm 1.34$ | $4.80 \pm 0.50$ | $\mathbf{0.50 \pm 0.06}$ |
| | MAE ($\downarrow$) | $1.42 \pm 0.14$ | $1.90 \pm 0.17$ | $1.95 \pm 0.26$ | $4.25 \pm 1.48$ | $5.12 \pm 0.47$ | $\mathbf{0.53 \pm 0.07}$ |
| 0.2 | RMSE ($\downarrow$) | $2.30 \pm 0.51$ | $1.88 \pm 0.09$ | $3.34 \pm 0.10$ | $4.82 \pm 1.97$ | $4.60 \pm 1.08$ | $\mathbf{0.42 \pm 0.04}$ |
| | MAE ($\downarrow$) | $1.75 \pm 0.18$ | $1.58 \pm 0.12$ | $3.16 \pm 0.43$ | $4.73 \pm 1.21$ | $4.58 \pm 0.93$ | $\mathbf{0.42 \pm 0.01}$ |
| 2 | RMSE ($\downarrow$) | $3.75 \pm 0.50$ | $2.1 \pm 0.21$ | $3.04 \pm 0.48$ | $4.49 \pm 1.26$ | $2.92 \pm 0.32$ | $\mathbf{0.45 \pm 0.11}$ |
| | MAE ($\downarrow$) | $2.33 \pm 0.60$ | $2.49 \pm 0.30$ | $2.88 \pm 0.54$ | $4.12 \pm 1.00$ | $2.96 \pm 0.23$ | $\mathbf{0.41 \pm 0.07}$ |
| | | | | Task = *Extrapolation* ($\times 10^{-2}$) | | | |
| 0.02 | RMSE ($\downarrow$) | $1.37 \pm 0.11$ | $2.07 \pm 0.02$ | $1.59 \pm 0.05$ | $2.69 \pm 0.82$ | $2.24 \pm 0.03$ | $\mathbf{0.64 \pm 0.10}$ |
| | MAE ($\downarrow$) | $1.49 \pm 0.12$ | $2.02 \pm 0.07$ | $1.52 \pm 0.09$ | $2.83 \pm 0.88$ | $2.38 \pm 0.04$ | $\mathbf{0.65 \pm 0.08}$ |
| 0.2 | RMSE ($\downarrow$) | $2.96 \pm 1.46$ | $4.38 \pm 0.35$ | $4.78 \pm 0.15$ | $5.03 \pm 1.96$ | $3.81 \pm 1.02$ | $\mathbf{0.74 \pm 0.10}$ |
| | MAE ($\downarrow$) | $2.79 \pm 1.11$ | $3.35 \pm 0.43$ | $4.02 \pm 0.59$ | $4.72 \pm 1.96$ | $3.37 \pm 0.78$ | $\mathbf{0.69 \pm 0.09}$ |
| 2 | RMSE ($\downarrow$) | $5.36 \pm 0.97$ | $6.02 \pm 0.58$ | $5.70 \pm 0.57$ | $4.68 \pm 1.70$ | $3.44 \pm 1.06$ | $\mathbf{0.75 \pm 0.07}$ |
| | MAE ($\downarrow$) | $4.48 \pm 0.82$ | $4.84 \pm 0.43$ | $4.46 \pm 0.60$ | $4.29 \pm 1.34$ | $3.32 \pm 0.85$ | $\mathbf{0.69 \pm 0.07}$ |

#### C.1.2 Experiment Setting

We generate 300 spirals and 200/100 spirals are used for training/testing respectively. For each spiral, it randomly belongs to either a clockwise spiral or a counter-clockwise spiral, we sample $a \sim \mathcal{N}(0, \alpha), b \sim \mathcal{N}(0.3, \alpha)$ and we set $\alpha = 0.02$ for Table 2. Also, we add Gaussian noise from $\mathcal{N}(0, \beta)$ to all training samples and we set $\beta = 0.1$. We test different models for interpolation and extrapolation. Specifically, each spiral is sampled at 150 equally-spaced time points. To generate an irregular time series, we randomly sample 30 points from the first half of the trajectory. Thus, the first half of the trajectories are used for the interpolation task while the second half of the trajectories, which is totally unobserved during both training and testing, are used for the extrapolation task. All the models are trained for 5000 iterations with a batch size equal to 64.

#### C.1.3 Baseline Implementation

We compare ContiFormer with Neural ODE-based models and Transformer.

- *Neural ODE[9].* This model uses the hidden vectors from the last observation as input, and obtain both the interpolation and extrapolation result from a single ODE solver. We train the model by minimizing the mean square loss.

- *Latent ODE (w/ RNN Enc)[9].* We run an RNN encoder through the time series and infer the parameters for a posterior over $z_{t_0}$. Next, we sample $\mathbf{z}_{t_0} \sim q\left(\mathbf{z}_{t_0} \mid \{\mathbf{x}_{t_i}, t_i\}\right)$. Finally, we obtain both the interpolation and extrapolation result from a single ODE solver, i.e., $\text{ODESolve}(\mathbf{z}_{t_0}, f, \theta_f, t_0, ..., t_M)$, where $(t_0, t_1, ..., t_M)$ is the time of the trajectory. We train the model by maximizing the evidence lower bound (ELBO).

- *Latent ODE (w/ ODE Enc)[43].* We replace the RNN encoder in Latent ODE (w/ RNN Enc) with the ODE-RNN encoder following [43]. Again, we train the model by maximizing the evidence lower bound (ELBO).

- *Neural CDE[42].* We follow the official implementation to construct the CDE function. For the interpolation, the hidden vectors are directly obtained from the continuous-time path from Neural CDE. For the extrapolation, we input the hidden vectors from the last timestamp of the Neural CDE to a single ODE solver. We train the model by minimizing the mean square loss.

- *Transformer[51].* In order to obtain the prediction for both seen and unseen time points, we use a learnable mask for unseen inputs. Instead of using position embedding, the input is added with a temporal encoding before forwarding it to the transformer layer. We train the model by minimizing the mean square loss.

- *ContiFormer.* We use linear interpolation to deal with unseen observations to avoid the numerical error caused by constructing the continuous-time query. Similar to Transformer, temporal encoding is adopted and the model is trained by minimizing the mean square loss.

### C.1.4 Experimental Results with Different $\alpha$

In this subsection, we test Transformer, Neural ODE and ContiFormer with different $\alpha$ to show the robustness of these models. As shown in Table 7, our model can achieve the best result on different parameters.

### C.1.5 More Visualization Results

More visual results with $\alpha$=0.02 can be found in Figure 4.

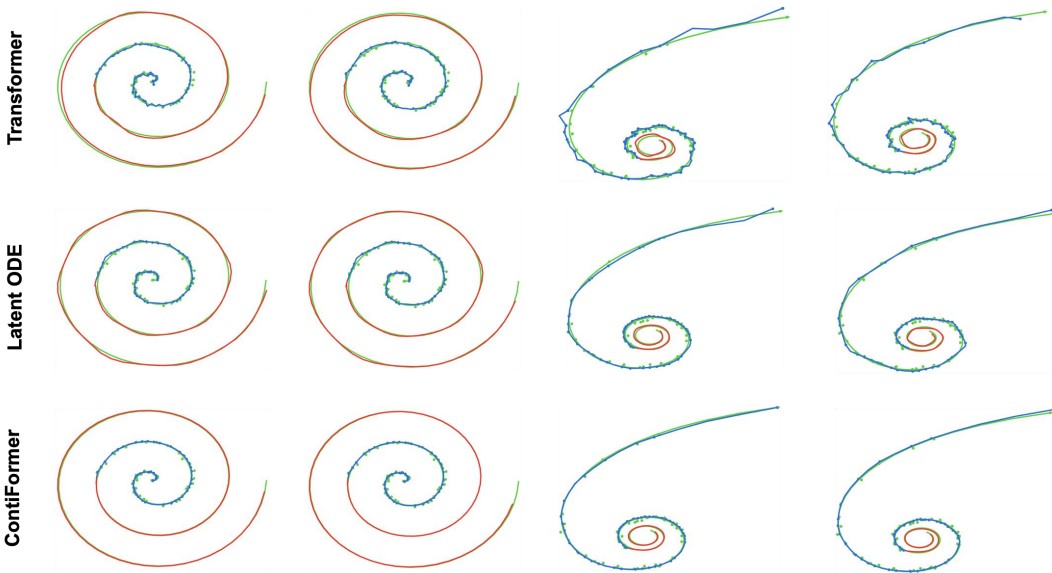

Figure 4: More visualization results with $\alpha = 0.02$. Here, Latent ODE refers to Latent ODE w/ RNN Encoder [9].

## C.2 Irregular Time Series Classification

### C.2.1 Training Details

For both our model and the baseline models, we adopted a fixed learning rate of $10^{-2}$ and a batch size of $64$. To ensure the reliability of our results, all experiments were conducted three times, and the reported values represent the mean. The training process for all models lasted for 1000 epochs. However, if the training accuracy failed to show any improvement for 100 consecutive epochs, the training was stopped to avoid unnecessary computation.

### C.2.2 Dataset Information

We select 20 datasets from UEA Time Series Classification Archieve with diverse characteristics in terms of the number, dimensionality and length of time series samples, as well as the number of classes. Detailed information is listed in Table 8.

### C.2.3 Full Experimental Results

We follow the setting of [30] to randomly drop either 30%, 50% or 70% observations. Experimental results are shown in Table 9, Table 10 and Table 11. We can find that ContiFormer outperforms RNN-based methods, ODE-based methods and Transformer-based methods. Also, we observe that ContiFormer outperforms all the baselines by a large margin when 70% observations are dropped, which confirms that ContiFormer is more suitable for irregular time series modeling.

Table 8: Dataset Information of UEA.

| Dataset | Ablation | # Train | # Test | Length | # Classes | # Variates |
|---|---|---|---|---|---|---|
| ArticularyWordRecognition | AWR | 275 | 300 | 144 | 25 | 9 |
| BasicMotions | BM | 40 | 40 | 100 | 4 | 6 |
| CharacterTrajectories | CT | 1422 | 1436 | 182 | 20 | 3 |
| DuckDuckGeese | DDG | 60 | 40 | 270 | 5 | 1345 |
| Epilepsy | EP | 137 | 138 | 207 | 4 | 3 |
| ERing | ER | 30 | 270 | 65 | 6 | 4 |
| FingerMovements | FM | 316 | 100 | 50 | 2 | 28 |
| HandMovementDirection | HMD | 160 | 74 | 400 | 4 | 10 |
| Handwriting | HW | 150 | 850 | 152 | 26 | 3 |
| Heartbeat | HB | 204 | 205 | 405 | 2 | 61 |
| JapaneseVowels | JV | 270 | 370 | 29 | 9 | 12 |
| Libras | LB | 180 | 180 | 45 | 15 | 2 |
| LSST | LSST | 2459 | 2466 | 36 | 14 | 6 |
| NATOPS | NATOPS | 180 | 180 | 51 | 6 | 24 |
| PEMS-SF | PEMS | 267 | 173 | 144 | 7 | 963 |
| PenDigits | PD | 7494 | 3498 | 8 | 10 | 2 |
| RacketSports | RS | 151 | 152 | 30 | 4 | 6 |
| SelfRegulationSCP1 | SR | 268 | 293 | 896 | 2 | 6 |
| SpokenArabicDigits | SA | 6599 | 2199 | 93 | 10 | 13 |
| UWaveGestureLibrary | UGL | 2238 | 2241 | 315 | 8 | 3 |

Table 9: Experimental result on UEA when 30% observations are dropped. (ODE. stands for ODE-RNN, N. CDE stands for Neural CDE, ACC. stands for Accuracy.)

| Dataset | GRU-D | GRU-Δt | ODE. | CADN | N. CDE | S5 | mTAN | TST | **ContiFormer** |
|---|---|---|---|---|---|---|---|---|---|
| AWR | 0.8533 | 0.8956 | 0.8789 | 0.8056 | 0.9089 | 0.9011 | 0.8067 | **0.9722** | 0.9633 |
| BM | 0.9333 | 0.9417 | 0.8250 | 0.9583 | **0.9917** | 0.9833 | **0.9917** | 0.9667 | 0.9750 |
| CT | 0.9325 | 0.8558 | 0.9415 | 0.9575 | 0.9276 | 0.9610 | 0.9529 | 0.9742 | **0.9833** |
| DDG | 0.5867 | 0.5933 | 0.5067 | 0.4600 | 0.3400 | 0.5667 | 0.6000 | 0.6533 | **0.7267** |
| EP | 0.7995 | 0.8043 | 0.8454 | 0.8123 | 0.7657 | 0.9074 | 0.9203 | **0.9589** | 0.9324 |
| ER | 0.7062 | 0.7148 | 0.6296 | 0.8623 | 0.7543 | 0.8346 | 0.7123 | 0.9160 | **0.9210** |
| FM | 0.6167 | 0.6167 | 0.5667 | **0.6233** | 0.5867 | 0.6033 | 0.5433 | 0.6133 | 0.6000 |
| HMD | 0.3559 | 0.4099 | 0.5450 | 0.4505 | 0.3333 | **0.6171** | 0.4414 | 0.5946 | 0.4550 |
| HW | 0.0722 | 0.1251 | 0.0675 | 0.0925 | 0.1831 | 0.1302 | 0.0937 | **0.2196** | 0.2000 |
| HB | 0.7577 | 0.7593 | 0.7805 | 0.7431 | 0.7333 | 0.7431 | **0.7789** | 0.7398 | 0.7561 |
| JV | 0.9703 | 0.9550 | 0.9432 | 0.9703 | 0.9225 | 0.9676 | 0.9595 | 0.9865 | **0.9919** |
| LB | 0.5037 | 0.4741 | 0.5222 | 0.5938 | 0.7481 | **0.9216** | 0.5426 | 0.7519 | 0.8778 |
| LSST | 0.5589 | 0.5615 | 0.5715 | 0.5574 | 0.3940 | **0.6389** | 0.5307 | 0.5520 | 0.6004 |
| NATOPS | 0.9204 | 0.9315 | 0.8704 | 0.8944 | 0.8148 | 0.8500 | 0.9019 | **0.9352** | 0.9222 |
| PEMS | 0.7938 | 0.7669 | 0.8208 | 0.7958 | 0.7572 | 0.9133 | 0.8189 | **0.8420** | 0.8401 |
| PD | 0.9309 | 0.9222 | 0.9433 | 0.9402 | 0.8422 | 0.8110 | 0.9029 | **0.9520** | 0.9517 |
| RS | 0.7434 | 0.7281 | 0.7654 | 0.7325 | 0.6798 | 0.7500 | 0.7325 | 0.8158 | **0.8487** |
| SR | 0.8942 | **0.9124** | 0.9101 | 0.8259 | 0.8783 | 0.8737 | 0.8646 | 0.9044 | 0.9101 |
| SA | 0.9663 | 0.9660 | 0.9830 | 0.9809 | 0.8998 | 0.9291 | 0.9321 | 0.9744 | **0.9836** |
| UGL | 0.6729 | 0.6625 | 0.6917 | 0.7479 | 0.8219 | 0.8042 | 0.7344 | **0.8552** | 0.8135 |
| **Acc. (↑)** | 0.7284 | 0.7298 | 0.7304 | 0.7402 | 0.7142 | 0.7854 | 0.7381 | 0.8089 | **0.8126** |
| **Rank (↓)** | 6 | 5.75 | 5.45 | 5.4 | 6.85 | 4.4 | 5.7 | 2.75 | **2.4** |

Table 10: Experimental result on UEA when 50% observations are dropped. (ODE. stands for ODE-RNN, N. CDE stands for Neural CDE, ACC. stands for Accuracy.)

| Dataset | GRU-D | GRU-Δt | ODE. | CADN | N. CDE | S5 | mTAN | TST | **ContiFormer** |
|---|---|---|---|---|---|---|---|---|---|
| AWR | 0.8811 | 0.8978 | 0.8989 | 0.8067 | 0.9133 | 0.9078 | 0.8067 | **0.9633** | **0.9633** |
| BM | 0.9417 | 0.8500 | 0.7667 | 0.9417 | 0.9583 | **0.9917** | 0.9583 | 0.9500 | 0.9583 |
| CT | 0.8872 | 0.9088 | 0.9506 | **0.9798** | 0.9285 | 0.9638 | 0.9431 | 0.9601 | **0.9798** |
| DDG | 0.5467 | 0.6000 | 0.4933 | 0.5467 | 0.3467 | 0.5667 | 0.6067 | 0.6533 | **0.6867** |
| EP | 0.7295 | 0.7899 | 0.7512 | 0.6728 | 0.7101 | 0.9012 | 0.9106 | **0.9589** | 0.9469 |
| ER | 0.7358 | 0.7432 | 0.5704 | 0.8527 | 0.7691 | 0.8284 | 0.6432 | **0.9012** | 0.8901 |
| FM | 0.5800 | 0.5967 | 0.5933 | 0.5933 | 0.6000 | 0.6033 | 0.5700 | **0.6067** | **0.6067** |
| HMD | 0.4099 | 0.4054 | 0.4685 | 0.4550 | 0.3243 | **0.5811** | 0.4099 | 0.5270 | 0.4595 |
| HW | 0.0769 | 0.0945 | 0.0588 | 0.1004 | 0.1855 | 0.1357 | 0.0737 | 0.1600 | **0.1757** |
| HB | 0.7642 | 0.7593 | **0.7707** | 0.7512 | 0.7333 | 0.7463 | 0.7789 | 0.7350 | 0.7610 |
| JV | 0.9604 | 0.9532 | 0.9126 | 0.9649 | 0.9180 | 0.9631 | 0.9595 | 0.9775 | **0.9856** |
| LB | 0.3796 | 0.4352 | 0.4537 | 0.5680 | 0.7315 | 0.6352 | 0.5259 | 0.6556 | **0.8537** |
| LSST | 0.5203 | 0.5292 | 0.5223 | 0.5222 | 0.3969 | **0.7204** | 0.5126 | 0.5151 | 0.5661 |
| NATOPS | 0.9185 | 0.9167 | 0.8444 | 0.8741 | 0.8019 | 0.8315 | 0.8778 | 0.8778 | **0.9222** |
| PEMS | 0.7823 | 0.7861 | 0.8170 | 0.7611 | 0.7534 | **0.9114** | 0.8015 | 0.7669 | 0.8401 |
| PD | 0.8245 | 0.7804 | 0.8563 | 0.8369 | 0.6389 | 0.6544 | 0.6898 | **0.8827** | 0.8799 |
| RS | 0.7412 | 0.7368 | 0.7127 | 0.7632 | 0.5526 | 0.7193 | 0.6601 | 0.7807 | **0.8355** |
| SR | 0.9135 | **0.9170** | 0.9044 | 0.8123 | 0.8760 | 0.8885 | 0.8658 | 0.8987 | 0.9067 |
| SA | 0.9583 | 0.9556 | 0.9759 | 0.9663 | 0.8907 | 0.9218 | 0.9254 | 0.9667 | **0.9767** |
| UGL | 0.6833 | 0.6583 | 0.6781 | 0.6531 | 0.8281 | 0.8042 | 0.7167 | **0.8490** | 0.8000 |
| **ACC.** (↑) | 0.7117 | 0.7157 | 0.7000 | 0.7211 | 0.6929 | 0.7638 | 0.7118 | 0.7793 | **0.7997** |
| **Rank** (↓) | 5.8 | 5.65 | 5.75 | 5.55 | 6.5 | 4.25 | 5.85 | 3.15 | **1.9** |

Table 11: Experimental result on UEA when 70% observations are dropped. (ODE. stands for ODE-RNN, N. CDE stands for Neural CDE, ACC. stands for Accuracy)

| Dataset | GRU-D | GRU-Δt | ODE. | CADN | N. CDE | S5 | mTAN | TST | **ContiFormer** |
|---|---|---|---|---|---|---|---|---|---|
| AWR | 0.8689 | 0.8822 | 0.8433 | 0.7911 | 0.9178 | 0.8956 | 0.7956 | 0.9089 | **0.9511** |
| BM | 0.7583 | 0.7917 | 0.7167 | **0.9750** | 0.9000 | 0.9167 | 0.9417 | 0.9167 | 0.9167 |
| CT | 0.8433 | 0.8496 | 0.9471 | 0.9582 | 0.9241 | 0.9547 | 0.9009 | 0.9313 | **0.9675** |
| DDG | 0.5200 | 0.6000 | 0.4800 | 0.5667 | 0.3533 | 0.5667 | 0.6067 | 0.5867 | **0.7267** |
| EP | 0.7464 | 0.7222 | 0.6643 | 0.6667 | 0.6667 | 0.8802 | 0.9058 | **0.9420** | 0.9275 |
| ER | 0.6741 | 0.6617 | 0.5383 | **0.8647** | 0.7630 | 0.8025 | 0.7407 | 0.8198 | 0.8580 |
| FM | 0.5467 | 0.5933 | 0.5733 | 0.6000 | 0.5767 | 0.6133 | 0.5700 | 0.5300 | **0.6233** |
| HMD | 0.3829 | 0.3784 | 0.4459 | 0.4324 | 0.3378 | **0.5991** | 0.3694 | 0.4324 | 0.4144 |
| HW | 0.0784 | 0.0831 | 0.0541 | 0.0957 | **0.1533** | 0.1078 | 0.0604 | 0.1255 | 0.1247 |
| HB | 0.7610 | 0.7528 | 0.7593 | 0.7610 | 0.7561 | 0.7512 | **0.7691** | 0.7593 | 0.7593 |
| JV | 0.9514 | 0.9414 | 0.9261 | 0.9685 | 0.8964 | 0.9559 | 0.9613 | 0.9595 | **0.9766** |
| LB | 0.3648 | 0.3722 | 0.3463 | 0.5412 | 0.6796 | 0.6278 | 0.4926 | 0.5500 | **0.7648** |
| LSST | 0.4803 | 0.4895 | 0.4505 | **0.5278** | 0.4093 | 0.6778 | 0.4992 | 0.4874 | 0.5149 |
| NATOPS | 0.8722 | 0.8981 | 0.7889 | 0.9074 | 0.7944 | 0.8259 | 0.8463 | 0.8315 | **0.9204** |
| PEMS | 0.7418 | 0.7611 | 0.7842 | 0.7726 | 0.7303 | **0.8902** | 0.7861 | 0.6840 | 0.8227 |
| PD | 0.7451 | 0.6670 | 0.7802 | 0.7596 | 0.5201 | 0.5197 | 0.6063 | **0.8078** | 0.8069 |
| RS | 0.6711 | 0.6776 | 0.6316 | 0.7018 | 0.5570 | 0.6382 | 0.6228 | 0.6930 | **0.7697** |
| SR | 0.9090 | **0.9170** | 0.9158 | 0.8515 | 0.8726 | 0.8874 | 0.8635 | 0.8612 | 0.8953 |
| SA | 0.9304 | 0.9295 | 0.9523 | 0.9513 | 0.8622 | 0.9024 | 0.8925 | 0.9447 | **0.9600** |
| UGL | 0.6031 | 0.6219 | 0.5896 | 0.6729 | **0.8344** | 0.7896 | 0.6792 | 0.8219 | 0.7969 |
| **ACC.** (↑) | 0.6725 | 0.6795 | 0.6594 | 0.7183 | 0.6753 | 0.7401 | 0.6955 | 0.7297 | **0.7749** |
| **Rank** (↓) | 6.15 | 5.7 | 6.4 | 3.9 | 6.3 | 4.45 | 5.4 | 4.2 | **2.1** |

## C.3 Predicting Irregular-Sampled Sequences

### C.3.1 Background: Marked Temporal Point Process (MTPP)

The marked Temporal Point Process (MTPP) has been widely studied in recent years. We define a sequence of events and types as $\Gamma = [(t_1, k_1), (t_2, k_2), \ldots, (t_N, k_N)]$, where $t_i$ is the time point when the $i$-th event occurs, with $t_1 < t_2 < \ldots < t_N$, $k_i \in \{1, 2, \ldots, \mathcal{K}\}$ is the type of the $i$-th event and $\mathcal{K}$ is the total number of event types. A history event is the event sequence up to but not including

time $t$, denoted by $\mathcal{H}_t = (t_j, k_j) : t_j < t$. The goal of the temporal point process is to model the conditional intensity function (CIF), denoted by $\lambda_k^*(t)$, which represents the likelihood of an event occurring at time $t$, given the historical events. Models are trained so that the log probability of an event sequence, i.e.,

$$\log p(\Gamma) = \sum_{j=1}^{N} \log \lambda_{k_j}^* (t_j) - \int_{t_1}^{t_N} \lambda^*(\tau) d\tau \;. \tag{45}$$

### C.3.2 Dataset Description

We use one synthetic dataset and five real-world datasets, namely Synthetic, Neonate [50], Traffic [32], MIMIC [16], BookOrder [16] and StackOverflow [33] to evaluate our model.

**Synthetic**  To evaluate the effectiveness of the proposed method, we create a synthetic dataset. Specifically, we consider 10 types of event where each type of event contain three properties, i.e., $(X_i, V_i, D_i)$. Hence, given an event $\mathbf{e}$ with type $j$ is represented as $\mathbf{e} = (e.x, e.v, e.d, e.k)$, where $e.x \sim \mathcal{N}(X_j, \mu_1)$, $e.v \sim \mathcal{N}(V_j, \mu_2)$, $e.d = D_j > 0$ and $e.k$ is the type of the event. We set $\mu_1 = \mu_2 = 0.01$ and $X, V, D \in \mathbb{R}^{10}$ is randomly generated and pre-defined.

To model both the correlation between events and the dynamic changes of events within the system. We first define the dynamic attention (weight) for each event, which is composed of inter-event dependency modeling and intra-event dynamic modeling. For inter-event dependency, it aims to calculate time-aware attention (weight) between each pair of events. Specifically, we use the following equation to define the weight between event $\mathbf{e}$ and $\hat{\mathbf{e}}$,

$$\omega(\mathbf{e}, \hat{\mathbf{e}}) = (e.t - \hat{e}.t) \cdot (e.x * \hat{e}.v - \hat{e}.x * e.v) \;. \tag{46}$$

Next, the define $\lambda_k^*(t)$ and $\lambda^*(t)$ in which we assume that only the last event attributes to the decision of the next event, following Markov decision process (MDP), i.e.

$$\lambda_k^*(t) = P(k|\hat{e}.k) \cdot \text{Softplus}\left( \sum_{\mathbf{e} \in \mathcal{H}_t} \omega(\mathbf{e}, \hat{\mathbf{e}}) \cdot \exp\left( \frac{-(t - e.t)}{e.d} \right) \right) \;,$$
$$\lambda^*(t) = \text{Softplus}\left( \sum_{\mathbf{e} \in \mathcal{H}_t} \omega(\mathbf{e}, \hat{\mathbf{e}}) \cdot \exp\left( \frac{-(t - e.t)}{e.d} \right) \right) \;. \tag{47}$$

where $\exp\left(-(t - e.t)/e.d\right)$ is the exponential decay effect for each event, $\hat{\mathbf{e}}$ is the last event as of time $t$. The existence of the exponential decay function ensures that all the events will eventually disappear in the system as time goes to infinity so as to stabilize the conditional intensify function. $P(k|\hat{k})$ is a pre-defined matrix that means the probability of the next event to with type $k$ given that the type of the previous event is $\hat{\mathbf{e}}$.

Next, we predict the occurrence time of the next event with

$$\widehat{t}_{j+1} = \int_{t_j}^{\infty} t \cdot p\left(t \mid \mathcal{H}_t\right) dt \;,$$
$$\text{where} \quad p\left(t \mid \mathcal{H}_t\right) = \lambda^*(t) \exp\left( -\int_{t_j}^{t} \lambda^*(t) dt \right) \;. \tag{48}$$

To predict the type of the next event, we sample the type of the next event from $P(\cdot|\hat{e}.k)$, where $\hat{e}.k$ is the type of the previous event.

Finally, we use 500 different random seeds to generate a total of 500 event sequences. Also, we use 4-fold cross-validation for evaluation. The generation process is shown in Algorithm 1.

**Neonate.**  Neonate dataset is a clinical dataset from 79 patients, among which 48 patients have incurred at least two seizures during the observation. The dataset contains only one type of event, i.e., the start point of a seizure (in minutes). The task is to predict when the next seizure will happen. Note that we only clear the start point of the seizure. Thus, we only have one type of event.

**Algorithm 1** Generate Synthetic Dataset with Time-Aware Dynamic Kernel

---

**Input** $seed \in [0, 500)$, $tmax = 20$, $X = [0.78, 0.11, -1.0, -0.56, -0.78, 1.0, 0.56, -0.33, 0.3,$
$-0.11]$, $V = [-0.17, 0.5, -0.28, 0.28, 0.17, 0.39, -0.05, 0.05, -0.39, -0.5]$, $D = [0.5, 0., 0.28,$
$0.17, 0.22, 0.44, 0.33, 0.11, 0.05, 0.39]$
**Output** A generated event sequence $E$.
Initialize the random state of Numpy using *seed* and defined the probabilistic matrix $P$.
Initialize the event sequence $E$ to be an empty array, and set current time $T = 0$.
Random generate the type of the first event $e.k$ and sample $e.x, e.v$ and $e.d$.
Add the event to the event sequence.
**while** $t \leq$ *tmax* **do**
    Calculate the conditional intensity function using Eq. (47).
    Predict the time of the next event $e.t$ using Eq. (48).
    Assume the type of previous event is $\hat{k}$, sample the type of the next event $e.k$ from $P(\cdot|\hat{k})$.
    Sample sample $e.x, e.v$ and $e.d$ given $e.k$.
    Add the generated event to the event sequence $E$.
    Update $T = e.t$.
**end while**
**return** $E$

---

**Traffic.** Traffic dataset is a benchmark dataset for time-series forecasting collected by the Caltrans Performance Measurement System (PeMS). This dataset contains the traffic state monitoring of 861 sensors in a two-year period. The data is collected every 1 hour. We extract time points with fluctuations higher than 50% as events, whose types are up and down according to the direction. We consider two events that are within 6 hours as duplicated events and remove the latter one from the dataset. Besides, for each sensor, it contains a long sequence, we partition these event sequences by month, i.e., for each event sequence, it contains the event of a sensor in one month.

**MIMIC.** MIMIC dataset is a clinical dataset that collects patients' visits to a hospital's ICU in a seven-year period. We treat the visits of each patient as a separate sequence.

**BookOrder.** BookOrder dataset is a financial dataset that contains transaction records of stock in one day. We record the time (in milliseconds) and the action that was taken in each transaction. The dataset is a single long sequence with only two types of events: "buy" and "sell". We clip the event sequences so that the maximum length is 500 in both the train and test sets. Furthermore, the mean $\mu$ and standard derivative $\sigma$ of the time difference between two events in the dataset is 1.32 and 20.24 respectively. Therefore, we clip the maximum time difference to 70.

**StackOverflow.** StackOverflow dataset is a question-answering website. The website rewards users with badges to promote engagement in the community, and the same badge can be rewarded multiple times to the same user. We collect data in a two-year period, and we treat each user's reward history as a sequence.

For MIMIC, Bookorder, and StackOverflow datasets, we use the dataset from the Google Drive[10]. Statistic information can be found in Table 12. We use the 4-fold cross-validation scheme for Synthetic, Neonate, and Traffic datasets following [17], and the 5-fold cross-validation scheme for the other three datasets following [39, 64].

### C.3.3 Implementation Details

**Input Embedding** Given the history event $\mathcal{H}_t = \{(t_j, k_j) : t_j < t\}$ with a total of $\mathcal{K}$ events, the input of Transformer-based model and RNN-based model is obtained through event embedding and temporal encoding, i.e., for the $i$-th event $x_i = \text{emb}(k) + \text{enc}(t)$, where $\text{emb}(k)$ is the $k$-th column of the embedding table $\mathcal{E} \in \mathbb{R}^{\mathcal{K} \times d}$ and temporal encoding is defined by a combination of sine and cosine function [64].

---

[10]https://drive.google.com/drive/folders/0BwqmV0EcoUc8UklIR1BKV25YR1U?resourcekey=
0-OrlU87jyc1m-dVMmY5aC4w

Table 12: Dataset statistics: name of the dataset, number of event types, number of events in the dataset, average length per sequence, and number of sequences in training/test sets.

| Dataset | K | # Events | Avg. Length | # Train Seqs | # Test Seqs |
|---|---|---|---|---|---|
| Synthetic | 10 | 8, 618 | 17 | 375 | 125 |
| Neonate | 1 | 534 | 11 | 38 | 10 |
| Traffic | 2 | 154, 747 | 60 | 1, 938 | 647 |
| MIMIC | 75 | 2, 419 | 4 | 527 | 65 |
| BookOrder | 2 | 414, 800 | 2, 074 | 90 | 100 |
| StackOverflow | 22 | 480, 413 | 72 | 4, 777 | 1, 326 |

**Conditional Intensity Function**   Given the output from the model, i.e. $\mathbf{h}(t_1), \mathbf{h}(t_2), .., \mathbf{h}(t_N)$. The dynamics of the temporal point process are described by a continuous conditional intensity function. To reduce the memory cost and accelerate the training process. Instead of using the continuous-time output from ContiFormer, we only use ContiFormer to generate a hidden representation for discrete time points, i.e., $t_1, t_2, ..., t_j, ..., t_N$. Following [64], the conditional intensity function is defined by

$$\lambda(t \mid \mathcal{H}_t) = \sum_{k=1}^{K} \lambda_k(t \mid \mathcal{H}_t) , \tag{49}$$

where each of the type-specific conditional intensity function is defined by

$$\lambda_k(t \mid \mathcal{H}_t) = f_k\left(\alpha_k \frac{t - t_j}{t_j} + \mathbf{w}_k^\top \mathbf{h}(t_j) + b_k\right) , \tag{50}$$

where on interval $t \in [t_j, t_{j+1})$ and $f_k(x) = \beta_k \log(1 + \exp(x/\beta_k))$ is the softplus function with parameter $\beta_k$. The use of such a closed-form intensity function allows for $O(1)$ interpolation.

**Training Loss**   We train our model and all the baseline models by jointing maximizing the log-likelihood of the sequence and minimizing the prediction losses.

Specifically, for a sequence $\mathcal{S}$ over an interval $[t_i, t_N]$, given the conditional intensity function $\lambda(t \mid \mathcal{H}_t)$, the log-likelihood is given by

$$\ell_{LL}(\mathcal{S}) = \underbrace{\sum_{j=1}^{N} \log \lambda(t_j \mid \mathcal{H}_j)}_{\text{event log-likelihood}} - \underbrace{\int_{t_1}^{t_N} \lambda(t \mid \mathcal{H}_t) \, dt}_{\text{non-event log-likelihood}} . \tag{51}$$

One challenge to compute the log-likelihood is to solve the integral, i.e., $\Lambda = \int_{t_1}^{t_N} \lambda(t \mid \mathcal{H}_t) \, dt$. Since *softplus* function has no closed-form computation, we apply the Monte Carlo integration method, i.e.,

$$\widehat{\Lambda}_{\mathrm{MC}} = \sum_{j=2}^{P} (t_j - t_{j-1}) \left(\frac{1}{P} \sum_{i=1}^{P} \lambda(u_i)\right) , \tag{52}$$

where $u_i \sim \mathrm{Unif}(t_{j-1}, t_j)$ for $i \in [1, ..., P]$ is sampled from a uniform distribution over an interval $[t_{j-1}, t_j]$.

Besides the log-likelihood loss, we also use the cross-entropy loss and regression loss for predicting the type and time of the next events.

Specifically, given the conditional intensity function, the next time stamp prediction is given by

$$p(t \mid \mathcal{H}_t) = \lambda(t \mid \mathcal{H}_t) \exp\left(-\int_{t_j}^{t} \lambda(\tau \mid \mathcal{H}_\tau) \, d\tau\right) ,$$
$$\widehat{t}_{j+1} = \int_{t_j}^{\infty} t \cdot p(t \mid \mathcal{H}_t) \, dt . \tag{53}$$

Thus, the regression loss and the cross-entropy loss are given by

$$\ell_{reg}(\mathcal{S}) = \sum_{i=2}^{N} (\widehat{t}_i - t_i)^2 \, , \ell_{pred}(\mathcal{S}) = \sum_{i=2}^{N} -\log\left(\frac{\lambda_{k_i}\left(\widehat{t}_i\right)}{\lambda\left(\widehat{t}_i\right)}\right) \, . \tag{54}$$

Overall, the modeling is trained in a multi-task manner, i.e.,

$$\ell(\mathcal{S}) = \ell_{LL} + \alpha_1 \ell_{reg}(\mathcal{S}) + \alpha_2 \ell_{pred}(\mathcal{S}) \, , \tag{55}$$

where $\alpha_1$ and $\alpha_2$ are the pre-defined weight. By default, we set $\alpha_1 = 0.01$ and $\alpha_2 = 1$.

### C.3.4 Significance test on Synthetic and BookOrder datasets

To demonstrate the statistical superiority of ContiFormer, we conducted a t-test on Synthetic and BookOrder datasets using 10 random seeds. The t-test results are listed in Table 13.

Table 13: Significance test on Synthetic and BookOrder dataset with 10 repeats. The P-value represents the significant value between the ContiFormer and the baseline model. Bolder values represent P-value $< 10^{-6}$ in significance test.

| Dataset | Synthetic | | | | | | BookOrder | | | | | |
|---|---|---|---|---|---|---|---|---|---|---|---|---|
| Metric | LL (↑) | | Accuracy (↑) | | RMSE (↓) | | LL (↑) | | Accuracy (↑) | | RMSE (↓) | |
| Statistic | Mean | P-value | Mean | P-value | Mean | P-value | Mean | P-value | Mean | P-value | Mean | P-value |
| GRU-dt | -0.8611 | 5.38E-08 | 0.8420 | 0.003 | 0.2556 | 3.49E-09 | -1.0137 | 1.07E-31 | 0.6274 | 3.11E-08 | 3.6680 | 1.32E-05 |
| NSTKA | -1.0088 | 1.27E-21 | 0.8419 | 0.0003 | 0.3863 | 1.74E-12 | -1.0916 | 2.95E-27 | 0.6223 | 5.48E-15 | 3.7175 | 1.88E-08 |
| ODE-RNN | -0.8432 | 2.39E-11 | 0.8420 | 0.098 | 0.2401 | 3.77E-08 | -0.9458 | 1.44E-26 | 0.6278 | 1.17E-07 | 3.5898 | 1.94E-03 |
| RMTPP | -1.0132 | 3.57E-19 | 0.8420 | 0.0143 | 0.3711 | 6.70E-18 | -0.9551 | 9.70E-26 | 0.6245 | 1.45E-13 | 3.6475 | 9.76E-04 |
| SAHP | -0.6247 | 2.21E-05 | 0.8420 | 0.1778 | 0.5218 | 2.37E-10 | -0.3053 | 1.83E-07 | 0.6227 | 1.45E-13 | 3.6810 | 9.55E-07 |
| THP | -0.6035 | 6.28E-06 | 0.8420 | 0.1354 | 0.2100 | 4.85E-06 | -1.0788 | 1.55E-17 | 0.6251 | 1.96E-06 | 3.6907 | 8.02E-07 |
| ContiFormer | **-0.5445** | N/A | 0.8421 | N/A | **0.1943** | N/A | **-0.2745** | N/A | **0.6295** | N/A | 3.6171 | N/A |

### C.4 Regular Time Series Forecasting

Time series forecasting has garnered considerable attention in recent years, as evidenced by various studies [35, 53, 61]. In this study, we assess ContiFormer against several models tailored for regular time series forecasting.

**Datasets**   We evaluate ContiFormer on five experimental datasets. i) ETT dataset [61] contains the data collected from electricity transformers, including load and oil temperature that are recorded every 15 minutes between July 2016 and July 2018. Here, we use ETTm2 and ETTh2 datasets. ii) Exchange dataset [53] records the daily exchange rates of eight different countries ranging from 1990 to 2016. iii) Weather dataset is recorded every 10 minutes for 2020 whole year, which contains 21 meteorological indicators, such as air temperature, humidity, etc. v) ILI dataset includes the weekly recorded influenza-like illness (ILI) patients data from Centers for Disease Control and Prevention of the United States between 2002 and 2021.

**Baselines**   We compare ContiFormer with several transformer-based models for regular time series forecasting under the multivariate setting. These transformer-based models includes: FEDformer[62], Autoformer[53], Informer[61] and Reformer[31]. Also, we chose one recent state-of-the-art model. i.e. TimesNet[52] for comparison.

**Results of Time Series Forecasting**   The experimental results are depicted in Table 14. ContiFormer achieves the best performance on ILI dataset. Specifically, it gives **10%** MSE reduction ($2.874 \rightarrow 2.632$). Also, ContiFormer outperforms Autoformer and FEDformer on Exchange, ETTm2, and Weather datasets. Therefore, it demonstrates that ContiFormer performs competitively with the state-of-the-art model.

### C.5 Pendulum Regression Task

The objective of pendulum regression is to assess the models' capability in accommodating observations obtained at irregular intervals [45, 49]. The input sequence comprises $50$ images, each

Table 14: Results of regular time series forecasting. We set the input length as 36 for ILI and 96 for the others. Avg. stands for Average. Bold values indicate the best performance and underlined values indicate the second-best performance (lower the better).

| Models | | ContiFormer | | TimesNet[52] | | DLinear[56] | | FEDformer[62] | | Autoformer[53] | | Transformer[51] | | Informer[61] | | Reformer[31] | |
|---|---|---|---|---|---|---|---|---|---|---|---|---|---|---|---|---|---|
| Metrics | | MSE | MAE | MSE | MAE | MSE | MAE | MSE | MAE | MSE | MAE | MSE | MAE | MSE | MAE | MSE | MAE |
| Exchange | 96 | 0.105 | 0.232 | 0.107 | 0.238 | **0.088** | **0.217** | 0.153 | 0.284 | 0.153 | 0.283 | 0.891 | 0.743 | 1.348 | 0.966 | 1.208 | 0.961 |
| | 192 | 0.215 | 0.339 | 0.201 | 0.327 | **0.175** | **0.313** | 0.264 | 0.375 | 0.327 | 0.413 | 1.338 | 0.955 | 1.296 | 0.941 | 1.329 | 1.001 |
| | 336 | 0.336 | 0.427 | 0.354 | 0.435 | **0.310** | **0.421** | 0.437 | 0.485 | 0.555 | 0.560 | 1.553 | 1.056 | 1.641 | 1.072 | 3.023 | 1.427 |
| | 540 | 0.698 | 0.598 | 0.614 | 0.582 | **0.534** | **0.555** | 0.710 | 0.639 | 0.724 | 0.648 | 1.515 | 1.042 | 2.430 | 1.284 | 1.801 | 1.159 |
| | **Avg.** | 0.339 | 0.399 | 0.320 | 0.396 | **0.277** | **0.377** | 0.392 | 0.446 | 0.440 | 0.477 | 1.325 | 0.949 | 1.679 | 1.066 | 1.841 | 1.137 |
| ETTm2 | 24 | 0.126 | 0.226 | **0.104** | **0.200** | 0.109 | 0.213 | 0.149 | 0.253 | 0.157 | 0.262 | 0.282 | 0.397 | 0.391 | 0.475 | 0.405 | 0.499 |
| | 48 | 0.175 | 0.276 | **0.143** | **0.238** | 0.145 | 0.249 | 0.172 | 0.268 | 0.177 | 0.276 | 0.501 | 0.535 | 0.420 | 0.484 | 0.541 | 0.569 |
| | 168 | 0.272 | 0.336 | **0.255** | 0.308 | 0.262 | 0.345 | **0.255** | 0.318 | 0.262 | 0.324 | 0.824 | 0.730 | 1.068 | 0.869 | 1.149 | 0.846 |
| | 336 | 0.331 | 0.361 | 0.352 | **0.359** | 0.366 | 0.414 | **0.326** | 0.361 | 0.328 | 0.365 | 1.149 | 0.871 | 1.379 | 0.983 | 2.258 | 1.153 |
| | **Avg.** | 0.226 | 0.300 | 0.214 | 0.276 | 0.220 | 0.305 | 0.226 | 0.300 | 0.231 | 0.307 | 0.689 | 0.633 | 0.815 | 0.703 | 1.088 | 0.767 |
| ETTh2 | 24 | 0.237 | 0.327 | **0.196** | **0.281** | 0.179 | 0.277 | 0.262 | 0.340 | 0.266 | 0.343 | 0.946 | 0.800 | 1.408 | 1.028 | 0.762 | 0.703 |
| | 48 | 0.287 | 0.355 | **0.244** | **0.317** | 0.247 | 0.329 | 0.291 | 0.354 | 0.294 | 0.358 | 1.009 | 0.826 | 1.642 | 1.100 | 1.101 | 0.856 |
| | 168 | 0.417 | 0.426 | **0.397** | **0.409** | 0.430 | 0.448 | 0.403 | 0.420 | 0.403 | 0.419 | 3.014 | 1.483 | 3.261 | 1.511 | 2.076 | 1.144 |
| | 336 | 0.484 | 0.471 | **0.446** | **0.449** | 0.577 | 0.532 | 0.448 | 0.458 | 0.445 | 0.456 | 3.357 | 1.606 | 3.155 | 1.481 | 2.633 | 1.297 |
| | **Avg.** | 0.356 | 0.395 | **0.321** | **0.364** | 0.358 | 0.396 | 0.351 | 0.393 | 0.352 | 0.394 | 2.081 | 1.179 | 2.366 | 1.280 | 1.643 | 1.000 |
| ILI | 24 | **2.391** | **1.004** | 2.925 | 1.061 | 4.412 | 1.576 | 4.979 | 1.668 | 4.334 | 1.527 | 4.789 | 1.431 | 5.718 | 1.623 | 4.678 | 1.455 |
| | 36 | **2.673** | **1.006** | 3.145 | 1.127 | 4.314 | 1.529 | 4.812 | 1.628 | 4.084 | 1.438 | 4.995 | 1.474 | 5.353 | 1.564 | 4.801 | 1.490 |
| | 48 | **2.536** | 1.039 | 2.716 | 1.027 | 3.910 | 1.424 | 4.112 | 1.465 | 3.870 | 1.408 | 5.283 | 1.545 | 5.120 | 1.537 | 4.946 | 1.510 |
| | 60 | 2.930 | 1.117 | **2.711** | **1.049** | 4.114 | 1.446 | 3.965 | 1.428 | 3.895 | 1.411 | 5.454 | 1.578 | 5.556 | 1.613 | 5.134 | 1.545 |
| | **Avg.** | **2.632** | **1.042** | 2.874 | 1.066 | 4.188 | 1.493 | 4.467 | 1.547 | 4.046 | 1.446 | 5.130 | 1.507 | 5.437 | 1.584 | 4.890 | 1.500 |
| Weather | 96 | **0.170** | **0.218** | 0.171 | 0.221 | 0.194 | 0.250 | 0.254 | 0.323 | 0.253 | 0.317 | 0.579 | 0.539 | 0.599 | 0.541 | 0.940 | 0.675 |
| | 192 | **0.229** | **0.271** | 0.230 | 0.269 | 0.237 | 0.297 | 0.287 | 0.344 | 0.293 | 0.347 | 0.738 | 0.613 | 0.459 | 0.453 | 1.061 | 0.735 |
| | 336 | 0.288 | 0.317 | 0.289 | 0.302 | 0.282 | 0.334 | 0.336 | 0.374 | 0.346 | 0.380 | 0.464 | 0.466 | 0.575 | 0.495 | 1.201 | 0.802 |
| | 540 | 0.341 | 0.355 | 0.345 | 0.348 | 0.325 | 0.365 | 0.384 | 0.400 | 0.379 | 0.394 | 0.909 | 0.700 | 0.862 | 0.625 | 0.958 | 0.702 |
| | **Avg.** | **0.257** | **0.290** | 0.259 | 0.285 | 0.260 | 0.311 | 0.315 | 0.360 | 0.318 | 0.359 | 0.672 | 0.579 | 0.624 | 0.529 | 1.040 | 0.729 |

measuring 24 by 24 pixels, and is afflicted by correlated noise. These images are sampled at irregular intervals from a continuous trajectory spanning a duration of 100 units, while the targets correspond to the sine and cosine values of the pendulum's angle. It's important to note that the pendulum follows a nonlinear dynamical system, with the velocity remaining unobserved. Table 15 summarizes the results of this experiment. ContiFormer outperforms CRU and ODE-RNN.

Table 15: Regression MSE $\times 10^{-3}$ (mean $\pm$ std) on pendulum regression task.

| Model | Regression MSE ($\times 10^{-3}$) |
|---|---|
| mTAN [48] | 65.64 (4.05) |
| ODE-RNN [43] | 7.26 (0.41) |
| CRU [45] | 4.63 (1.07) |
| S5 [49] | **3.41 (0.27)** |
| ContiFormer | 4.21 (0.24) |

# D    Ablation Study and Parameter Study

## D.1    Effects of Different Attention Mechanism

We study the impact of different attention mechanisms to show the significance of both continuous-evolving attention scores and continuous value functions. We claim that both of these two continuous designs attribute to the modeling of complex dynamic systems. We conduct the experiment on 6 datasets for event prediction task. As shown in Table. 16, ContiFormer outperforms both of the two variants, which shows that jointly modeling the dynamic relationship between input observations and meanwhile capturing the dynamic change of the observations is critical for modeling complex event sequences in reality.

## D.2    Effects of Different Numeric Approximation Methods

As discussed in Appendix A.2, linear interpolation and Gauss-Legendre Quadrature are two methods for approximating an integral in a close interval $[-1, 1]$. We study the impact of these two methods and set $P$, i.e., the number of intermediate steps for integral approximation, in the Gauss-Legendre quadrature from $\{2, 3, 4, 5\}$. We conducted the experiment on 6 datasets for event prediction task.

Table 16: Effects of different attention mechanism design on event prediction task. Attn. stands for Attention. ✔ refers to a continuous version of attention/value, while ✗ refers to a static version. It is important to note that when both attention and value are static, the model is equivalent to Transformer. Conversely, when both are continuous, the model is referred to as ContiFormer.

| Dataset | | Synthetic | | | Neonate | | Traffic | | |
|---|---|---|---|---|---|---|---|---|---|
| Attn. | Value | LL (↑) | ACC (↑) | RMSE (↓) | LL (↑) | RMSE (↓) | LL (↑) | ACC (↑) | RMSE (↓) |
| ✗ | ✗ | -0.589 | 0.841 | 0.205 | -2.702 | 9.471 | 0.569 | 0.818 | 0.332 |
| ✗ | ✔ | -0.576 | **0.842** | 0.202 | -2.634 | 9.249 | 0.598 | 0.819 | **0.327** |
| ✔ | ✗ | -0.563 | 0.842 | 0.200 | -2.590 | 9.267 | 0.544 | 0.814 | 0.329 |
| ✔ | ✔ | **-0.535** | **0.842** | **0.192** | **-2.550** | **9.233** | **0.635** | **0.822** | 0.328 |

| Dataset | | MIMIC | | | BookOrder | | | StackOverflow | | |
|---|---|---|---|---|---|---|---|---|---|---|
| Attn. | Value | LL (↑) | ACC (↑) | RMSE (↓) | LL (↑) | ACC (↑) | RMSE (↓) | LL (↑) | ACC (↑) | RMSE (↓) |
| ✗ | ✗ | -1.137 | 0.834 | 0.843 | -0.302 | 0.622 | 3.688 | -2.354 | 0.468 | 0.951 |
| ✗ | ✔ | -1.147 | 0.832 | **0.837** | -0.312 | 0.627 | 3.690 | -2.354 | 0.468 | 0.950 |
| ✔ | ✗ | -1.137 | **0.837** | 0.838 | -0.272 | **0.629** | 3.632 | -2.337 | **0.472** | **0.948** |
| ✔ | ✔ | **-1.135** | 0.836 | **0.837** | **-0.270** | 0.628 | **3.614** | **-2.334** | **0.472** | **0.948** |

Table 17: Effects of difference numerical approximation methods on event prediction task. (default to linear approximation instead of the Gauss-Legendre quadrature method).

| P \ Dataset | Synthetic | | | Neonate | | Traffic | | |
|---|---|---|---|---|---|---|---|---|
| | LL (↑) | ACC (↑) | RMSE (↓) | LL (↑) | RMSE (↓) | LL (↑) | ACC (↑) | RMSE (↓) |
| Approxiamation Method = *Linear* | | | | | | | | |
| – | -0.535 | **0.842** | **0.192** | **-2.550** | **9.233** | **0.635** | **0.822** | **0.328** |
| Approxiamation Method = *Gauss-Legendre Quadrature* | | | | | | | | |
| 2 | -0.545 | **0.842** | 0.198 | -2.571 | 9.235 | 0.599 | 0.817 | 0.330 |
| 3 | **-0.531** | **0.842** | 0.194 | -2.594 | 9.236 | 0.610 | 0.820 | **0.328** |
| 4 | -0.556 | **0.842** | 0.199 | -2.590 | 9.253 | 0.589 | 0.819 | **0.328** |
| 5 | -0.550 | **0.842** | 0.193 | -2.602 | 9.258 | 0.601 | 0.820 | **0.328** |

| P \ Dataset | MIMIC | | | BookOrder | | | StackOverflow | | |
|---|---|---|---|---|---|---|---|---|---|
| | LL (↑) | ACC (↑) | RMSE (↓) | LL (↑) | ACC (↑) | RMSE (↓) | LL (↑) | ACC (↑) | RMSE (↓) |
| Approxiamation Method = *Linear* | | | | | | | | | |
| – | -1.135 | 0.836 | **0.837** | **-0.270** | **0.628** | 3.614 | **-2.332** | **0.473** | **0.948** |
| Approxiamation Method = *Gauss-Legendre Quadrature* | | | | | | | | | |
| 2 | -1.144 | 0.834 | 0.840 | -0.291 | 0.627 | 3.653 | -2.337 | 0.472 | **0.948** |
| 3 | -1.142 | **0.837** | 0.839 | -0.294 | 0.627 | 3.667 | -2.337 | 0.471 | **0.948** |
| 4 | -1.143 | **0.837** | 0.839 | -0.287 | 0.627 | 3.647 | -2.340 | 0.471 | **0.948** |
| 5 | **-1.132** | 0.834 | 0.838 | -0.290 | **0.628** | **3.563** | -2.338 | 0.471 | **0.948** |

The experimental results can be found in Table 17. As shown in the table, our model exhibits low sensitivity to the choice of approximation methods.

### D.3  Effects of Tolerance Errors in ODESolver

ODE solvers play a crucial role in ensuring that the model's output remains within a specified tolerance of the true solution [9]. By adjusting this tolerance, we can observe changes in the behavior of the neural network. Given our use of a fixed-step ODE solver, we sought to investigate the impact of the step size on prediction performance throughout the experiment. Specifically, we explored different step sizes from the set $\{0.5, 0.1, 0.05, 0.01, 0.005\}$. A smaller step size corresponds to a greater number of forward passes required to solve the ordinary differential equations, resulting in more accurate solutions at the expense of increased computational time. We conducted extensive experiments on 6 datasets for the event prediction task. The experimental results are presented in Table 18. The table reveals that our model demonstrates low sensitivity to the tolerance error. Therefore, we set the tolerance error to $0.1$ in order to make a desirable balance between computational time and prediction accuracy. We explain the phenomena in two aspects. First, our framework, similar to the

Table 18: Effects of tolerance errors in ODESolver on event prediction task (default to 0.5 for Neonate dataset and 0.1 for other datasets). (The use of a double dash (–) indicates that the experiment requires an excessively long duration of time.)

| Dataset | Synthetic | | | Neonate | | Traffic | | |
|---|---|---|---|---|---|---|---|---|
| $P$ | LL (↑) | ACC (↑) | RMSE (↓) | LL (↑) | RMSE (↓) | LL (↑) | ACC (↑) | RMSE (↓) |
| 5e-1 | -0.559 | 0.841 | 0.196 | **-2.550** | 9.233 | 0.614 | 0.820 | **0.328** |
| 1e-1 | **-0.535** | **0.842** | **0.192** | -2.561 | **9.229** | **0.635** | **0.822** | **0.328** |
| 5e-2 | -0.549 | **0.842** | 0.195 | -2.568 | 9.236 | 0.620 | 0.819 | 0.329 |
| 1e-2 | -0.539 | **0.842** | 0.195 | -2.573 | 9.233 | 0.619 | 0.820 | 0.329 |
| 5e-3 | -0.549 | **0.842** | 0.199 | -2.559 | 9.238 | 0.613 | 0.819 | 0.329 |

| Dataset | MIMIC | | | BookOrder | | | StackOverflow | | |
|---|---|---|---|---|---|---|---|---|---|
| Stepsize | LL (↑) | ACC (↑) | RMSE (↓) | LL (↑) | ACC (↑) | RMSE (↓) | LL (↑) | ACC (↑) | RMSE (↓) |
| 5e-1 | -1.138 | **0.837** | **0.837** | -0.268 | **0.629** | 3.619 | -2.333 | **0.473** | **0.948** |
| 1e-1 | **-1.135** | 0.836 | **0.837** | -0.270 | 0.628 | 3.614 | **-2.332** | **0.473** | **0.948** |
| 5e-2 | -1.137 | 0.835 | **0.837** | **-0.263** | **0.629** | **3.609** | **-2.332** | **0.473** | **0.948** |
| 1e-2 | -1.143 | 0.836 | 0.838 | **-0.263** | **0.629** | 3.612 | – | – | – |
| 5e-3 | -1.147 | **0.837** | 0.838 | – | – | – | – | – | – |

Transformer architecture, circumvents cumulative errors by eliminating the necessity of sequentially passing neural networks in a regressive manner. Second, since the output from the attention module is actually a weighted sum of the tokens, the total variance is lowered. For instance, assume that $X_1, X_2, ..., X_N \sim \mathcal{N}(\mu, \sigma^2)$, then the total variance of the mean value $\bar{x} = \frac{1}{N}(X_1 + X_2 + ... + X_N)$ is $\frac{\sigma^2}{N}$, and the variance is significantly lowered given the large . Overall, our model is not sensitive to tolerance error, which makes our model robust to different scenarios.

### D.4 Effects of Vector Fields in ODE

Neural ODEs are controlled by a vector field that defines the change of $x$ over time, i.e.,

$$\frac{d\mathbf{x}}{dt} = f(\mathbf{x}, t) . \tag{56}$$

where $f : \mathbb{R}^{d+1} \to \mathbb{R}^d$. We investigate two types of $f$ to implement Eq. (1).

Specifically, we denote Concat as

$$f(\mathbf{x}, t) = \text{Actfn}(\text{Linear}^{d,d}(\text{Linear}^{d,d}(\mathbf{x}) + \text{Linear}^{1,d}(t))) , \tag{57}$$

and ConcatNorm as

$$f(\mathbf{x}, t) = \text{Actfn}(\text{LN}(\text{Linear}^{d,d}(\text{Linear}^{d,d}(\mathbf{x}) + \text{Linear}^{1,d}(t)))) , \tag{58}$$

where Actfn$(\cdot)$ is either tanh or sigmoid activation function, Linear$^{a,b}(\cdot) : \mathbb{R}^a \to \mathbb{R}^b$: is a linear transformer from dimension $a$ to dimension $b$, LN denotes the layer normalization. We conduct the experiment on 6 datasets for event prediction task. The experimental results are shown in Table 19. As shown in the table, incorporating normalization before the activation function leads to improved predictive performance.

### D.5 Effects of Dropout Rate

Dropout in the Transformer can prevent overfitting and boost performance. However, with the increment of the dropout rate, we output of ContiFormer tends to be less continuous, leading to a sub-optimal solution for tasks that prefer a continuous-time output. To this end, we conduct experiments on both modeling continuous-time function and event prediction tasks. As for the task of modeling continuous-time function, it is desirable to have a dynamic system where outputs are "almost" continuous and smooth. As shown in Figure 5, the performance of both Transformer and ContiFormer drops significantly as the dropout rate increases. For prediction tasks, the dynamic system can be shaky, which reflects noise or other uncontrolled influences on the system. The experimental results on event prediction can be found in Table 20. As shown in the table, the dropout rate has low sensitivity to the prediction results.

Table 19: Effects of different vector field designs on event prediction task. Attn. stands for Attention. ✔ refers to ConcatNorm, while ✘ refers to Concat.

| Dataset | | Synthetic | | | Neonate | | Traffic | | |
|---|---|---|---|---|---|---|---|---|---|
| Norm | Actfn | LL (↑) | ACC (↑) | RMSE (↓) | LL (↑) | RMSE (↓) | LL (↑) | ACC (↑) | RMSE (↓) |
| ✘ | tanh | -0.552 | **0.842** | 0.198 | **-2.550** | 9.233 | **0.639** | 0.820 | 0.329 |
| ✘ | sigmoid | -0.561 | **0.842** | 0.201 | -2.602 | 9.263 | 0.635 | **0.822** | **0.328** |
| ✔ | tanh | -0.556 | **0.842** | 0.199 | -2.584 | 9.242 | 0.628 | 0.820 | **0.328** |
| ✔ | sigmoid | **-0.535** | **0.842** | **0.192** | -2.601 | **9.232** | 0.616 | 0.819 | 0.329 |

| Dataset | | MIMIC | | | BookOrder | | | StackOverflow | | |
|---|---|---|---|---|---|---|---|---|---|---|
| Norm | Actfn | LL (↑) | ACC (↑) | RMSE (↓) | LL (↑) | ACC (↑) | RMSE (↓) | LL (↑) | ACC (↑) | RMSE (↓) |
| ✘ | tanh | -1.162 | 0.831 | **0.837** | -0.274 | 0.627 | 3.615 | -2.337 | 0.472 | 0.948 |
| ✘ | sigmoid | -1.144 | **0.836** | 0.838 | **-0.270** | **0.628** | **3.614** | -2.334 | 0.472 | 0.948 |
| ✔ | tanh | -1.157 | 0.833 | 0.840 | -0.307 | 0.622 | 3.681 | -2.335 | **0.473** | **0.947** |
| ✔ | sigmoid | **-1.135** | **0.836** | **0.837** | -0.302 | 0.625 | 3.689 | **-2.332** | **0.473** | 0.948 |

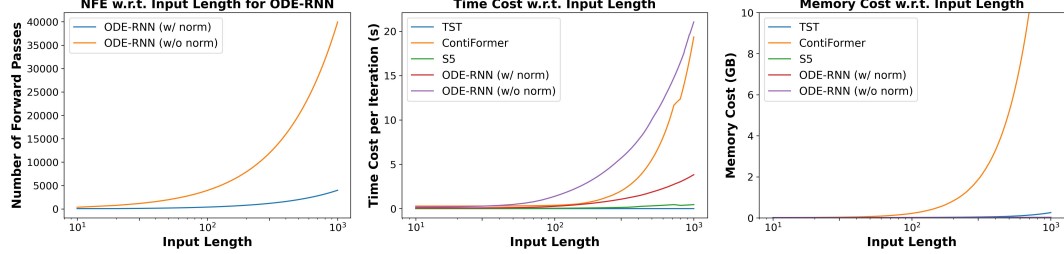

Figure 5: Interpolation results on 2D spiral under different dropout rates.

## D.6 Effects of Time Normalization

As shown in Eq. (3) and Eq. (5), due to the nature of sequence data, it is common to encounter abnormal points, such as events with significantly large time differences. To avoid numeric instability during training, we divide the integrated solution by the time difference. To examine the effect of normalization, we conduct experiments on 6 datasets for event prediction task. The experimental results can be found in Table 21, which shows that the normalization is significant on most datasets, including Synthetic, Neonate, MIMIC and StackOverflow datasets.

## E Time Cost v.s. Memory Cost v.s. Accuracy

Figure 6: Time Cost v.s. Memory Cost v.s. Accuracy.

Utilizing continuous-time modeling in ContiFormer often results in substantial time and GPU memory overhead. We have conducted a comparative analysis of ContiFormer's actual time costs in comparison to Transformer-based models (TST), ODE-based models (ODE-RNN, Neural CDE), and SSM-based models (S5). Throughout our analysis, we employ an RK4 solver for ODE-based models and our model, setting the step size to 0.1 and 0.01. It's worth noting that, for ODE-RNN and other

Table 20: Effects of dropout rate in ContiFormer on event prediction task (default to 0.1).

| Dataset / Dropout | Synthetic LL (↑) | ACC (↑) | RMSE (↓) | Neonate LL (↑) | RMSE (↓) | Traffic LL (↑) | ACC (↑) | RMSE (↓) |
|---|---|---|---|---|---|---|---|---|
| 0 | -0.557 | **0.842** | **0.191** | -2.588 | **9.215** | 0.633 | **0.824** | **0.328** |
| 0.1 | **-0.535** | **0.842** | 0.192 | **-2.550** | 9.233 | **0.635** | 0.822 | **0.328** |
| 0.3 | -0.644 | **0.842** | 0.223 | -2.583 | 9.242 | 0.577 | 0.814 | 0.332 |
| 0.5 | -0.696 | 0.841 | 0.230 | -2.579 | 9.280 | 0.525 | 0.813 | 0.331 |

| Dataset / Dropout | MIMIC LL (↑) | ACC (↑) | RMSE (↓) | BookOrder LL (↑) | ACC (↑) | RMSE (↓) | StackOverflow LL (↑) | ACC (↑) | RMSE (↓) |
|---|---|---|---|---|---|---|---|---|---|
| 0 | -1.193 | 0.825 | 0.839 | -0.268 | 0.629 | 3.616 | -2.338 | 0.472 | 0.949 |
| 0.1 | **-1.135** | **0.836** | 0.837 | -0.270 | 0.628 | 3.614 | **-2.332** | **0.473** | **0.948** |
| 0.3 | -1.142 | 0.833 | **0.836** | **-0.266** | **0.630** | **3.608** | -2.336 | 0.472 | **0.948** |
| 0.5 | -1.149 | 0.834 | 0.844 | -0.270 | 0.629 | 3.624 | -2.342 | 0.471 | 0.948 |

Table 21: Effects of normalization on event prediction task. Attn. stands for Attention. ✔ refers to normalizing the attention/value as shown in Eq. (3)/Eq. (5), while ✗ refers to removing the normalization, i.e. $t - t_i$. Note that ContiFormer use both normalization for attention and values.

| Dataset Attn. | Value | Synthetic LL (↑) | ACC (↑) | RMSE (↓) | Neonate LL (↑) | RMSE (↓) | Traffic LL (↑) | ACC (↑) | RMSE (↓) |
|---|---|---|---|---|---|---|---|---|---|
| ✗ | ✗ | -0.618 | **0.842** | 0.215 | -2.612 | 9.277 | **0.721** | **0.822** | 0.329 |
| ✗ | ✔ | -0.601 | 0.841 | 0.206 | -2.576 | 9.240 | 0.536 | 0.814 | 0.336 |
| ✔ | ✗ | -0.584 | **0.842** | 0.205 | -2.601 | 9.272 | 0.576 | 0.818 | **0.327** |
| ✔ | ✔ | **-0.535** | **0.842** | **0.192** | **-2.550** | **9.233** | 0.635 | **0.822** | 0.328 |

| Dataset Attn. | Value | MIMIC LL (↑) | ACC (↑) | RMSE (↓) | BookOrder LL (↑) | ACC (↑) | RMSE (↓) | StackOverflow LL (↑) | ACC (↑) | RMSE (↓) |
|---|---|---|---|---|---|---|---|---|---|---|
| ✗ | ✗ | -1.167 | 0.826 | 0.848 | -0.311 | 0.627 | 3.710 | -2.346 | 0.470 | **0.948** |
| ✗ | ✔ | -1.149 | 0.833 | 0.840 | **-0.269** | **0.630** | 3.634 | -2.341 | 0.471 | **0.948** |
| ✔ | ✗ | -1.172 | 0.826 | 0.844 | -0.277 | 0.629 | 3.654 | -2.336 | **0.472** | **0.948** |
| ✔ | ✔ | **-1.135** | **0.836** | **0.837** | -0.270 | 0.628 | **3.614** | **-2.334** | **0.472** | **0.948** |

ODE-based models, the input times have an impact on the number of forward passes when using a fixed-step ODE solver, thus, affecting the time costs.

The left figure in Figure 6 illustrates the influence of input times concerning input length. For the model with normalization (a.k.a., w/ norm), the input times consistently range from $[0, 1]$, regardless of the input length. Conversely, for the model without normalization (a.k.a., w/o norm), the input times span from $[0, L]$, where $L$ represents the input length.

As depicted in the middle figure in Figure 6, our model performs comparably to ODE-RNN (w/ norm) when the input length is up to 100, but becomes approximately four times slower as the input length extends to 1000. The right figure in Figure 6 displays GPU memory usage, revealing a significantly higher memory cost associated with our model.

Table 22 provides a more comprehensive overview of different models on UEA datasets. It is evident that our model strikes a better balance between accuracy and time cost. Specifically, our model achieves the best performance on 3 out of 6 selected datasets while incurring only twice the time cost compared to S5.

Table 22: Time Cost v.s. Accuracy for different models on UEA classification datasets (0% dropped). For all ODE-based models and our model, we adopt the RK4 solver and step size refers to the amount of time increment for the RK4 solver. ↑ (↓) indicates the higher (lower) the better. Note that since the results are aggregated over 3 random seeds, achieving the same results doesn't indicate the same outcome for each random seed.

| Dataset | | BasicMotions | | JapaneseVowels | | Libras | |
|---|---|---|---|---|---|---|---|
| Model | Step Size | Time Cost (↓) | Accuracy (↑) | Time Cost (↓) | Accuracy (↑) | Time Cost (↓) | Accuracy (↑) |
| TST | - | 0.56× | 0.975 | 0.10× | 0.986 | 0.36× | 0.843 |
| S5 | - | 1× | 0.958 | 1× | 0.926 | 1× | 0.655 |
| ODE-RNN (w/o norm) | 0.1 | 3.06× | 0.917 | 0.79× | 0.952 | 2.67× | 0.602 |
| | 0.01 | 18.94× | 0.925 | 5.26× | 0.955 | 16.29× | 0.611 |
| ODE-RNN (w/ norm) | 0.1 | 1.46× | **1.000** | 0.32× | 0.980 | 1.24× | 0.637 |
| | 0.01 | 1.59× | **1.000** | 0.47× | 0.980 | 1.59× | 0.619 |
| Neural CDE | 0.1 | 4.47× | 0.958 | 1.16× | 0.932 | 3.87× | 0.763 |
| | 0.01 | 34.26× | 0.958 | 9.57× | 0.942 | 28.48× | 0.744 |
| ContiFormer | 0.1 | 2.89× | 0.975 | 0.49× | 0.990 | 2.26× | **0.870** |
| | 0.01 | 13.75× | 0.975 | 1.87× | **0.991** | 8.74× | **0.870** |
| Dataset | | NATOPS | | PEMS-SF | | RacketSports | |
| Model | Step Size | Time Cost (↓) | Accuracy (↑) | Time Cost (↓) | Accuracy (↑) | Time Cost (↓) | Accuracy (↑) |
| TST | - | 0.32× | **0.963** | 0.15× | 0.890 | 0.14× | 0.826 |
| S5 | - | 1× | 0.839 | 1× | **0.896** | 1× | 0.772 |
| ODE-RNN (w/o norm) | 0.1 | 2.67× | 0.909 | 4× | 0.778 | 2.47× | 0.827 |
| | 0.01 | 16.52× | 0.907 | 28.45× | 0.761 | 14.65× | 0.796 |
| ODE-RNN (w/ norm) | 0.1 | 1.28× | 0.898 | 1.47× | 0.775 | 1.15× | 0.785 |
| | 0.01 | 1.45× | 0.898 | 1.44× | 0.775 | 1.57× | 0.787 |
| Neural CDE | 0.1 | 3.83× | 0.789 | 6.04× | 0.763 | 3.48× | 0.743 |
| | 0.01 | 29.24× | 0.787 | 50.36× | 0.763 | 25.04× | 0.748 |
| ContiFormer | 0.1 | 2.32× | 0.935 | 5.45× | 0.823 | 1.87× | **0.836** |
| | 0.01 | 9.76× | 0.935 | 37.08× | 0.823 | 5.88× | **0.836** |

# F  Broader Impact

The proposed model, ContiFormer, presents several potential impacts within the field of machine learning and irregular time series analysis. These include:

- Enhanced Modeling Capability: ContiFormer extends the Transformer architecture to effectively capture complex continuous-time dynamic systems. By leveraging continuous-time modeling techniques, the model demonstrates improved performance in handling irregular time series data, offering a more flexible and powerful modeling ability.

- Practical Applications: Our proposed method achieves state-of-the-art performance in a variety of tasks and real-life datasets, which makes it more promising to tackle real-world applications. The ability to model continuous-time dynamics offers valuable insights into time-evolving systems, facilitating improved decision-making and predictive capabilities in diverse domains.

