# OpenReview forum: "ContiFormer: Continuous-Time Transformer for Irregular Time Series Modeling"
_NeurIPS.cc/2023/Conference — NeurIPS 2023 poster_

### Official Review · Reviewer_jYof · 2023-06-19

**Soundness:** 2 fair
**Presentation:** 1 poor
**Contribution:** 3 good
**Rating:** 5
**Confidence:** 4

**Summary:**

The authors propose a new method, a continuous-time transformer, ContiFormer, for irregular time series modelling. The proposed method extends vanilla Transformer to a continuous domain. In particular, the model incorporates the continuous dynamic modelling of Neural ODE with the attention mechanism of a transformer. Moreover, the authors show that many transformer variants designed for irregular time series modelling are a special case of the proposed ContiFormer. The authors evaluated the method on a continuous time function (spiral), irregular sampled time series for classification and irregular time series for event prediction. The results indicate that the proposed method obtains better results than the baseline models.

**Strengths:**

Originality:
The proposed work is original. The author's propose a novel continuous-time attention mechanism (CT-MHA).

Quality:
The authors have included also a comparison of complexity analysis of their method compared to other. The authors have tested the proposed method on multitude of datasets: irregularly-sampled time series for classification with increasing number of data dropped, as well as event prediction. Especially for the latter, the authors compare their method to existing state-of-the-art models for event forecasting (SAHP, THP) and obtain an improved performance.

Clarity:
Please see weaknesses.

Significance:
The authors provide a unique theoretical approach, and the obtained results demonstrate a benefit of the proposed method.


**Weaknesses:**

Originality, related work:
The related work is a bit convoluted. It would be better if the authors divide the related work into 3 subsections: Transformers for time-series modelling, Transformer for irregular time series, Continuous time models (NODEs). The literature of RNN is not as relevant to this work as it builds upon transformers and NODEs.

With respect to NODE related work:
Line 43-45, the authors say 'the recursive nature of Neural ODEs can lead to cumulative errors if the number of iterations is numerous'. This is not precise. The cumulative error is dependent of the solver used, and there are numerous works which by enforcing, for example, conservation of energy, keep the error tolerance low, for example, https://arxiv.org/abs/1909.12077. Moreover, by selecting difference step size, solver type (adaptive or fixed) one can control/adjust for the numerical error (as mentioned in https://arxiv.org/abs/1806.07366). Please rephrase.

With respect to Transformer related work:
In line 47-48, the authors mention that due the fixed time encoding or learning upon certain kernel functions the resulting models struggle to model complex data patterns in real-world practice.  However, I do not see how this claim is supported. Even in the appendix, for given percentages of observations dropped (50%) mTAN model outperforms ContiFormer. Therefore, the claim is not supported. Please rephrase.

The proposed work differs from the existing work in the field, however,  some of the claims made about the previous work are not well supported in the current version of the paper.

Quality:
The proposed continuous time attention mechanism, eq. 3, is unclear, as the underlying functions are not clearly defined. The authors' mention that previous transformer works for irregular time series are special cases of their model, however, the proof of this only mentioned in the Appendix, I would recommend moving the core parts of the proof to the main part of the paper.

Lines 281 - 283, the authors say that ContiFormer consistently outperforms all the baselines on all three settings, while in the appendix it can be seen that there are also instances were the baseline models outperform ContiFormer, therefore, I would soften the claim.

For lines 303-304, I am missing a comparison to methods that are more aligned with the task at hand. The authors compare against TST and mTAN, however TST  model is focused on multivariate time series (not decomposition of different dynamic trends), while mTAN although also designed for irregular time series modelling the paper's focus is on temporally distributed latent representations. For extracting temporal dynamics/trends of the data, there are other works, Autoformer (https://arxiv.org/abs/2106.13008), FEDformer (https://arxiv.org/abs/2201.12740) that would be perhaps a more fair comparison to the present method.

Clarity:
The submission is not clearly written, thus negatively affecting the understanding of the proposed method. Please clarify for all functions, v(), k(), q() the input, output dimensionalities, as well what these functions are. As this is unclear, it is hard to evaluate what exactly the inner product of the two function spaces is computing.


Line 147. Imprecise phrasing: the authors mention that self-attention involves calculating the correlation between queries and keys, which is incorrect. The computation represents an inner dot product, which is not a correlation metric but rather a similarity measure, please rephrase.

The main figure (figure 1), is also not immediately clear for the reader. Would suggest to adjust the figure.  It is hard to follow all the colors and how they correspond to which input.

In addition, for the NODE model compared with, as well as for the ODESolver used in author own work it is missing what kind of solver is used: adaptive or fixed step? This could greatly affect the performance, hence, it is a crucial implementation detail.

In the appendix mTAN model has the wrong reference. The reference is for the Autoformer model, please correct.

Lastly, for the experiments is it not clear what is the input/ouput data length for the model.

Overall, the technical details of the paper are not clear. As this is the main contribution of the paper, it is essential that it is clear so that the extensive experimental results can be objectively evaluated given the method at hand. I would suggest the authors to rewrite this section.


**Questions:**


In line 83, the authors say that previous models 'are insufficient in capturing input-dependent dynamic systems'. How is this claim supported? The previous work also is also condition on the input data.

In line 99-100, the author's say 'ContiFormer, employs a non-autoregressive paradigm', however, the vanilla transformer decoder at inference time is an auto-regressive model, can you clarify this? Furthermore, to perform the ODESolver step, you perform auto-regressive P steps?

Eq. (1), the formulation is unclear and incorrect: the author's define both the ODE as well as the continuous latent variable with the same parameter. Furthermore, if k_i(\tau) is the ODE, what is f()?

In line 136, could you please clarify why you have modeled each observation as a separate ODE? As showed in https://arxiv.org/abs/1806.07366 NODE models can model irregular time-series with a single ODE specification, even more so, close by observations most likely follow the same ODE.

In line 136, could you clarify if the ODE is a MLP or any other architecture?

Line 138: would suggest the authors to adjust this assumption/claim: from a dynamical systems perspective a future event cannot affect a past event. Vast majority of works modelling dynamics are often based on Markov property.

line 144 in text you mention that q(t_i) is a cubic spline approximation, however, in the appendix you clarify that linear interpolation is used due to the lower computational complexity. please clarify the approximation used in your proposed method. What is the dimensionality of Q_i?


Eq. (4) Based on the introduced notation it seems like the introduced computation ends up being a matrix (Q_i), vector (k_i() \in R^d) product. Where the Q_i is an approximation of the process via cubic spline, and k_i() is the latent trajectory by an ODE, this would imply that the proposed mechanism measure the similarity of the learned approximation of the ODE to the approximation by the cubic spline function. Is this correct?

Eq. (7) For multi-head attention, you also apply a linear transformation to the latent continuous state, q(),v()?

How sensitive is the method of using a different approximation method instead of cubic spline?

Does each latent continuous trajectory span only two adjacent time points t:[-1,1]? Meaning that depending input length L (data points), you would have L independent latent ODE trajectories? It is unclear from the text.

The authors mention that their attention mechanism is measuring the correlation between different observations (line 298), therefore, I would suggest the authors to compare their work to Autoformer (that measures correlation between different time series) or Fourier-based (FEDformer) attention mechanism. Have such comparisons been made?

**Limitations:**

The authors have not explicitly addressed the limitation of the presented work. Based on the text, the proposed method models each observation with a separate ODE function. This, however, seems to be sub-optimal as data points close in time probably follow the same underlying dynamics function, furthermore, the authors have not explained why such a design choice would be necessary, as by default Neural ODEs can account for irregularly sampled time series. Moreover, it would be nice if the authors would have addressed (as an ablation) the model's performance on regular time-series data compared to existing state-of-the art methods.

---

> ### Author Rebuttal · Authors · 2023-08-10
>
> Thank you for your valuable suggestion. We will diligently reassess the detailed statements made in the paper and diligently bolster each claim with appropriate supporting evidence. Below we try to address your problems.
> > **W1: Table 8 mTAN model sometimes outperforms ContiFormer.**
>
> Contiformer outperformed mTAN on 18 datasets, while mTAN only outperformed our model on 2 datasets. In Table 3 in the main paper, Contiformer consistently outperformed mTAN on all three data settings (30%, 50%, 70% drop-ratio).
> > **W2: Which ODESolver was used and what kind of steps are missing**
>
> We use fixed-step RK4 solver, L578-581 in Appendix B.3.
> > **W3: What is the input/output data length**
>
> Input: $X\in \mathbb{R}^{N \times f}$ with $N$ as sequence length and $f$ as input feature dimension; Output is in $\mathbb{R}^{N \times d}$ where $d$ is the hidden dimension.
>
> > **Q1: L83, clarify 'previous works are insufficient in capturing input-dependent dynamic systems'**
> Most Transformer-based methods, with the exception of mTAN, capture input-dependent patterns, rather than the underlying dynamic systems, which is a fundamental mathematical and conceptual framework used to describe the evolution of variables over time.
>
> For mTAN, although it models continuous-time dynamic systems, its attention mechanism is primarily dependent on time variables, rather than the input information. Consequently, its behavior lacks input-dependent attributes, a crucial factor for effective modeling in complex scenarios.
> > **Q2: In L99, clarify 'ContiFormer, employs a non-autoregressive paradigm'.**
>
> Here "autoregressive" means that the stepwise model inference is upon the output from the last timestep, even given the whole-length sequence input. We mean non-autoregressive for the *encoding part* of Transformer model and our model.
> > **Q3: if $k_i(\tau)$ is the ODE, what is $f()$?**
>
> $k_i(\tau)$ is a time-function controlled by ODE, and $f()$ is implemented as an MLP.
> > **Q4: L136, why modeled each observation as a separate ODE?**
>
> Each observation will derive a corresponding trajectory modeling *result*, rather than assuming each observation follows different ODEs, since all these ODE functions modeling the latent trajectories share the same ODE function parameter, i.e., $\theta_k$ and $\theta_v$ in Eq. (1).
> > **Q5: L138, from a dynamical systems perspective a future event cannot affect a past event.**
>
> Our method can model the relationship among these observations through continuous-time attention, just like the pairwise attention modeling in vanilla Transformer model. Sorry for the confusion.
> > **Q6: The approximation used for $q(t_i)$. What is the dimensionality of $Q_i$?**
>
> We use cubic spline function to obtain $q(t_i)$ and $Q_i \in \mathbb{R}^d$.
> > **Q7: The attention mechanism ends up being a matrix, vector product.**
>
> $q(\cdot): \mathbb{R} \rightarrow \mathbb{R}^{d}$ is a time function, *not* a matrix. $k_i(\cdot): \mathbb{R} \rightarrow \mathbb{R}^{d}$ is also a time function. We actually calculate the similarity (or relationship) between continuous functions within a closed interval. Contiformer extends the vanilla Transformer into the continuous-time domain, we emphasize that our attention mechanism operates directly on continuous-time inputs, in contrast to discrete matrices/vectors.
> > **Q8: Eq. (7) For multi-head attention, is linear transformation applied?**
>
> Yes, as shown in Eq. (7) of our paper.
> > **Q9: How sensitive is the method of using a different approximation method instead of cubic spline?**
>
> We have conducted an ablation study on UEA classification and our model is insensitive to the utilized approximation method.
>
> |Drop Ratio (%)|70||
> |-|-|-|
> ||Linear Interpolation|Cubic Spline|
> |**Avg. Acc**|**0.7775**|0.7749|
> |**# Top 1**|**13**|7|
> > **Q10: Does each latent continuous trajectory span only two adjacent time points t:[-1,1]?**
>
> Latent continuous trajectory evolves along time, i.e., from $t_1$ to $t_N$, where $t_i$ is the time of the $i$-th observation ($i \in [1,N]$). Therefore, it does *not* span only two adjacent time points.
> > **Q11: The proposed method models each observation with a separate ODE function. This seems to be sub-optimal.**
>
> We would like to clarify that all the observations are modeled with *the same* ODE function with only one set of parameters $\theta$ in Eq. (1), *rather than* modeling by *different* ODE functions.
> > **Q12: Why such a design choice would be necessary?**
>
> The necessary is discussed in **Q4, Q10**, and **Q11**. Besides, Neural ODEs can only calculate the ODE function from the initial time point to the last one, while our method can calculate the ODE function from $t_1$ to $t_N$ separately and parallelly. Please refer to **Q3 in General Response** for more details.
> > **Q13: Experiment on regular time series.**
>
> We further experiment additionally on UEA classification following section 4.2.
>
> |Drop Ratio (%)|30|||50|||70|||
> |-|-|-|-|-|-|-|-|-|-|
> ||AutoFormer|FedFormer|ContiFormer|AutoFormer|FedFormer|ContiFormer|AutoFormer|FedFormer|ContiFormer|
> |**Avg. ACC**|0.7035|0.5994|**0.8126**|0.6900|0.5655|**0.7997**|0.6368|0.5312|**0.7749**|
> |**Avg. Rank**|2|2.7619|**1.1905**|1.8571|2.8571|**1.2857**|1.9524|2.7619|**1.1905**|
>
> We also investigate Contiformer's performance on long times series forecasting using Exchange dataset. Contiformer can outperform both models. Following Autoformer, input length is $96$ and the target horizon is in $[96, 192, 336]$.
>
> |Exchange dataset|Autoformer||FEDformer||Contiformer||
> |-|-|-|-|-|-|-|
> |Horizon len.|MSE|MAE|MSE|MAE|MSE|MAE|
> |96|0.1409|0.2711|0.1356|0.2643|**0.1301**|**0.2521**|
> |192|0.8544|0.6915|0.2764|0.3833|**0.1966**|**0.3193**|
> |336|0.707|0.6445|0.4657|0.5069|**0.4315**|**0.4852**|
>
> We may not conduct full experiments on all the datasets, but we believe these additional experiments have addressed your concerns about our work.
>
> [1] Informer: Beyond efficient transformer for long sequence time-series forecasting. AAAI 2021.

---

> > ### Comment · Reviewer_jYof · 2023-08-12
> >
> > Thank you for the detailed response to my questions voiced. I appreciate the authors effort to perform additional comparisons with Autoformer and FEDformer, and clarification on the model details, as such I will increase my score to 5.

---

> > > ### Author Response · Authors · 2023-08-13
> > > **Thank you for your reply!**
> > >
> > > We express our sincere gratitude for your thoughtful response and specific recommendations regarding our paper. Your insightful suggestions and comments have significantly contributed to enhancing the quality of our work, encompassing aspects such as related research, theorems, and technical intricacies.
> > >
> > > We sincerely thank you for raising the rating score regarding our paper. We are also open to further comments and suggestions for improving our paper.

---

### Official Review · Reviewer_GCXG · 2023-07-03

**Soundness:** 3 good
**Presentation:** 3 good
**Contribution:** 3 good
**Rating:** 7
**Confidence:** 3

**Summary:**

This paper introduces the ContiFormer, a novel continuous-time Transformer model designed for handling irregular time series data. In this model, the keys, queries and values are vector-valued functions indexed on time. Each input observation gives rise to a key function, given by the solution of a Neural ODE started at an initial condition determined by the observation. While a similar model is adopted for the value functions, the query function is obtained using natural cubic spline interpolation of discrete query values. By replacing dot products (between discrete queries and keys) with an $L^2$ inner product, a continuous-time attention function is derived. The continuous-time output can be evaluated on a discrete time grid allowing for the stacking of such layers. To reduce the computational time, the paper leverages a reparametrization trick, which makes it possible to solve simultaneously multiple Neural ODEs on different time intervals. The ContiFormer model is evaluated on various time series tasks and datasets demonstrating its effectiveness in handling irregular time-series data.

**Strengths:**

- Given the empirical success of transformer models on machine learning tasks involving sequential data, enhancing their capability of effectively handling irregular time series data is an important research question.
- This paper proposes an elegant solution based on Neural ODEs and other ideas from the field of neural differential equations such as the use of cubic splines in Neural CDEs.
- The paper is clearly written and easy to follow.
- The empirical evaluation is well conducted, extensive and provides a compelling demonstration of the superior performance of the ContiFormer model.

**Weaknesses:**

- The theoretical results are deferred to the appendix. I think the main theorem (Thm 1 in Appendix) should at least be formally stated and further commented on in the main paper. This would better motivate and support the modelling choices.

- The paper is a bit repetitive at times (l100-102, l108-117, l175-178) while some components might deserve further explanation: 1. the main theoretical result;  2. although the $i^{th}$ key and value functions are well defined at $t<t_i$, it could be further commented by expanding l138, and perhaps adapting Fig. 1 with $t<t_i$; 3. the rationale for modelling the key and value functions differently from the query function.

- As mentioned in the related work section, a continuous-time attention model based on Neural ODEs has previously been proposed in [10]. While several baselines are considered in the experimental section it is surprising that this model is not included without a justification.

- The attention model is not fully formulated in continuous-time, in the sense that it is expressed as a sum over the number of value functions which depends on the number of observations. A natural thought of experiment for a continuous-time model is to replace the input sequences with continuous paths. Is there a way to define the ContiFormer on such paths?

**Questions:**

- Why did you choose to model the key and value functions differently from the query function? Can the ContiFormer handle partially observed data, given how the key and value functions are defined?
- Do you have an intuition/explanation for the fact that $P$ can be chosen to be very small (l 224) and the insensitivity of the ContiFormer to the tolerance error (Appendix F12, F13)?
- Related to the comment above about the continuous-time formulation, does the ContiFormer offer any advantage for handling both irregular and long time series?
- Although computational complexities are provided, would it be possible to report the actual time it takes to train and evaluate the model?


**Limitations:**

Yes, some limitations have been discussed.

---

> ### Author Rebuttal · Authors · 2023-08-10
>
> We sincerely thank you for your comprehensive comments.
>
> > **W1: The theoretical results should at least be formally stated and further commented on in the main paper.**
>
> Thanks for your great suggestion! In the revised manuscript, we intend to consider your suggestion.
>
> > **W2: L138, adapting functions to $t<t_i$.**
>
> We are pleased to adopt your suggestion to make it more clear for all $t$ in the whole time horizon.
>
> > **W3: Compare Contiformer with CADN [1].**
>
> We re-implement the model and conduct experiments on the UEA classification task. Below, we show a summary of the results. We apologize that due to the time limit of the rebuttal period, we only manage to run 15 out of 20 datasets for CADN. We conclude that Contiformer outperforms CADN on most datasets.
>
>
> |Drop Ratio (%)|30||50||70||
> |-|-|-|-|-|-|-|
> |Metric|CADN|Contiformer|CADN|Contiformer|CADN|Contiformer|
> |Avg. ACC|0.6869|**0.7879**|0.6852|**0.7786**|0.6898|**0.7563**|
> |# Top 1|1|**13**|1|**14**|4|**11**|
>
> > **W4: Define the ContiFormer on continuous paths.**
>
> Thank you for the insightful perspective! ConfiFormer incorporates continuous paths in a latent space. Thus, if the input is a continuous path, Contiformer can directly apply ordinary differential equation (ODE) modeling (as described in Section 3.1) on the input sequence and calculate our proposed continuous-time attention upon that. However, directly modeling continuous paths using the ODE assumption will break the parallel execution of Transformer architecture. Thus, as described in Section 3.3, we further propose a novel reparameterization trick to split the time horizon into several fixed-length pieces $t\in[-1, 1]$, parallelly executing continuous-time attention calculation on these pieces and then map back to the original time horizon without loss of performance. This parallel execution requires sampling timing points on the continuous path, which needs further refinement and exploration in our future work.
>
> > **Q1: Why model key & value differently as query? Can Contiformer handle partially observed data?**
>
> Thanks for your insightful question. The effects of the query function and key and value functions are somehow different. Thus, we separate the roles of query and key/value into distinguishable ones. This design aligns the original attention mechanism such as [2, 3] where the query, key, and value functions are different.
>
> Indeed, Contiformer can handle partially observed data. Our model is designed for irregular time series modeling, including irregularly-sampled data caused by missing time points (if we correctly understand your mentioned partially observed data). The result shown in section 4.2 also supports the clarification, of which observations are randomly dropped. Technically, our model inputs an irregular time series in any format and outputs a continuous-time latent trajectory of the system.
>
> > **Q2: Why $P$ can be very small? Why Contiformer's insensitivity to the tolerance error?**
>
> Thanks for your insightful question. As shown in Tables 12 and 13 in the Appendix, our model is robust to the tolerance error of the ODE solver and the numerical error. We explain the phenomena in two aspects. First, our framework, akin to the Transformer architecture, circumvents cumulative errors by eliminating the necessity of sequentially passing neural networks in a regressive manner. Second, we guess that, since the output from the attention module is actually a weighted sum of the tokens, therefore, the total variance is lowered. For instance, assume that $x_1, ..., x_N \sim \mathcal{N}(\mu, \sigma^2)$, then the total variance of the mean value $\bar{x} = \frac{1}{N}(x_1 + x_2 + ... + x_N)$ is $\frac{\sigma_2}{N}$, and the variance is significantly lowered given large $N$. Overall, our model is not sensitive to tolerance error, which makes our model robust to different scenarios.
>
> > **Q3: Can Contiformer handle irregular long time series?**
>
> As shown in Table 10 in the Appendix, the BookOrder dataset with both irregular properties and a relatively long time series. Our model outperforms all the baselines on this dataset. Nevertheless, due to the nature of Transformer-based architecture, it might necessitate increased memory capacity and a richer training dataset to achieve optimal performance. But we believe that the other works alleviating the burden of memory cost of the vanilla Transformer model may also inspire the improvement of our method, which is a promising direction for future work. Thank you for your valuable comment!
>
> > **Q4: The actual time it takes to train and evaluate the model.**
>
> We appreciate the reviewer's interest in the practical aspects of our work's computational performance. While we have provided comprehensive computational complexity analysis, we acknowledge the importance of reporting actual training and evaluation times for a more practical understanding. We report the actual time in the below table on the Synthetic dataset for event prediction task relative to RMTPP, which is a recurrent neural network. Experiments are conducted on NVIDIA RTX A6000.
>
> |Model|RMTPP|THP|GRU-$\Delta$t (atol=0.1)|mTAN|ContiFormer (atol=0.1)|
> |-|-|-|-|-|-|
> |Relative Time for Training|1 $\times$|1.07 $\times$|3.22 $\times$|1.05 $\times$|6.96 $\times$|
> |Relative Time for Testing|1 $\times$|1.05 $\times$|2.82 $\times$|1.02 $\times$|4.18 $\times$|
>
>
> As demonstrated in the table, our model exhibits an approximate slowdown of $6$ times compared to the vanilla Transformer model (THP), and it demonstrates a $2$ times decrease in speed compared to the ODE-based model (ODE-RNN).
>
> [1] Chien, Jen-Tzung, and Yi-Hsiang Chen. "Learning continuous-time dynamics with attention." TPAMI 2022.
>
> [2] Bahdanau, Dzmitry, Kyunghyun Cho, and Yoshua Bengio. "Neural machine translation by jointly learning to align and translate." ICLR 2015.
>
> [3] Sukhbaatar, Sainbayar, Jason Weston, and Rob Fergus. "End-to-end memory networks." NIPS 2015.

---

> > ### Comment · Reviewer_GCXG · 2023-08-19
> > **Reply to authors rebuttal**
> >
> > Thank you for thoroughly addressing my questions and for your comprehensive rebuttals that are substantiated with further experimental results. Your rebuttal work will certainly enhance the quality of your paper and increases my confidence in my evaluation.

---

> > > ### Author Response · Authors · 2023-08-19
> > > **Thank you for your reply!**
> > >
> > > We sincerely thank your valuable insights and suggestions regarding our paper. We are also open to further comments and suggestions for improving our paper.

---

### Official Review · Reviewer_PCsf · 2023-07-05

**Soundness:** 2 fair
**Presentation:** 3 good
**Contribution:** 2 fair
**Rating:** 5
**Confidence:** 4

**Summary:**

The paper describes a continuous time extention of the transformer architecture using ODE blocks to propagate the effect of each observation individually through time. For computing attention values inner products between functions are used, where in the implementation the resulting integral is approximated. The paper demonstrates the method's application to irregular time-series data and in temporal point processes in different tasks including interpolation, extrapolation, classificaton.



**Strengths:**

The paper gives a continuous-time extension of transformers capable of handling irregularly sampled continuous time series, and point processes.
Computational complexity is assessed in details which is quite important for practical application, and helps the reader to better estimate the real life cost of scaling the methods to real life problems.


**Weaknesses:**

My main problem is with the missing statistical rigor during evaluation. More specifically:

In Table 3 the authors report averages over 20 datasets. I assume these datasets have different sizes, complexity, dataset imbalance etc., therefore it is hard to rigorously evaluate the average metrics.There are dataset-by-dataset level results in the appendix but there is still no standard deviation.

Similarly in Table 4, no standard deviations. Probably 3 std differences are significant, but with 3 repeats I am not sure sure 2 std is very convincing. (For just a very rough estimate how big delta is needed check a 3 sample unpaired t-test table)





**Questions:**

Please give standard deviations and mark significant differences.

Also it is interesting to see that NeuralCD is outperformed by ODE-RNN here for the 30 and 50% dropped case. As there is result provided for CharacterTrajectories in the appendix (this dataset was used in the original NeuralCDE paper) I compared the results.
format:  present paper, vs, Kidger et al.

| Method | 30% dropped | 50% dropped | 70% dropped |
|-----------|-----------|-----------|-----------|
GRU-D |  0.9325 vs. 0.942 | 0.8872 vs. 0.902 | **0.8433 vs. 0.919** |
GRU-deltaT | **0.8558 vs. 0.936** |  0.9088 vs. 0.913 | **0.8496 vs. 0.904** |
N CDE |  **0.9276 vs. 0.987** | **0.9285 vs. 0.988** | **0.9241 vs. 0.986** |
ODERNN | 0.9415 vs. 0.954 | 0.9506 vs. 9.960 | 0.9471 vs. 0.953 |

These are serious differences on the same dataset. I can maybe explain away the case with Neural CDE as the creator of the method may be able to tune it better (this is in fact the main drawback of this entire comparison table idea we do, but this is a problem we are not going to solve here :) But results of several other baselines are quite different. This is concerning.

"We generated a dataset of 300 2-dimensional spirals, sampled at 150 equally-spaced time points" <-- what is the temporal dimension? is this CD spiral a parametric curve where time is the parameter?

Are you using vanilla NeuralODE in this experiment? I find it hard to understand why you get the interpolation behavior from NODEs as it is visible in Figure 2. The interpolation looks like piece-wise linear. Can you provide interpolation results with ODE-RNNs and neural CDEs?

Do you see a way to avoid resampling between layers?

It would be useful and quite interesting to see some discussion about the apparent contradiction in using transformers (long range interactions) and ODEs. ODEs operates with derivatives and their solutions are locally described. The assumptions of a smooth differentiable solution seems to stand in contradiction with long range interactions unmediated by values in between, or in a case of a latent-ODE decriptions by latent system states. Can you formalize the properties of systems (e.g. the mentioned stock market time series) where you expect this type of modelling give the most benefit?


Minor:

"To overcome the discrete nature of RNNs, a different strategy involves the exponential decay of
the hidden state between observations, governed by modifiable decay parameters [5, 9, 37]."  [9] is the reference for the Neural ODE paper from Chen at al., it does not apply exponential decay.

typo:  denoted vs donated

Please stress the difference between irregularly sampled data and temporal point processes eg. in [37] more clearly (missing data vs. events).

NOTE: the authors clarified some points, provided new results, and we identified a misunderstanding about the Neural ODE figure. The authors used Latent-ODE, but cited a previous paper in the table: that lead me believe that they fitted a Vanilla NODE to the trajectory. I raised my score by one.




**Limitations:**

No systematic discussion on limitations provided.

---

> ### Author Rebuttal · Authors · 2023-08-10
>
> > **W1: Table 3 covers 20 datasets, hard to rigorously evaluate the average metrics.**
>
> 1. We would like to clarify that our benchmarking follows the prevalent common practice [1, 2], where the averaged metrics are over all the datasets in the UEA benchmark.
> 2. The experiments on irregular time-series data follow the settings of Neural CDE [3], [3] only evaluated one of the UEA datasets while we evaluate over 20 datasets. The full results are in Appendix E.2.2.
>
> > **W2: The results in the appendix lack standard deviation (std.)**
>
> We quite appreciate your reminder and *we have included the standard deviation of results for the UEA benchmark in Table 1 of the attached PDF file*, on the setting when 70% of observations are dropped, due to the word limit. We will incorporate the std. results for all three settings in the revised paper. Please refer to **Q2 in General Response** for more details.
>
> > **W3: With 3 repeats, not sure 2 std is very convincing.**
>
> For the statistical significance in Sec. 4.3, we further perform experiments on Synthetic and Bookorder datasets, with *10* random seeds for each method. We report the complete experimental result in the attached PDF file, Table 2 and Table 3. Also, we conduct a significance test. We can conclude that when the p-value is less than $10^{-6}$, Contiformer can significantly outperform these baselines in 4 out of 6 metrics, i.e., LL and RMSE for Synthetic, LL, and Accuracy for BookOrder. In the revised manuscript, we will conduct a significance test on all the datasets. Please refer to **Q2 in General Response** for more details.
>
> > **Q1: The results of several baselines are different from the original paper**
>
> Thanks for your concern. After carefully checking the original paper (Kidger et al. https://arxiv.org/pdf/2005.08926.pdf) we find a significant difference in data setting. In both ours and Neural CDE paper, the raw dataset was randomly dropped by the ratio of 30%, 50%, and 70% to conduct the irregularly sampled dataset. In Section 4.1 of Kidger et al., "The randomly removed data is *the same* for every model and every repeat."; however, in our setting, we apply randomness on dropping the data points for each random run, which is more general to evaluate the robustness than on the same dataset dropped only once. This explains that our evaluation results of rerunning the baseline methods are different from those in the Neural CDE paper.
>
> Next, we checked the implementation details of these ODE-based baselines, which are the same as the official repository of [Neural CDE](https://github.com/patrick-kidger/NeuralCDE/tree/master/experiments/models).
>
> We also put openly available implementations in the original submission to ensure transparency and fairness. We will also revise our paper to clarify this.
>
> > **Q2: What is the temporal dimension? is this CD spiral a parametric curve where time is the parameter?**
>
> Sorry for the confusion. Temporal dimension means the variable $t$. This 2D spiral is a parametric curve. As in Eqs. (47, 48) in Appendix E.1.1, the 2D spiral function $(x(t), y(t)) \in \mathbb{R} \rightarrow \mathbb{R}^2$ is dependent on the time variable $t$, with the hyper-parameters $a, b$ controlling the curve shape.
>
> > **Q3: Are you using vanilla NeuralODE? In Fig. 2 why the interpolation looks like piece-wise linear?**
>
> Thanks for your valuable inquiry. Yes, we are using the vanilla NeuralODE implementation in our experiment. The reason why the interpolation from NODEs in Figure 2 looks like piece-wise linear is intrinsically from the more challenging data setting than the original one, as explained in **Q1 of the General Response** part.
>
> > **Q4: Do you see a way to avoid resampling between layers?**
>
> Insightful question! To our knowledge, neural networks can not directly handle continuous-time input while stacking multiple layers. To support stackable property, one can either discretize the output in continuous-time dimension, like our resampling method or discretize the parameter space, like the state space model [4, 5].
>
> > **Q5: Can you formalize the properties of systems where you expect this type of modeling give the most benefit?**
>
> Our model hinges on the premise of continuous-time dynamics. We treat the irregularly-sampled data as a sequence of observations stemming from an underlying continuous-time process [3]. Moreover, we posit self-correlation among the observations [6]. Given these foundational assumptions, ContiFormer is poised to excel, capitalizing on the power of continuous-time modeling and inherent self-correlation within the data.
>
> > **Q6: "...decay parameters [5, 9, 37]." [9] is incorrect.**
>
> Thanks for the suggestion. Fixed in the revised version.
>
> > **Q7: difference between irregularly sampled data and temporal point processes e.g., in [37]?**
>
> Based on our understanding of your question, here "irregularly sampled data" refers to the irregularly sampled time series data, a kind of sequence of signal values sampled in irregular time intervals; And "temporal point processes" are a branch of modeling methods such as [37], which are often utilized to model event sequences that is a type of sequence containing discrete events happened at irregular time points.
>
> The differences are as below.
> |Aspect|Irregularly sampled time series data|Event sequence data (w/ temporal point processes)|
> |-|-|-|
> |Collection method| Irregularly sampled|Triggered by events|
> |Data type|Signal values|Event features|
>
> [1] A transformer-based framework for multivariate time series representation learning. KDD 2021.
>
> [2] Timesnet: Temporal 2d-variation modeling for general time series analysis. ICLR 2023.
>
> [3] Neural controlled differential equations for irregular time series. NeurIPS 2020.
>
> [4] Simplified state space layers for sequence modeling. ICLR 2023.
>
> [5] Efficiently modeling long sequences with structured state spaces. ICLR 2022.
>
> [6] Attention is all you need. NeurIPS 2017.

---

> > ### Comment · Reviewer_PCsf · 2023-08-11
> > **Answer to Bebuttal**
> >
> > I would like to thank the authors for their time to answering my (and other reviewers') questions.
> >
> > **W1**:Being "prevalent common practice " (which is hard to assess based on 2 papers) does not automatically mean good practice.
> > However, I appreciate the fact that if it is reported like this in previous works, you followed this way. Also you report individual results (now with error bars), which make it possible to check these. I agree with **jYof** on that, claiming "overall superiority" is a bit strong based on "average" superiority.
> >
> > The provided error bars and significance tests are appreciated.
> >
> > **Q1**: This may explain the difference. I am not sure however that rerandomizing for all test is the better approach. In a real life scenario the data is given, so there is no possibility to execute these repeats, while random restarts can be performed even in a real life scenario. So, keeping the data fixed is more lifelike. But your evaluation is still valid of course.
> >
> > **Q3 (General Q1)**: **[my major remaining problem]** We (you, **2E9G** and myself) would be in an easier position if you would have added code to the supplementary.
> > You mention you use the same code as the original. Can you point to this original (git of original paper's repo, specific file describing the experiment setup most similar to your special experiment setup with the spiral - Be careful **not** to link own repo due to blind review!)
> > My problem is, even if you use the same model code, it is still possible that the model used differently during experimental setup.
> > Can you describe how you train and interpolate with NODE in a few points? Like: "I take the 2D time series data (x(t), y(t)), initialize a NODE with 2 hidden state, with XY architecture.., I propagate back XY error (MSE I assume)".. and so on.
> >
> > **Q5: [second remaining issue]** What you wrote in your reply, you already properly stated in the paper. I understand this. My question was: isn't there a contradiction here? You write for example for **jYof**'s question 1 that you assume ODE like dynamics is fundamental in your datasets, this is why transformers are not enough. Now this type of dynamics are by definition smooth and local (in time), therefore why to expect that the Markov assumption is violated in your datasets in the same time, when you assume an ODE structure is a good prior. Now I do not claim this type of data does not exist. I just asking: Can you give a convincing example?

---

> > > ### Author Response · Authors · 2023-08-13
> > > **Rely to your remaining problems**
> > >
> > > Thank you for your quick reply and explanation.
> > >
> > > **Reply to W1**:
> > >
> > > We agree that not all common practices meet standards. Error bars and significance tests will be put into our revised version. We are also committed to tempering the assertiveness of our claims.
> > >
> > > **Reply to Q1**:
> > >
> > > We understand the importance of realism in fixed data outcomes. Your input has spurred us to enhance our evaluation methodology.
> > >
> > > **Reply to Q3**:
> > >
> > > > **Point to the repo with a similar experiment setting.**
> > >
> > > We follow the code from [link](https://github.com/rtqichen/torchdiffeq/blob/master/examples/latent_ode.py) as mentioned in **General Response Q1**.
> > >
> > > For spiral data generation, following Line 31-105, where we modify Line 86-98, to generate irregularly-sample data and add noise to parameters $a$ and $b$ as claimed in the paper and Appendix E.1.2.
> > >
> > > For model training and implementation, we follow the code in Lines 181-194 to calculate the loss function and use the same code as Lines 108-159 to build the Neural ODE model.
> > >
> > > > **Describe your special experiment setup of NODE?**
> > >
> > > The training flow of Neural ODE is as follows: given a batch of 2D spirals, i.e., (200, 50, 2) for (batch size, sequence length, feature dimension).
> > >
> > > We use a LatentODEfunc with 3 hidden layers with ELU activation function and set the hidden dimension to 20.
> > >
> > > The model contains a RecognitionRNN to encode the input, followed by ODE to reconstruct the hidden state trajectory.
> > >
> > > Finally, a decoder with 2 hidden layers and ReLU activation is adopted to obtain the output.
> > >
> > > We train Neural ODE with ELBO loss as stated in Appendix E.1.3 and keep it the same as the original code.
> > >
> > > To better address your concern, we dig into the issue of why Neural ODE generates piece-wise linearity in our data setting. The main conclusions are as follows:
> > >
> > > 1. Sampling Strategy: Line 96 of the original implementation used regular sampling. Oriented toward irregular time series, we adopted irregular sampling. Unexpectedly, this may reveal linear piece-wise patterns. The impact of sampling is noted by Reviewer 2E9G (Q2).
> > >
> > > 2. Random Seed: We ran our experiments 10 times. Notably, Neural ODE may yield smooth outputs in some random tests. We suspect this might stem from training uncertainty, potentially influencing the linearly piecewise behavior. We are ready to provide the visualization by anonymous link (not recommended by NeurIPS official instruction email.)
> > >
> > > Overall, among all the random runs, we observe that the prediction result of Contiformer can outperform Neural ODE with p-value $< 10^{-6}$ in significant tests.
> > >
> > > Moreover, we alter these data settings to enhance the evaluation of irregular time series modeling and to effectively showcase method performance, as detailed in the General Response. We hope that all these explanations would help address your questions.
> > >
> > >
> > > **Reply to Q5**:
> > >
> > > > Any contradiction here? Can you give a convincing example?
> > >
> > > Thank you for your further explanation!
> > >
> > > We first express our understanding of your mentioned "contradiction". On one hand, ODE assumes data following Markov property and models the dynamical changes, which is "smooth and local" as you mentioned. Meanwhile, Transformer assumes that the data share both short-term and long-term dependencies, which seems "rough and global" and violates Markov assumptions.
> > >
> > > If it's correct, we kindly provide some examples below.
> > >
> > > 1. From a practical view, stock prices initially exhibit smooth auto-regressive patterns but are also affected by business conditions and market events. For instance, large technical companies' stock prices (e.g., MSFT and GOOG) display consistent evolving trends yet are impacted by short-term and long-term events such as the release of large language models, etc. Similarly, traffic data display both local similarities within short periods and global seasonality over longer periods (Fig. 4 in [1]). That intuitively explains why the time series will follow both the Markov property that ODE models and the global influence properties that Transformer models.
> > >
> > > 2. From the literature view, one of our baselines, the Transformer Hawkes Process [2] assumes the Hawkes process which is locally defined on the infinitesimal time interval. Besides, [2] also claims the shortcomings of Hawkes process assumption that "fails to capture complicated short-term and long-term temporal dependencies". Thus, they also incorporated other approaches leveraging long-term dependencies to close the gap.
> > >
> > > In conclusion, in our paper, we want to construct a system that is smooth in time span (like ODE-based methods) while capturing long-term dependency using powerful Transformer architecture.
> > >
> > > We hope our provided examples can address your questions. And we will also refine our description of our assumption part in our paper correspondingly.
> > >
> > > [1] HetETA: Heterogeneous information network embedding for estimating time of arrival. KDD 2020.
> > >
> > > [2] Transformer hawkes process. ICML 2020.

---

> > > > ### Comment · Reviewer_PCsf · 2023-08-14
> > > > **Will raise my score one point.**
> > > >
> > > > First, I would like to apologize, that I did not spot the link in your General answer when I first read it.
> > > >
> > > > However we had a bit of a misunderstanding, you used **Latent-ODE** from Rubanova et al, to make that spiral figure. What I mean by **Vanilla NODE**, is really that, just and end-to-end trained NODE from Chen et al. You can just fit a NODE as it would be a regular ODE just now with a black box form, to the trajectory, no EncoderRNN. In this case i would have been surprised by this behavior. In case of Latent-ODE we are on the same page, the result is of course perfectly possible.
> > > >
> > > > **Note that** in Table 2 in the manuscript you cite [9] Chen et al. not Rubanova et al.. Different papers, mostly same authors. Please clarify this.
> > > >
> > > > On the examples: Your event based disruption example on stock market is true, but it still not tell me why Transformer would solve this, it is not that this event can be predicted by attending the past, it is a disruption hard to predict.
> > > >
> > > > Still, we managed to clear up some of my concerns, and your method can be of interest for the community. I respect the work you spent answering my concerns. I will raise my score by one.

---

> > > > > ### Author Response · Authors · 2023-08-16
> > > > > **Reply to your remaining problems**
> > > > >
> > > > > > **Q1: You used Latent-ODE from Rubanova et al, to make that spiral figure. In Table 2 in the manuscript you cite [9] Chen et al. not Rubanova et al.. Please clarify this.**
> > > > >
> > > > > Just as you understand, we use the Latent ODE model with an RNN encoder and Neural ODE decoder to make that spiral figure.
> > > > >
> > > > > However, it is worth noticing that the Latent ODE model in our experiment is truly from Chen et al. [9]. Specifically, in Section 5 of [9], titled "A generative latent function time-series model", the author introduce the Latent ODE model.
> > > > >
> > > > > Moreover, we have conducted a comparison experiment on the Latent ODE model from Rubanova et al. [42], where it replaces the RNN encoder with an ODE encoder. This model is named ODE-RNN in our paper.
> > > > > The visualization result of this model from Rubanova et al. [42] has been shown in the attached PDF file (Fig. 2 ODE-RNN) in **General Response** part.
> > > > >
> > > > > We acknowledge that the use of "Neural ODE" in Table 2 is indeed confusing, we will replace the term with "Latent ODE" in the revised manuscript, but there should be no error in the citation as we understand.
> > > > >
> > > > > > **Q2: You can just fit a NODE as it would be a regular ODE just now with a black box form, to the trajectory, no EncoderRNN.**
> > > > >
> > > > > Thanks for your valuable suggestion. It is indeed possible to fit a regular ODE with an initial point (possibly the first observation) and reconstruct the latent trajectory by an ODE solver. We have also conducted extensive experiments to compare Neural ODE (w/o RNN enc), Latent ODE (w/ RNN enc), and our method Contiformer. The results are listed below with $\alpha=0.02$ and $\beta=0.1$, which follow the settings of Table 2 in our paper. From the new result table below, we can conclude that Neural ODE w/o RNN encoder yields poor prediction results, especially for *extrapolation*. Besides, from the visualization result, we can indeed observe a smooth outcome from Neural ODE.
> > > > >
> > > > > | Model | Interpolation (RMSE) | Interpolation (MAE) | Extrapolation (RMSE) | Extrapolation (MAE) |
> > > > > |-|-|-|-|-|
> > > > > | Neural ODE (w/o RNN enc) | 0.0140 ± 0.0010 | 0.0138 ± 0.0010 | 0.0259 ± 0.0021 | 0.0269 ± 0.0020 |
> > > > > | Latent ODE (w/ RNN enc) | 0.0209 ± 0.0022 | 0.0195 ± 0.0025 | 0.0159 ± 0.0005 | 0.0152 ± 0.0005 |
> > > > > | Contiformer | **0.0049 ± 0.0006** | **0.0052 ± 0.0006** | **0.0064 ± 0.0009** | **0.0065 ± 0.0008** |
> > > > >
> > > > > > **Q3: The example in the stock market still does not illustrate why Transformer would solve this, it is not that this event can be predicted by attending the past, it is a disruption hard to predict.**
> > > > >
> > > > > Thank you for your inquiry. Predicting disruptions in the stock market is undoubtedly challenging, especially considering the underlying complex market behavior. Nevertheless, we believe that the sequence patterns before the event may share some similarity of correlation to those historical sequences before those historical events, which may somehow help future prediction.
> > > > >
> > > > > Furthermore, we also provided another illustration of traffic analysis in our previous comment. We highlight the importance of local similarities within short periods and the influence of global seasonality over longer periods. In Section 4.3, we empirically showcase Contiformer's strength on the Traffic dataset, underscoring its ability to effectively capture these intricate and compound dependencies. This highlights the value of constructing a system that spans time smoothly and meanwhile still capturing long-term dependencies. This approach may hold promise for modeling irregular time series effectively in such a scenario.
> > > > >
> > > > > [9] Chen, Ricky TQ, et al. "Neural ordinary differential equations." NeurIPS 2018.
> > > > >
> > > > > [42] Rubanova, Yulia, Ricky TQ Chen, and David K. Duvenaud. "Latent ordinary differential equations for irregularly-sampled time series." NeurIPS 2019.
> > > > >
> > > > > ***
> > > > >
> > > > > In short, we sincerely thank you for raising the rating score regarding our paper. We are also open to further comments and suggestions for improving our paper. We will follow your suggestion to improve the quality of our paper.

---

### Official Review · Reviewer_v819 · 2023-07-06

**Soundness:** 1 poor
**Presentation:** 2 fair
**Contribution:** 1 poor
**Rating:** 5
**Confidence:** 2

**Summary:**

The paper proposes a new deep learning model called ContiFormer to model continuous-time dynamics on irregular time series data. The paper argues that existing methods such as recurrent neural networks (RNNs), Neural Ordinary Differential Equations (ODEs), and Transformers have limitations in modeling continuous-time data and fail to capture intricate correlations within these sequences. The proposed ContiFormer model extends the relation modeling of the Transformer model to the continuous domain and incorporates the modeling abilities of continuous dynamics of Neural ODE with the attention mechanism of Transformers. Both numerical and theoretical justifications are provided.

Different from the standard Transformer model, the proposed model is parameterized on the first-order derivative of $q$, $k$, and $v$. The real $q$, $k$ used in the attention part is approximately computed with finite sum (e.g., via Runge-Kutta 4th (RK4) order ODE solver). By restricting the function form/class of the first-order derivative on some smooth enough kernel, the proposed model could yield better interpolation/extrapolation ability.

I have two concerns about the proposed model.

The first one is the usage of ODE solvers may introduce computation overhead. In Table 1, the order of computation cost is already $O(N^2 \cdot P \cdot d^2)$, where $P$ is the iterative number in RK4 solver, and in my understanding, $P$ should be 4. In L580 at Appendix B. the RK4 ODE may require 80 forward passes to finish one of the RK4 round. It implies the order of computation cost for a one layer attention model should be $O(N^2\cdot 80\cdot d^2)$, which can be very computationally expensive even for models with moderate size. Moreover, if the proposed model has more than one layer (e.g., 2 layers), in order to finish the second layer's attention computation, we will need first to finish the computation of the first layer with $O(N^2\cdot 80\cdot d^2)$ cost. After that, every second layer's computation will still involve the forward pass of the first layer, which becomes $O(2\cdot N^2\cdot 80\cdot d^2)$. Therefore, for the $M$ layer attention model, the total computation cost could be $O(M^2\cdot N^2\cdot 80\cdot d^2)$.

The second one the underlying construction is kind of similar to the Rotary Position Embedding in [1]. In this paper, the kernel trick is used to facilitate the computation (e.g.,  Eq. (27)- Eq. (32) in Appendix C). Based on the current presentation, the kernel function is only time point $t$ dependent but independent on the detailed $q,k$. Given the choice of the kernel function in Eq. (30), the effects of introducing this kernel function look like just adding a rotary positional embedding.

At present, the authors have not provided sufficient evidence to demonstrate the contribution of their model, which may not be compelling enough for top machine learning conferences like NeurIPS. Despite this, the reviewer is willing to reconsider the decision after the authors' rebuttal. Overall, the paper presents an interesting approach for modeling continuous-time dynamics, but the issues raised above need to be addressed before its acceptance for a top-tier conference.


Reference: [1] Su, J., Lu, Y., Pan, S., Murtadha, A., Wen, B., & Liu, Y. (2021). Roformer: Enhanced transformer with rotary position embedding. arXiv preprint arXiv:2104.09864.






**Strengths:**

The paper proposes a new deep transformer model incoperating the continuous-time dynamics on irregular time series data. Both numerical and theoretical justifications are provided.

**Weaknesses:**

Please see my comments in the Summary section.

**Questions:**

Please see my comments in the Summary section.

**Limitations:**

The authors include a section on the limitations of the proposed work in Appendix G at Page 30.

---

> ### Author Rebuttal · Authors · 2023-08-10
>
> Thank you for your comprehensive comments.
>
> > **Q1: The order of computation cost one layer attention should be $O(N^2 \times 80 \times d^2)$; for the $M$ layer attention model, it could be $O(M \times N^2 \times 80 \times d^2)$.**
>
> Thank you for your inquiry about time complexity. In Table 1 of our paper, we initially specified the computation cost as $O(N^2 \times P \times d^2)$, where $P$ represented RK4 solver iterations.
>
> Upon re-evaluation, we acknowledge that the actual number of forward passes relates to the product of RK solver order and the number of segments across the reprameterized variable interval $[-1, 1]$. This leads to a refined time complexity of $O(N^2 \times \tilde{P} \times d^2)$, where $\tilde{P}$ signifies true forward pass counts.
>
> Moreover, in our time complexity analysis in our paper, we focus on the computation within one individual layer, as stated in the caption of Table 1 and the corresponding text description in our paper. We assume that all the compared models can be stacked with multiple layers. Thus, we omit the variable of the layer number in this analysis.
>
> Thank you for your comment. In the revised manuscript, we are committed to refining Table 1 to ensure its alignment with our findings and discussions.
>
> > **Q2: The underlying construction is kind of similar to the Rotary Position Embedding in [1], where the kernel trick is used to facilitate the computation (e.g., Eq. (27)- Eq. (32) in Appendix C). The effects of introducing this kernel function look like just adding a rotary positional embedding.**
>
> We are grateful for your question, which provides an opportunity to clarify the distinctions between our paper and Roformer [1]. We are confident that our work stands out from Roformer due to several essential reasons.
>
> The first clarification is that the discussion from Eq. (27)- Eq. (32) in Appendix C in our paper is about the Kernalized Attention mechanism, which is *not* to categorize our method as Kernalized Attention, but rather to demonstrate that Kernalized Attention is a form of Transformer attention variant. In fact, we mathematically prove that, as stated in Lines 583-584 in Appendix C, with carefully designed function hypotheses, various Transformer variants (including those with Kernelized Attention) can be encompassed as special cases of ContiFormer.
>
> Consequently, given Roformer's utilization of a pre-defined "position" matrix, i.e., $\boldsymbol{R}_{\Theta, m}^d \in \mathbb{R}^{d \times d}$ in Eq. (15) of Roformer's paper. It falls into the category of the "Time Embedding Method", as defined in Eq. (29) in the Appendix of our paper. As evidenced by Theorem 1 in Line 643 of our Appendix, Roformer (as well as all the relative position embedding methods) can be seen as a specific instance of ContiFormer.
>
> Technically, ContiFormer stands apart from Roformer in the below two aspects.
>
> 1. We prioritize modeling intricate dependencies of the inputs and learning the continuous-time evolving mechanism in irregularly-sampled data through a novel parametric continuous-time attention mechanism, differing from Roformer's focus on integrating positional information into the attention mechanism.
>
> 2. Our attention mechanism (Eq. (3), Section 3.1) captures evolving relationships of the input which is input dependent and (continuous-)time-aware, beyond reliance on only categorical positional information in Roformer which cannot model the continuously evolving dynamics of the system in the irregular time-series data.
>
> We greatly appreciate your query, as it allows us to elaborate on these nuanced distinctions between our work and Roformer, providing a clearer understanding of our contributions.
>
> > **Q3: At present, the authors have not provided sufficient evidence to demonstrate the contribution of their model.**
>
> Our work introduces a continuous-time attention approach tailored to the continuity and self-related nature of irregular time series. The contribution is three-fold.
>
> 1. To the best of our knowledge, we are the first to incorporate a continuous-time mechanism into attention calculation in Transformer as shown in Section 3 of our paper, which is novel and captures the continuity of the underlying system of the irregularly sampled time-series data, as shown in our experiments.
> 2. To tackle the conflicts between continuous-time calculation in continuous attention mechanism and parallel calculation property of the Transformer model, we also provided a novel reparameterization method in Section 3.3, to divide the whole time horizon and map them into a fixed time range, and then we can parallelly execute the continuous time attention in the different time range, without hurting the model capacity (as shown in our experiments). This can provide another novel perspective to accelerate the calculation of continuous-time modeling in other works such as Neural ODE.
> 3. Notably, we mathematically proved in Section 3.4 and Appendix C.2 that, our proposed continuous attention mechanism is a universal attention approximator, and various Transformer variants including kernel-based methods can be viewed as special instances of our model. Our approach uniquely aligns with the intricate characteristics of irregular time series and offers a broader scope that encompasses Transformer variants, which can shed some light on the further exploration of continuous process modeling in the Transformer model.
>
>
> We thank you for your pointing out the confusion about our description. We will further refine our paper according to your valuable suggestions.
>
> [1] Roformer: Enhanced transformer with rotary position embedding. (2021).

---

> > ### Comment · Reviewer_v819 · 2023-08-15
> > **Thanks for your comments.**
> >
> > Thanks for the authors' comments. After reading the authors‘ feedback and other reviewers' comments. Most of my concerns are addressed and I'll increase my rate accordingly. My remaining concern is mainly about the computation cost, the usage of RK4 solver can be viewed as an implicit layer that is typically very time composing for large-scale problems or large-scale model configurations. However, given the capacity of modern deep-learning hardware and the scale of real-world time series problems, I believe the computation cost of the proposed method won't be a major issue. It would be great if the authors could include a stress test in the final version on the computation cost v.s. problem scale/model size and discuss the *sweet zone* of the proposed method.

---

> > > ### Author Response · Authors · 2023-08-16
> > > **Rely to your remaining problems**
> > >
> > > > **Q1: The usage of RK4 solver can be very time consuming.**
> > >
> > > Thank you for your insightful feedback. Utilizing an ODE solver, such as the RK4 solver, requires performing multiple forward passes, which can lead to computationally demanding operations.
> > >
> > > In our reply to reviewer GCXG, we can see that the time for training and validation is roughly $6$ times as vanilla Transformer model, which is somehow acceptable, with RK4 step size equal to $0.1$ and hidden dimension as $8$ in our experiment.
> > >
> > > We can also leverage some other techniques and packages to accelerate the ODE solver procedure, like diffrax as mentioned by reviewer 2E9G. These efforts are orthogonal to our work and we leave that as one important future work. Thank you for your suggestion.
> > >
> > > > **Q2: It would be great if the authors could include a stress test in the final version on the computation cost v.s. problem scale/model size and discuss the sweet zone of the proposed method.**
> > >
> > > Thanks for your valuable suggestion. Although we have measured the time cost on a single dataset, a stress test is necessary for a complete evaluation of the time cost and scalability of our model.
> > >
> > > In our revised manuscript, we are committed to addressing this aspect by incorporating a stress test that assesses how our proposed method performs across various input lengths and hidden sizes. Also, we intend to report the precise training/inference time for a wide range of datasets.
> > >
> > > ***
> > >
> > > In short, we sincerely thank you for raising the rating score regarding our paper. We are also open to further comments and suggestions for improving our paper. We will follow your suggestion to improve the quality of our paper.

---

### Official Review · Reviewer_2E9G · 2023-07-07

**Soundness:** 3 good
**Presentation:** 2 fair
**Contribution:** 3 good
**Rating:** 6
**Confidence:** 3

**Summary:**

This paper introduces ContiFormer, a continuous time transformer-based model that leverages parallelism and can handle irregularly sampled data well, thereby removing the need to transform these datasets into discrete uniform bins. This set-up incorporates the continuous dependence on the data from differential equation based neural networks. Finally, an extensive range of experiments on irregularly sampled data are performed to show promising results for ContiFormer.

**Strengths:**

This paper presents an original architecture class that can encompass many existing models. The discussion on attention combined with the continuous time set-up is limited and therefore this paper will be a significant contribution in this area. The paper is generally well-written and extensive numerical experiments were performed.

**Weaknesses:**

Whilst in general a fairly well-written paper, key parts of the model can be made clearer. For example, it is not clear for line 137 how the key and values are initialised. There is also an assumption that is not explained "we assume that every observation has influence on the dynamic system even before its occurrence." Combined with the fact that the query uses natural cubic spline, does that mean the model is not causal and cannot be applied in an online fashion (see for example [Morrill et al 2022 On the Choice of Interpolation Scheme for Neural CDEs])?

This paper does not appear to be comparing with the state-of-the-art models. To my knowledge, the S5 model (Simplified State Space layers for sequence modeling by Smith et al 2022), appears to already outperform quite a few of the baselines chosen in this paper, specifically, mTAN, ODE-RNN, GRU-$\Delta$t for irregularly sampled pendulum exercise. Therefore this should be added into a benchmark. The pendulum regression task used in Smith et al. seems to be used in quite a few irregular sampling task papers, and therefore it would be of interest to see how the ContiFormer performs in this case.

Whilst there is an extensive range of experiments, the full results with the standard deviation across the 3 repeats for Section 4.2 and Section 4.3 seem to be missing. These seem quite key to back up some results, particularly for Table 4, where the standard deviation is referred to. Without this, it is difficult to quantify "overall statistical superiority of ContiFormer" as claimed.

There does not appear to be any evaluation on regularly sampled datasets. Whilst I appreciate that the marketed strength of this method is in irregularly sampled data, it would be useful to see what the performance is on regularly sampled datasets (against benchmarks) and whether one can just choose this model to be applied on all timeseries.

**Questions:**

Since ContiFormer requires $N^2$ ODEs to be solved, how does the $P$ relate to $L$ in Table 2? Are these on similar orders?

It seems like the neural ODE performs much worse than expected even for the interpolation compared with the example we see on the diffrax website https://docs.kidger.site/diffrax/examples/neural_ode/. Is this because of the regular vs irregular sampling issue?

Figure 4 in appendix does not seem to be plotting the same curve at each row? Neural ODE clearly is on a different scale or a different curve in the third row quite clearly?

Minor point: Line 119: donated -> denoted

**Limitations:**

Limitations were not discussed in the paper.

Given that the model makes use of cubic splines, the literature suggests this is will not be causal. Performance on regularly sampled dataset not clear.

---

> ### Author Rebuttal · Authors · 2023-08-10
>
> We sincerely thank you for your comprehensive comments.
> > **W1: L137 how key & values initialised.**
>
> As in L124, the input ($Q$), $K$, and $V$ are initially derived from the input variable $X$. Besides, we also explain in Equations (7) and (8) where $K_i = X_i W^K$ and $V_i = X_i W^V$. Moreover, the evolution of $k_i$ and $v_i$ is controlled by a function $f(\cdot): \mathbb{R}^{d+1} \rightarrow \mathbb{R}^d$, which is implemented as an MLP as elaborated in Appendix F.4.
> > **W2: L138, an assumption that is not explained.**
>
> We sincerely apologize for any confusion. Each observation's latent trajectory, rather than influence, spans the entire time horizon, as exemplified in Figure 1. Our method will further model the correlation among the underlying latent trajectories in the evolving system, which closely resembles the operation of Transformer, where each token computes attention scores across all tokens along the temporal axis. For auto-regressive prediction scenarios, we can apply causal attention with a masking mechanism, similar to vanilla Transformer, to avoid mistakenly leveraging future information.
> > **W3: Can Contiformer be applied in an online fashion?**
>
> Thank you for your great point! We indeed use cubic spline interpolation, which might limit its application to online fashion. However, we want to clarify that the core architecture is not limited to the interpolation method. Therefore, we can explore alternatives to support online applications [1].
> > **W4: The paper does not compare to SOTA models.**
>
> We apologize for the oversight in comparison to this model. Due to the limited timeline, we leverage the public implementation of S5 model, i.e., [s5-pytorch](https://github.com/i404788/s5-pytorch), and derive the experimental comparison on the benchmark of UEA Classification in Section 4.2.
>
> Our model Contiformer exhibits a notable performance advantage over S5 UEA task. Besides, we believe that, with a wider range of hyperparameter searches, S5's performance might be improved.
>
> |Drop Ratio (%)|30||50||70||
> |-|-|-|-|-|-|-|
> |Model|S5|ContiFormer|S5|ContiFormer|S5|ContiFormer|
> |Avg. ACC|0.7139|**0.8126**|0.6831|**0.7997**|0.6455|**0.7749**|
> |# Top 1 |2|**18**|2|**18**|1|**19**|
>
> > **W5: The pendulum regression task should be included.**
>
> Due to the limited timeline, we do not perform much parameter search and use most of the default hyper-parameter settings from the UEA task. The preliminary results are shown below. We run the experiment 3 times and report the mean and standard deviation.
>
> |Model|MSE ($\times 10^{-3}$)(std.)|
> |-|-|
> |ODE-RNN|7.26(0.41)|
> |CRU (original)|4.63(1.07)|
> |CRU (in S5 paper)|3.94(0.21)|
> |S5|**3.41**(0.27)|
> |Contiformer|4.21(0.24)|
>
> We wish to emphasize that ContiFormer showcases superior performance compared to the S5 model within the UEA classification benchmark as shown in the response to **Q4** above.
> > **W6: The standard deviation result is missing for sections 4.2 and 4.3.**
>
> We recognize the importance of incorporating standard deviation as a vital component for robust statistical analysis. We report the results in the attached PDF file (Table 1 for section 4.2, Table 2 for section 4.3). Moreover, it's important to note that for the UEA task in Section 4.2, our reporting approach aligns with the prevalent practice in the field of time series analysis [2， 3], where the overall performance of the mean accuracy and mean ranking are reported. Please refer to **Q2 in General Response** for more details.
> > **W7: Missing evaluation on regularly sampled datasets.**
>
> We reevaluate our model and other baselines on a regularly sampled time-series setting (i.e., with the drop-ratio as 0%). We also include an additional strong baseline Autoformer [4], which is specifically designed for regular time-series data.
>
> |Model|ODE-RNN|Neural CDE|Autoformer|TST|Contiformer|
> |-|-|-|-|-|-|
> |**Avg. Rank**|3.5|4.2|3.3|2.1|**1.85**|
>
> We also want to recall that Contiformer has illustrated superiority over all the baselines in the irregularly sampled data setting, as shown in Table 3 of our paper.
> > **Q1: How does the $P$ relate to $L$ in Table 2? Are these on similar orders?**
>
> $L$ represents the number of function evaluations that the ODE solver requests in a single forward pass from $t_1$ to $t_N$, which is related to the sequence length $N$ as well as the time difference $(t_N - t_1)$. While $P$ is the number of intermediate steps for integral approximation, which is a constant that is not related to either $N$ or the time difference $(t_N - t_1)$. Therefore, they are not on the same order and $P \ll L$. Also, as is shown in L224, we set $P \leq 5$ in our experiments.
> > **Q2: Neural ODE performs worse for interpolation.**
>
> We want to clarify that the codes in our experiments are almost the same as that in the original implementation [Neural ODE repo](https://github.com/rtqichen/torchdiffeq/blob/master/examples/latent_ode.py), we have kept most experimental settings the same. We modify the data generation process (a more comprehensive response can be found in **Q1 in General Response** section), and as a result, Transformer and Neural ODE may yield poor interpolation. In contrast, Contiformer is more robust with better visualization. More visualization results are uploaded in the attached PDF file. Sorry for the confusion and we will clarify the details of this experiment in our revised paper.
> > **Q3: Incorrect Figure 4 in the Appendix.**
>
> Apologies for the confusion. We have uploaded a figure in the attached PDF file. Please refer to **Q1 in General Response** for more details.
>
> [1] On the choice of interpolation scheme for neural CDEs. TMLR 2022.
>
> [2] A transformer-based framework for multivariate time series representation learning. KDD 2021.
>
> [3] Timesnet: Temporal 2d-variation modeling for general time series analysis. ICLR 2023.
>
> [4] Autoformer: Decomposition transformers with auto-correlation for long-term series forecasting. NIPS 2021.

---

> > ### Comment · Reviewer_2E9G · 2023-08-15
> >
> > Thank you to the authors for responding to some of the points that I raised and my questions, in particular, for doing further experiments and some comparisons with S5 and testing on the pendulum task in such a short time period.
> >
> > With regards to online applications, I would still like to check its practical implementation. Of course I see that the interpolation methods can be changed and you have done an ablation study responding to **jYof** to show that the results may not be so sensitive. In terms of your parallelization, it seems that each irregular time period is rescaled to $[-1,1]$ first and then functions are suitably transformed. The point of this seems to be "decoupling" the problem along the time axis, is there anything to ensure continuity in time?

---

> > > ### Author Response · Authors · 2023-08-16
> > > **Rely to your remaining problems**
> > >
> > > > **Q1: With regard to online applications, I would still like to check its practical implementation.**
> > >
> > > Thanks for your inquiry. We acknowledge that our current implementation using natural cubic spline may suffer in online applications. However, we can make the following change in our codes to support interpolation methods like Cubic Hermite splines with backward differences [1], as you suggested.
> > >
> > > For implementation, given the inputs matrix $Q$ and the corresponding time for each observation $T$, we will invoke `coeffs=torchcde.natural_cubic_coeffs(Q, t=T)` to construct the continuous path. Therefore, in the case of online applications, we can replace it with `coeffs=torchcde.hermite_cubic_coefficients_with_backward_differences(Q, t=T)`.
> > >
> > > > **Q2: It seems that each irregular time period is rescaled to $[-1, 1]$ first and then functions are suitably transformed. The point of this seems to be "decoupling" the problem along the time axis, is there anything to ensure continuity in time?**
> > >
> > > Thanks for your great point! The rescale operation used in Eq. (9) can enable the parallel computation of the calculation over time. However, this operation may induce additional numerical errors. Therefore, unfortunately, there is no guarantee to ensure continuity in the time axis. However, we would like to highlight that there are two factors that can help control the error, and therefore relieve influence of the continuity issue of the latent trajectories in time.
> > >
> > > 1. Error in ODE Solver: When utilizing a numerical ODE solver, inherent errors may arise during the solution process. Nevertheless, the mitigation of such errors is attainable by adopting a smaller step size or error tolerance, the errors can be effectively controlled [2].
> > >
> > > 2. Error in numerical approximation. As shown in Eq. (9) of the paper, we use a numerical approximation method (e.g., Gauss-Legendre Quadrature approximation) to approximate an integral from $[-1, 1]$. However, the approximation error can be bounded. Besides, we can increase $P$, which is the number of intermediate steps for integral approximation, to achieve a lower approximation error.
> > >
> > > Overall, as we discussed in Appendix D Line 658-660, the output of ContiFormer can be considered “continuous” only if we overlook the approximation and numerical errors in the ODESolver. Also, our framework allows the user to trade off speed for precision.
> > >
> > > Furthermore, the empirical findings presented in Section 4 illustrate that Contiformer consistently delivers commendable performance across diverse tasks with numerical error.
> > >
> > > In the revised manuscript, we will provide a more detailed discussion about the continuity of the Contiformer. We hope these explanations have solved your concern.
> > >
> > > [1] On the choice of interpolation scheme for neural CDEs. TMLR 2022.
> > >
> > > [2] Chen, Ricky TQ, et al. "Neural ordinary differential equations." NeurIPS 2018.

---

> > > > ### Comment · Reviewer_2E9G · 2023-08-18
> > > >
> > > > Thank you to the authors for addressing my remaining questions. I maintain that this is a paper with a novel method which will have moderate-to-high impact and as such should be accepted to neurips. My score remains at 6.

---

> > > > > ### Author Response · Authors · 2023-08-19
> > > > > **Thank you for your reply!**
> > > > >
> > > > > We sincerely thank you for your positive comment regarding our paper. In our final version, we are committed to integrating supplementary experiments to more comprehensively showcase our model's capabilities. Moreover, we are dedicated to augmenting the lucidity of our explanations to ensure a clearer presentation.

---

### Author Rebuttal · Authors · 2023-08-10

## General Response

We thank all the reviewers' valuable and insightful suggestions! And we are encouraged by positive comments from the reviewers, e.g.,
* Addressing import research problems with high practical value (Reviewer GCXG)
* The proposed method is novel with significant contribution in the area (Reviewer 2E9G, v819, GCXG, jYof)
* The paper is well-written and easy to follow (Reviewer 2E9G, GCXG)
* Experimental results are promising with theoretical analysis. (Reviewer 2E9G, GCXG, jYof)

Below we try to clarify a few concepts in our paper and address some common problems.

***

> **Q1: Concern about interpolation setting and visualization results in Section 4.1.**

As raised by reviewers 2E9G, PCsf. The interpolation result of Neural ODE in Fig. 2 of our paper is confusing. Here, we make the following explanation.

1. We want to clarify that the codes in our experiments are the same as that in the original implementation [Neural ODE repo](https://github.com/rtqichen/torchdiffeq/blob/master/examples/latent_ode.py), we have kept most experimental settings the same.
2. To better illustrate the relative performance of the compared methods, we slightly modified the ground-truth spiral data generation process by adding some random noise to the underlying spiral function hyper-parameters (i.e., $a$ and $b$ as discussed in Appendix E.1.1). The task becomes more difficult since the dataset incorporates some distributional shifts by this modification.

As a result, the compared baselines (Transformer and Neural ODE) could not handle this challenge well and illustrated poor interpolation. In contrast, our proposed Contiformer is more robust in this setting showing better interpolation performance with lower error (shown in Table 2) and better interpolation visualization (in Fig. 2) in our paper.

We also reproduce the experiment with *the same* spiral data generation in [Neural ODE repo](https://github.com/rtqichen/torchdiffeq/blob/master/examples/latent_ode.py), without changing any of the code, and we find that when Neural ODE is not well-trained, piece-wise linear results may still occur.

To address the concern, we have provided more visualization results in the attached PDF file (Fig. 1 and Fig. 2).

> **Q2: Missing of standard deviation for significant test in Section 4.2 and Section 4.3.**

As pointed out by reviewers 2E9G and PCsf, showcasing the results with standard deviations is crucial to establish the statistical excellence of Contiformer. We regret for our oversight on this part. We have included the standard deviation result in the attached PDF file (Table 1 and Table 2).

Moreover, it's important to note that for the UEA task in Section 4.2, our reporting approach aligns with the prevalent practice in the field of time series analysis [1, 2], where the overall performance of the mean accuracy and mean ranking are reported and compared.

What's more, as pointed out by reviewer PCsf, with 3 repeats, it is not sure 2 std. is very convincing. Therefore, it is suggested to check the unpaired t-test table. We really appreciate the valuable suggestion. However, due to the rebuttal timeline, we only manage to perform a complete t-test on Synthetic and BookOrder datasets, using *10* random seeds over Contiformer and 6 baseline models. The significant test results can be found in the attached PDF file (Table 3 and Table 4 for Synthetic and BookOrder datasets, respectively.) which illustrate the significant improvement of our method.

[1] A transformer-based framework for multivariate time series representation learning. KDD 2021.
[2] Timesnet: Temporal 2d-variation modeling for general time series analysis. ICLR 2023.

> **Q3: More detailed illustration of how we achieve parallelization; why reparameterize the time horizon to [-1, 1]**

In L199-207 of our paper, we state that to preserve the parallelization of Transformer architecture and meanwhile implement the continuous-time attention mechanism in Eq. (6), we first adopt time variable ODE [1] to reparameterize ODEs into a single interval [−1, 1], followed by numerical approximation method to approximate the integrals. Here, we believe that it is novel to incorporate the time variable ODE to achieve the parallelization of our model. We would like to discuss more about this novelty below.

Originally, to calculate the ODE process from $t_1$ to $t_N$ for $L$ latent ODE trajectories, we need to iteratively forward from $t_1$ to $t_N$, which is hard to parallel. To resolve the problem, rather than applying the ODE solver directly modeling along the whole time horizon, we split the time horizon and reparameterize the time range $[t_i, t_j]$ ($1\leq i \leq j \leq N$) as $[-1, 1]$. Then, the single invoke of the ODE solver will be applied to these time ranges since these trajectory pieces share the same ODE function parameters in Eq. (1) of our paper. Through this novel way, we can parallelly calculate these ODE functions without sequentially calculating from $t_1$ to $t_N$.

We believe that this is aligned with the parallel execution of attention calculation in the vanilla Transformer model (applying the same attention function on the copied & stacked input sequence to enable parallel execution), and it will further facilitate the parallel ability of our Contiformer model.

[1] Ricky T. Q. Chen, et al. Neural spatio-temporal point processes. ICLR, 2021.

---

### Decision · Program_Chairs · 2023-09-21

**Decision:**

Accept (poster)

**Comment:**

This paper introduces ContiFormer to handle irregular time series data. The innovation lies in a continuous-time attention module with the keys, queries and values are vector-valued functions indexed on time. Techniques such as neural ODE and natural cubic spline interpolations are used to compute these key, query and value vectors. Algorithms for speeding up the computations are also proposed. The ContiFormer model is evaluated on various time series tasks and datasets with very promising results.

Reviewers found the approach to be novel and the idea to be interesting in solving irregular time series modelling task. They raised a few major concerns regarding:
1. Missing baselines for some other time-series modelling approaches;
2. The computation time, after tricks presented in the paper, is still significantly higher.

In author feedback, the first concern is largely addressed due to the extra baseline results provided by the authors. The authors acknowledged the high run-time issue.

In final revision, I'd encourage the authors to include all the new baseline results as well as to make better clarifications regarding time complexity figures of the proposed approach.